# Learning to Integrate Diffusion ODEs by Averaging the Derivatives

**Wenze Liu**  **Xiangyu Yue**[*]

MMLab, The Chinese University of Hong Kong

## Abstract

To accelerate diffusion model inference, numerical solvers perform poorly at extremely small steps, while distillation techniques often introduce complexity and instability. This work presents an intermediate strategy, balancing performance and cost, by learning ODE integration using loss functions derived from the derivative-integral relationship, inspired by Monte Carlo integration and Picard iteration. From a geometric perspective, the losses operate by gradually extending the tangent to the secant, thus are named as secant losses. The target of secant losses is the same as that of diffusion models, or the diffusion model itself, leading to great training stability. By fine-tuning or distillation, the secant version of EDM achieves a 10-step FID of $2.14$ on CIFAR-10, while the secant version of SiT-XL/2 attains a 4-step FID of $2.27$ and an 8-step FID of $1.96$ on ImageNet-$256 \times 256$. Code is available at `https://github.com/poppuppy/secant-expectation`.

## 1 Introduction

Diffusion models [1, 2, 3, 4] generate images by reversely denoising their noised versions, which can be formulated by stochastic differential equations (SDEs) and the corresponding probability flow ordinary differential equations (PF-ODEs) [4]. Over recent years, diffusion models have transformed the landscape of generative modeling, across multiple modalities such as image [5, 6, 7, 8], video [9, 10, 11] and audio [12, 13, 14]. Despite their remarkable performance, a significant drawback of diffusion models is their slow inference speed: they typically require hundreds to thousands of number of function evaluations (NFEs) to generate a single image. Considerable research has focused on reducing the number of required steps, with common approaches falling into categories including faster samplers [15, 16, 17, 18] for diffusion SDEs or PF-ODEs, and diffusion distillation [19, 20, 21, 22, 23, 24, 25, 26, 27, 28, 29, 30, 31, 32, 33, 34, 35] methods that distill pretrained diffusion models to few-step generators. However, faster samplers experience significant performance degradation when operating with a small number of function evaluations (typically fewer than 10 NFEs). Meanwhile, distillation approaches frequently introduce substantial computational overhead, complex training procedures, or training instabilities. Additionally, they sometimes face risks of model collapse and over-smoothing in the generated outputs.

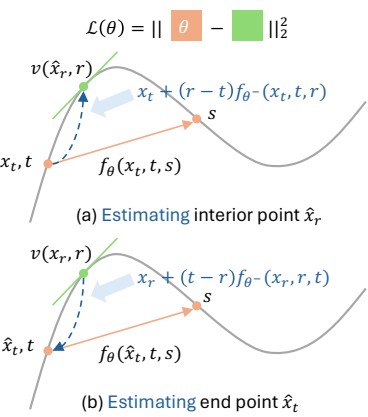

Figure 1: Formulation of secant losses. We employ an $\ell_2$ loss between the learned secant and a tangent at a random interior time point. By estimating either the interior or the end point, we derive two variants shown in (a) and (b).

---

[*]Corresponding author.

39th Conference on Neural Information Processing Systems (NeurIPS 2025).

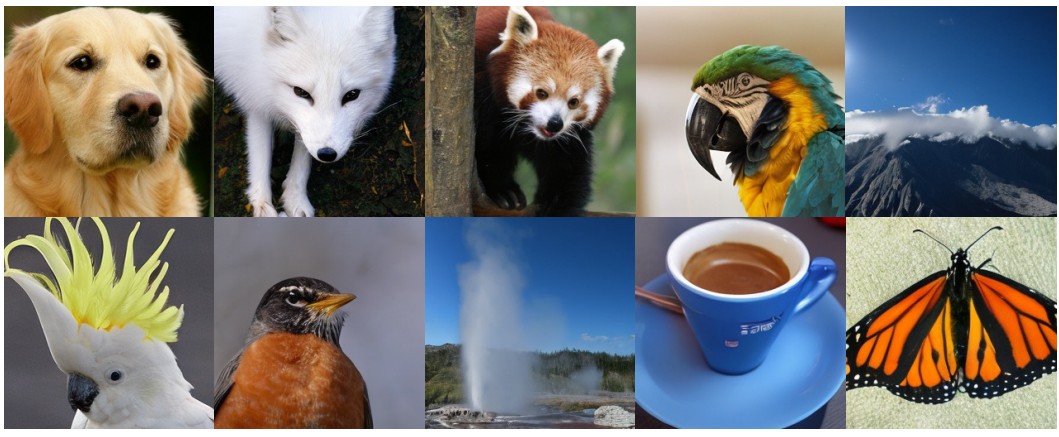

Figure 2: Selected 8-step samples on ImageNet $256 \times 256$.

In this work, we propose a simple intermediate solution to reduce the inference steps of diffusion models through easy distillation or fine-tuning. A diffusion model constructs the PF-ODE by fitting the derivative of the noised image with respect to time, as indicated by the green line in Fig. 1. From a geometric perspective, we refer to the diffusion model as the *tangent* since it describes the instantaneous rate of change of the noised image as the time difference approaches zero. Correspondingly, we define the rate of change of the noised image between two time points as the *secant*, marked in orange in the figure. Unlike diffusion models, we use neural networks to model the secant function instead. Based on the observation that the secant is the average of all tangents between two time points, we establish an integral equation as our loss function, termed *secant losses*. This average is implemented by taking the expectation with respect to random time variables in a Monte Carlo fashion. Since we can only sample one noised image for evaluating either the secant or the tangent at each training iteration, we let the model estimate the other, inspired by Picard iteration. Two scenarios for estimating the interior or the end point are shown in Fig. 1 (a) and (b), respectively. This results in a straightforward loss formulation: the distance between the learned secant with respect to two given time points and a tangent at a random intermediate time between them, as presented at the top of Fig. 1. Intuitively, secant losses work by gradually extending from the tangent to the secant. Unlike consistency models [33, 35, 34, 36], which either introduce discretization errors or rely on explicit differentiation in loss calculation, our approach avoids explicit differentiation while preserving accurate solutions at sufficiently small time intervals under mild conditions. More importantly, the target of our secant losses is identical to that of diffusion models or the diffusion model itself. This parallel to diffusion models provides significantly greater training stability compared with consistency models.

We evaluate our method on CIFAR-10 [37] and ImageNet-$256 \times 256$ [38] datasets, using EDM [6] and SiT [39] as teacher diffusion models, respectively. Our experiments demonstrate that diffusion models can be efficiently converted to their secant version, with significantly slower accuracy degradation compared to conventional numerical solvers as the step number decreases. On CIFAR-10, our approach achieves FID scores of 2.14 with 10 steps. For ImageNet-$256 \times 256$ in latent space, we obtain a 4-step FID of 2.78 and an 8-step FID of 2.33. With the guidance interval technique [40], the performance is further improved to 2.27 with 4 steps and 1.96 with 8 steps.

## 2   Related Work

**Few-step diffusion distillation and training.** Reducing inference steps in diffusion models is commonly achieved through distillation. As an application of knowledge distillation [41], direct distillation methods [19, 20] generate noise-image pairs by sampling from a pretrained diffusion model and train a one-step model on this synthetic dataset. Similarly, progressive distillation [21, 22] iteratively trains the models to merge adjacent steps toward a one-step model. These methods often introduce significant computational overhead due to extensive sampling or training costs. Adversarial distillation approaches [23, 24, 25] apply GAN-style losses [42] to supervise the one-step distribution,

potentially increasing training instability. Variational score distillation [26, 27, 28, 29, 30, 31, 32] and score identity distillation [43, 44] also employ distribution-level distillation, while introduce additional complexity due to the simultaneous optimization of two online models, and may potentially result in over-smoothing artifacts in the generated outputs. Consistency distillation [33, 34, 35] leverages the consistency property of PF-ODEs to train models to solve these equations directly, but faces stability challenges [33, 35] and may require model customization [35]. While direct training of consistency models [33, 45, 35] is feasible, they typically underperform consistency distillation under high data variance [35]. Similarly, Shortcut Models [46] utilize the consistency property as regularization in the loss function, which could be considered as simultaneously training and distillation. Rectified flow [47, 48] adopts a multi-time training strategy aimed at gradually straightening the trajectory of the PF-ODE. Similar to consistency models, our work also introduces losses for both distillation and training. However, our loss function emphasizes local accuracy, which enhances stability, though the accuracy decreases more at larger time intervals.

**Fast diffusion samplers.** Common numerical solvers, such as the Euler solver for PF-ODEs and the Euler-Maruyama solver for SDEs, typically require hundreds to thousands of NFEs to sample an image from diffusion models. Since SDEs involve greater randomness, samplers based on them generally require much more NFEs to converge [2, 49, 4, 6, 50, 51, 52, 53]. Consequently, existing fast samplers are usually based on PF-ODEs, including both training-free and learnable methods. Training-free methods leverage mathematical tools to analyze and formulate the solving process. Examples include high-order samplers like the Heun sampler [6], exponential samplers [15, 16, 17, 54, 55], and parallel samplers [56, 57]. Despite their convenience, these methods struggle in low NFE scenarios. Data-driven, learnable samplers enhance performance by training lightweight modules to learn hyperparameters in the sampling process, such as coefficients [58], high-order derivatives [59], and time schedules [18, 60, 61, 62, 63, 64]. However, both training-free and learned faster samplers still experience significant performance degradation when the step count falls below ten. Our method is also based on solving the diffusion ODE, and it degrades more gradually than conventional diffusion samplers as the step count decreases. Our approach can be seen as an effective compromise between fast samplers and diffusion distillation methods.

## 3 Diffusion Models

For simplicity, we review diffusion models within the flow matching framework [65, 47, 66]. Let $p_d$ denote the data distribution, and $\boldsymbol{x}_0 \sim p_d$ represent a data sample. Let $\boldsymbol{z} \sim \mathcal{N}(\boldsymbol{0}, \boldsymbol{I})$ be a standard Gaussian sample. The noised image with respect to $\boldsymbol{x}_0$ and $\boldsymbol{z}$ at time $t$ is defined as $\boldsymbol{x}_t = \alpha_t \boldsymbol{x}_0 + \sigma_t \boldsymbol{z}$, where $t \in [0, 1]$ represents the intensity of added noise. For a given neural network $\boldsymbol{v}_\theta$, the training objective is formulated as:

$$\mathcal{L}_{\text{Diff}}(\theta) = \mathbb{E}_{\boldsymbol{x}_0, \boldsymbol{z}, t} \|\boldsymbol{v}_\theta(\boldsymbol{x}_t, t) - (\alpha_t' \boldsymbol{x}_0 + \sigma_t' \boldsymbol{z})\|_2^2, \tag{1}$$

where $\alpha_t'$ and $\sigma_t'$ are time derivatives. An established result from the literature on diffusion and flow matching [65, 47, 66] is as follows:

**Proposition 1.** *When $\mathcal{L}_{Diff}(\theta)$ reaches the minimum, the optimal solution $\boldsymbol{v}_\theta^*(\boldsymbol{x}_t, t)$ is*

$$\boldsymbol{v}(\boldsymbol{x}_t, t) = \mathbb{E}_{\boldsymbol{x}_0, \boldsymbol{z}}(\alpha_t' \boldsymbol{x}_0 + \sigma_t' \boldsymbol{z} | \boldsymbol{x}_t). \tag{2}$$

*And the associated PF-ODE*

$$\frac{d\boldsymbol{x}_t}{dt} = \boldsymbol{v}(\boldsymbol{x}_t, t) \tag{3}$$

*is guaranteed to generate $\boldsymbol{x}_0 \sim p_d$ at $t = 0$ starting from $\boldsymbol{z} \sim \mathcal{N}(\boldsymbol{0}, \boldsymbol{I})$ at $t = 1$.*

## 4 Learning Integral with Secant Expectation

In this section, we first introduce how to parametrize the model as the secant function. We then explain our derivation of loss functions based on Monte Carlo integration and Picard iteration.

### 4.1 Secant Parametrization

Given the PF-ODE Eq. (3), to solve $\boldsymbol{x}_s$ at time $s$ starting from $\boldsymbol{x}_t$ at time $t$, one calculates

$$\boldsymbol{x}_s = \boldsymbol{x}_t + \int_t^s \boldsymbol{v}(\boldsymbol{x}_r, r) dr. \tag{4}$$

| **Algorithm 1** Secant Distillation by Estimating the Interior Point (SDEI) | **Algorithm 2** Secant Training by Estimating the End Point (STEE) |
|---|---|
| **Input:** dataset $\mathcal{D}$, neural network $\boldsymbol{f}_\theta$, teacher diffusion model $\boldsymbol{v}$, learning rate $\eta$ | **Input:** dataset $\mathcal{D}$, neural network $\boldsymbol{f}_\theta$, learning rate $\eta$ |
| **repeat** | **repeat** |
| $\quad \theta^- \leftarrow \theta$ | $\quad \theta^- \leftarrow \theta$ |
| $\quad$ Sample $\boldsymbol{x} \sim \mathcal{D}$, $\boldsymbol{z} \sim \mathcal{N}(\boldsymbol{0}, \boldsymbol{I})$ | $\quad$ Sample $\boldsymbol{x} \sim \mathcal{D}$, $\boldsymbol{z} \sim \mathcal{N}(\boldsymbol{0}, \boldsymbol{I})$ |
| $\quad$ Sample $t$ and $s$ | $\quad$ Sample $t$ and $s$ |
| $\quad$ Sample $r \sim \mathcal{U}[0,1]$, $r \leftarrow t + r(s-t)$ | $\quad$ Sample $r \sim \mathcal{U}[0,1]$, $r \leftarrow t + r(s-t)$ |
| $\quad \boldsymbol{x}_t \leftarrow t\boldsymbol{x} + (1-t)\boldsymbol{z}$ | $\quad \boldsymbol{x}_r \leftarrow r\boldsymbol{x} + (1-r)\boldsymbol{z}$ |
| $\quad \hat{\boldsymbol{x}}_r \leftarrow \boldsymbol{x}_t + (r-t)\boldsymbol{f}_{\theta^-}(\boldsymbol{x}_t, t, r)$ | $\quad \hat{\boldsymbol{x}}_t \leftarrow \boldsymbol{x}_r + (t-r)\boldsymbol{f}_{\theta^-}(\boldsymbol{x}_r, r, t)$ |
| $\quad \boldsymbol{v}_r \leftarrow \boldsymbol{v}(\hat{\boldsymbol{x}}_r, r)$ | $\quad \boldsymbol{u}_r \leftarrow \boldsymbol{x} - \boldsymbol{z}$ |
| $\quad \mathcal{L}(\theta) = \mathbb{E}_{\boldsymbol{x},\boldsymbol{z},t,s,r}\|\boldsymbol{f}_\theta(\boldsymbol{x}_t, t, s) - \boldsymbol{v}_r\|_2^2$ | $\quad \mathcal{L}(\theta) = \mathbb{E}_{\boldsymbol{x},\boldsymbol{z},t,s,r}\|\boldsymbol{f}_\theta(\hat{\boldsymbol{x}}_t, t, s) - \boldsymbol{u}_r\|_2^2$ |
| $\quad \theta \leftarrow \theta - \eta\nabla_\theta\mathcal{L}(\theta)$ | $\quad \theta \leftarrow \theta - \eta\nabla_\theta\mathcal{L}(\theta)$ |
| **until** convergence | **until** convergence |

We define the secant function from $\boldsymbol{x}_t$ at time $t$ to time $s$ as

$$\boldsymbol{f}(\boldsymbol{x}_t, t, s) = \begin{cases} \boldsymbol{v}(\boldsymbol{x}_t, t), & \text{if } t = s, \\ \dfrac{1}{s-t}\displaystyle\int_t^s \boldsymbol{v}(\boldsymbol{x}_r, r)dr, & \text{if } t \neq s. \end{cases} \tag{5}$$

Since

$$\boldsymbol{f}(\boldsymbol{x}_t, t, t) = \lim_{s\to t}\boldsymbol{f}(\boldsymbol{x}_t, t, s) = \lim_{s\to t}\frac{1}{s-t}\int_t^s \boldsymbol{v}(\boldsymbol{x}_r, r)dr = \boldsymbol{v}(\boldsymbol{x}_t, t), \tag{6}$$

$\boldsymbol{f}(\boldsymbol{x}_t, t, s)$ is continuous at $s = t$. And, we parametrize the neural network $\boldsymbol{f}_\theta(\boldsymbol{x}_t, t, s)$ to represent this secant function. If $\boldsymbol{f}_\theta(\boldsymbol{x}_t, t, s)$ is trained accurately fitting $\boldsymbol{f}(\boldsymbol{x}_t, t, s)$, we can directly jump from $\boldsymbol{x}_t$ to $\boldsymbol{x}_s$ using

$$\boldsymbol{x}_s = \boldsymbol{x}_t + \int_t^s \boldsymbol{v}(\boldsymbol{x}_r, r)dr = \boldsymbol{x}_t + (s-t)\boldsymbol{f}_\theta(\boldsymbol{x}_t, t, s). \tag{7}$$

We choose this parameterization for its simplicity and clearer geometric interpretation, though other similar formulations [34, 36, 46] would also be viable in practice.

## 4.2 Secant Expectation

Examining Eq. (5) from a probabilistic perspective, we have

$$\boldsymbol{f}(\boldsymbol{x}_t, t, s) = \frac{1}{s-t}\int_t^s \boldsymbol{v}(\boldsymbol{x}_r, r)dr = \mathbb{E}_{r\sim U(t,s)}\boldsymbol{v}(\boldsymbol{x}_r, r), \tag{8}$$

which indicates that we can calculate the integral $\frac{1}{s-t}\int_t^s \boldsymbol{v}(\boldsymbol{x}_r, r)dr$ by uniformly sampling random values of $r \sim \mathcal{U}(t,s)$[2] and computing their average. However, during training, we cannot obtain a large number of $\boldsymbol{x}_r$'s to calculate this integral accurately. We address this challenge by formulating the integral as the optimal solution to a simple objective that involves only one $\boldsymbol{x}_r$ at a time:

$$\mathcal{L}_{\text{Naïve}}(\theta) = \mathbb{E}_{\boldsymbol{x}_0,\boldsymbol{z},r\sim U(t,s)}\|\boldsymbol{f}_\theta(\boldsymbol{x}_t, t, s) - \boldsymbol{v}(\boldsymbol{x}_r, r)\|_2^2. \tag{9}$$

This approach is inspired by the diffusion objective that leads from Eq. (1) to Eq. (2). Inspecting Eq. (9), we observe that we can only access either $\boldsymbol{x}_t$ or $\boldsymbol{x}_r$ at each training step, but not both simultaneously. To address this issue, we draw inspiration from Picard iteration by estimating one of the solutions between $\boldsymbol{x}_t$ and $\boldsymbol{x}_r$ given the other. We refer to the family of obtained loss functions in this way as *secant losses*. Specifically, one way is to sample $\boldsymbol{x}_t = \alpha_t\boldsymbol{x}_0 + \sigma_t\boldsymbol{z}$, and estimate $\boldsymbol{x}_r$ using

$$\hat{\boldsymbol{x}}_r = \boldsymbol{x}_t + (r-t)\boldsymbol{f}_{\theta^-}(\boldsymbol{x}_t, t, r), \tag{10}$$

---

[2]We do not assume specific order relation between $t$ and $s$, and we sightly abuse the notion $r \sim \mathcal{U}(t,s)$ by meaning $r \sim \mathcal{U}(\min\{t,s\}, \max\{t,s\})$ for simplicity.

where $\theta^-$ denotes the `stop_gradient` version of $\theta$. This transforms the loss function in Eq. (9) into

$$\mathcal{L}_{\mathrm{SDEI}}(\theta) = \mathbb{E}_{\boldsymbol{x}_0,\boldsymbol{z},t,s,r\sim\mathcal{U}(t,s)}\|\boldsymbol{f}_\theta(\boldsymbol{x}_t,t,s) - \boldsymbol{v}(\boldsymbol{x}_t + (r-t)\boldsymbol{f}_{\theta^-}(\boldsymbol{x}_t,t,r),r)\|_2^2, \qquad (11)$$

where $\mathcal{L}_{\mathrm{SDEI}}$ stands for *secant distillation by estimating the interior point*, meaning we sample at the end point $t$ and estimate the interior solution at $r$. According to Eq. (6), we can also directly train the few-step model by incorporating the diffusion loss termed as

$$\mathcal{L}_{\mathrm{STEI}}(\theta) = \mathbb{E}_{\boldsymbol{x}_0,\boldsymbol{z},t,s,r\sim\mathcal{U}(t,s)}\|\boldsymbol{f}_\theta(\boldsymbol{x}_t,t,s) - \boldsymbol{f}_{\theta^-}(\boldsymbol{x}_t + (r-t)\boldsymbol{f}_{\theta^-}(\boldsymbol{x}_t,t,r),r,r)\|_2^2 \\ + \lambda\mathbb{E}_{\boldsymbol{x}_0,\boldsymbol{z},\tau}\|\boldsymbol{f}_\theta(\boldsymbol{x}_\tau,\tau,\tau) - (\alpha'_\tau\boldsymbol{x}_0 + \sigma'_\tau\boldsymbol{z})\|_2^2, \qquad (12)$$

where $\lambda$ is a constant to balance diffusion loss and secant loss, and $\tau$ is a time step indicating the time sampling in the two parts is independent. Here STEI denotes *secant training by estimating the interior point*.

Alternatively, we can sample $\boldsymbol{x}_r = \alpha_r\boldsymbol{x}_0 + \sigma_r\boldsymbol{z}$ and estimate the solution at $t$ from $r$ as

$$\hat{\boldsymbol{x}}_t = \boldsymbol{x}_r + (t-r)\boldsymbol{f}_{\theta^-}(\boldsymbol{x}_r,r,t), \qquad (13)$$

which we term *secant distillation by estimating the end point (SDEE)*. The loss from Eq. (9) then becomes

$$\mathcal{L}_{\mathrm{SDEE}}(\theta) = \mathbb{E}_{\boldsymbol{x}_0,\boldsymbol{z},t,s,r\sim\mathcal{U}(t,s)}\|\boldsymbol{f}_\theta(\boldsymbol{x}_r + (t-r)\boldsymbol{f}_{\theta^-}(\boldsymbol{x}_r,r,t),t,s) - \boldsymbol{v}(\boldsymbol{x}_r,r)\|_2^2. \qquad (14)$$

In the above equation, since we directly use the sampled $\boldsymbol{x}_r$ to evaluate $\boldsymbol{v}(\boldsymbol{x}_r,r)$, we can alternatively use $\alpha'_r\boldsymbol{x}_0 + \sigma'_r\boldsymbol{z}$ to estimate $\boldsymbol{v}(\boldsymbol{x}_r,r)$ like diffusion models do in Eq. (1) to Eq. (2). This leads to the training version termed as *secant training by estimating the end point (STEE)*:

$$\mathcal{L}_{\mathrm{STEE}}(\theta) = \mathbb{E}_{\boldsymbol{x}_0,\boldsymbol{z},t,s,r\sim\mathcal{U}(t,s)}\|\boldsymbol{f}_\theta(\boldsymbol{x}_r + (t-r)\boldsymbol{f}_{\theta^-}(\boldsymbol{x}_r,r,t),t,s) - (\alpha'_r\boldsymbol{x}_0 + \sigma'_r\boldsymbol{z})\|_2^2. \qquad (15)$$

As theoretical justification for the above losses, we present the following theorems, which corresponds to $\mathcal{L}_{\mathrm{SDEI}}(\theta)$ and $\mathcal{L}_{\mathrm{STEE}}(\theta)$. The other two can be derived from these.

**Theorem 2** (SDEI). *Let $\boldsymbol{f}_\theta(\boldsymbol{x}_t,t,s)$ be a neural network, and $\boldsymbol{v}(\boldsymbol{x}_t,t) = \mathbb{E}(\alpha'_t\boldsymbol{x}_0 + \sigma'_t\boldsymbol{z}|\boldsymbol{x}_t)$. Assume $\boldsymbol{v}(\boldsymbol{x}_t,t)$ is L-Lipschitz continuous in its first argument, i.e., $\|\boldsymbol{v}(\boldsymbol{x}_1,t) - \boldsymbol{v}(\boldsymbol{x}_2,t)\|_2 \le L\|\boldsymbol{x}_1 - \boldsymbol{x}_2\|_2$ for all $\boldsymbol{x}_1,\boldsymbol{x}_2 \in \mathbb{R}^n$, $t \in [0,1]$. Then, for each fixed $t$, in a sufficient small neighborhood $|s-t| \le h$ for some $h > 0$, if $\mathcal{L}_{\mathrm{SDEI}}(\theta)$ reaches its minimum, we have $\boldsymbol{f}_\theta(\boldsymbol{x}_t,t,s) = \boldsymbol{f}(\boldsymbol{x}_t,t,s)$.*

**Theorem 3** (STEE). *Let $\boldsymbol{f}_\theta(\boldsymbol{x}_t,t,s)$ be a neural network, and $\boldsymbol{v}(\boldsymbol{x}_t,t) = \mathbb{E}(\alpha'_t\boldsymbol{x}_0 + \sigma'_t\boldsymbol{z}|\boldsymbol{x}_t)$. Assume $\boldsymbol{f}_\theta(\boldsymbol{x}_t,t,s)$ is L-Lipschitz continuous in its first argument, i.e., $\|\boldsymbol{f}_\theta(\boldsymbol{x}_1,t,s) - \boldsymbol{f}_\theta(\boldsymbol{x}_2,t,s)\|_2 \le L\|\boldsymbol{x}_1 - \boldsymbol{x}_2\|_2$ for all $\boldsymbol{x}_1,\boldsymbol{x}_2 \in \mathbb{R}^n$, $t,s \in [0,1]$. Then, for each fixed $[a,b] \subseteq [0,1]$ with $b-a$ sufficiently small, if $\mathcal{L}_{\mathrm{STEE}}(\theta)$ reaches its minimum, we have $\boldsymbol{f}_\theta(\boldsymbol{x}_t,t,s) = \boldsymbol{f}(\boldsymbol{x}_t,t,s)$ for any $[t,s] \subseteq [a,b]$.*

*Remark* 4. There are notable differences between variants that estimate the interior or end point. As illustrated in Fig. 1 (a), the estimation direction of $\hat{\boldsymbol{x}}_r$ aligns with the direction from $\boldsymbol{x}_t$ to $\boldsymbol{x}_s$. Consequently, Theorem 2 states that we can fix $t$ and progressively move $s$ further away. In contrast, for variant (b), the direction between $\hat{\boldsymbol{x}}_t$ estimation and secant learning must be opposite. This requires sampling time pairs where $t$ and $s$ can appear in any order for accurate estimation. As a result, Theorem 3 shows that increasing the distance $|s-t|$ is accompanied by expanding the interval where $t$ and $s$ are randomly sampled rather than simply moving $s$. In short, $t$ and $s$ play symmetric roles in scenario (b), whereas their roles can be asymmetric in scenario (a).

The proofs of the above theorems rely on properties of conditional expectation and techniques related to the Picard-Lindelöf theorem, specifically by constructing a convergent sequence and applying the Banach fixed-point theorem. Detailed proofs of the total four losses are provided in Appendix A. We note that while the proofs only guarantee local accuracy, the expansion to larger time intervals is achieved through a bootstrapping process. We provide illustrations of using $\mathcal{L}_{\mathrm{SDEI}}$ (Eq. (11)) for distillation and $\mathcal{L}_{\mathrm{STEE}}$ (Eq. (15)) for training in Algorithm 1 and Algorithm 2, respectively. And we illustrate the other two in Appendix D.

**Comparison with consistency models.** To achieve the same objective of fitting the average velocity (the secant), consistency models [33] (MeanFlow [67]) uses differentiation to construct its loss, whereas our method uses integration. With identical model parameterization, the two methods can be viewed as differential and integral counterparts. The above fundamental difference leads to the following distinctive features of secant losses: i) *Local accuracy*. Consistency models rely on either difference-based approximations of derivatives or explicit derivative terms in their loss functions, which can lead to training instability. In contrast, our loss functions do not involve explicit derivatives, and the solution is accurate when $s$ is near to $t$.

Table 1: Cost comparison among diffusion loss and secant losses.

| Loss | Teacher | #Forward | #Backward |
|------|---------|----------|-----------|
| $\mathcal{L}_{\text{Diff}}$ | ✗ | 1 | 1 |
| $\mathcal{L}_{\text{SDEI}}$ | ✓ | 3 | 1 |
| $\mathcal{L}_{\text{STEI}}$ | ✗ | 4 | 2 |
| $\mathcal{L}_{\text{SDEE}}$ | ✓ | 3 | 1 |
| $\mathcal{L}_{\text{STEE}}$ | ✗ | 2 | 1 |

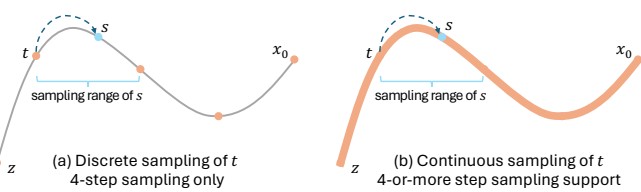

(a) Discrete sampling of $t$
4-step sampling only

(b) Continuous sampling of $t$
4-or-more step sampling support

Figure 4: Discrete vs. continuous sampling of $t$ when estimating the interior point.

ii) *Target stability*. We refer to the stability of the prediction target in the loss function as target stability. Under the flow matching interpolant and model parametrization Eq. (5), we compare diffusion loss Eq. 1, secant losses and consistency loss [33, 34, 67]

$$\mathcal{L}_{\text{CT}}(\theta) = \|\boldsymbol{f}_{\theta}(\boldsymbol{x}_t, t, s) - (\alpha'_t \boldsymbol{x}_0 + \sigma'_t \boldsymbol{z} + (s-t)\frac{d}{dt}\boldsymbol{f}_{\theta^-}(\boldsymbol{x}_t, t, s))\|_2 \tag{16}$$

and

$$\mathcal{L}_{\text{CD}}(\theta) = \|\boldsymbol{f}_{\theta}(\boldsymbol{x}_t, t, s) - (\boldsymbol{v}(\boldsymbol{x}_t, t) + (s-t)\frac{d}{dt}\boldsymbol{f}_{\theta^-}(\boldsymbol{x}_t, t, s))\|_2, \tag{17}$$

where CT and CD stand for consistency training and consistency distillation, respectively. One can see that in consistency losses, the item $\frac{d}{dt}\boldsymbol{f}_{\theta^-}(\boldsymbol{x}_t, t, s)$ (or its discrete version $\frac{\boldsymbol{f}_{\theta^-}(\boldsymbol{x}_t, t, s) - \boldsymbol{f}_{\theta^-}(\boldsymbol{x}_{t-\Delta t}, t, s)}{\Delta t}$) is model-dependent and susceptible to numerical issues, which contributes to training instability. In stark contrast, the target of secant losses is either identical to

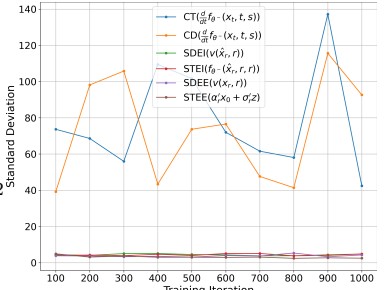

Figure 3: Standard deviation of inspected terms over training iterations. The label format follows *loss type (inspected term)*. The target stability of secant losses is significantly better than that of consistency losses.

the target of diffusion losses ($\alpha'_t \boldsymbol{x}_0 + \sigma'_t \boldsymbol{z}$), or the diffusion model itself ($\boldsymbol{v}(\boldsymbol{x}_t, t)$). We illustrate the standard deviation of different losses across training iterations in Fig. 3. This intrinsic target stability provides a compelling explanation for the robust training performance observed with secant losses.

### 4.3 Practical Choices

Our proposed method is designed for simplicity and robustness, mirroring the design of standard diffusion models. In general it uses: i) A simple time sampling strategy (following diffusion models), ii) a standard loss weighting (following diffusion models), iii) a simple Mean Squared Error (MSE) loss (following diffusion models), iv) a stable loss target discussed in Section 4.2. The strong parallels between our method and standard diffusion models provide a powerful cue: if a standard diffusion model loss works well on a given dataset, it is highly likely our secant losses will too. Here we will discuss more practical designs in application.

**Diffusion initialization.** As explained in Remark 4, we can use a pretrained diffusion model as the initialization such that $\boldsymbol{f}_{\theta}(\boldsymbol{x}_t, t, t) = \boldsymbol{v}(\boldsymbol{x}_t, t)$, which largely accelerates the learning process.

**Application of classifier-free guidance (CFG) [68].** For all the secant losses except $\mathcal{L}_{\text{STEE}}$, we can embed CFG as an additional input into the model [22]. Specifically, in the loss functions we substitute $\boldsymbol{v}(\boldsymbol{x}_t, t)$ with

$$\boldsymbol{v}^g(\boldsymbol{x}_t, t) = \boldsymbol{v}^u(\boldsymbol{x}_t, t) + w(\boldsymbol{v}^c(\boldsymbol{x}_t, t) - \boldsymbol{v}^u(\boldsymbol{x}_t, t)), \tag{18}$$

where $w$ is the guidance scale, and the superscripts $g$, $u$ and $c$ stand for the model with guidance, the unconditional model and the conditional model, respectively. In contrast, the treatment to $\mathcal{L}_{\text{STEE}}$ is similar to training diffusion models. We can train unconditional and conditional models by randomly dropping the class label, and apply CFG at inference via

$$\boldsymbol{f}_{\theta}^g(\boldsymbol{x}_t, t, s) = \boldsymbol{f}_{\theta}^u(\boldsymbol{x}_t, t, s) + w(\boldsymbol{f}_{\theta}^c(\boldsymbol{x}_t, t, s) - \boldsymbol{f}_{\theta}^u(\boldsymbol{x}_t, t, s)). \tag{19}$$

**Step intervals at each sampling step.** Adjusting the step intervals may lead to better performance [6, 18]. While for simplicity, we always use uniform sampling steps. Specifically, if the number of steps

is $N$, then the steps $(t, s)$ are $(\frac{N-i}{N}, \frac{N-i-1}{N})$, $i = 0, ..., N - 1$. We provide details of sampling in Appendix F.

**The sampling of $t$ and $s$ when estimating the interior point.** Theorem 2 states that we can fix $t$ and randomly sample $s$. This enables few-step inference at a fixed step count $N$ by setting $t \in \{1, \frac{N-1}{N}, ..., \frac{i}{N}, ..., \frac{1}{N}, 0\}$ in training. When $t = \frac{i}{N}$, we randomly sample $s \in [\frac{i-1}{N}, \frac{i}{N}]$. However, this approach constrains us to using exactly $N$ steps during inference. Figure 4 (a) illustrates this with an example where $N = 4$. If we attempt to use more NFEs, such as $2N$, we cannot determine the value of $\boldsymbol{f}_\theta(\boldsymbol{x}_{\frac{2N-1}{2N}}, \frac{2N-1}{2N}, \frac{2N-2}{2N})$ at the second step, as the model was not trained with $t = \frac{2N-1}{2N}$. Similarly, with fewer NFEs, such as $\frac{N}{2}$, we cannot compute $\boldsymbol{f}_\theta(\boldsymbol{x}_1, 1, \frac{N-2}{N})$ because the case where $t = 1$ and $s = \frac{N-2}{N}$ was not included in training. To enable a flexible step-accuracy trade-off in $\mathcal{L}_{\text{SDEI}}$, we should continuously sample $t$, as shown in Figure 4 (b). This approach allows inference with any number of NFEs greater than $N$. Furthermore, if the model is required to perform inversion from images back to Gaussian noise, we should incorporate cases of $s < t$ during training. The sampling strategy of $t$ and $s$ can be chosen according to specific application requirements, due to the limited model capacity. However, for variants that estimate the end point, the sampling of $t$ and $s$ must be both continuous and bidirectional.

**The sampling of $r$.** In Eq. (8), we can alter the sampling distribution of $r$ by

$$
\begin{aligned}
\boldsymbol{f}(\boldsymbol{x}_t, t, s) &= \int_t^s \boldsymbol{v}(\boldsymbol{x}_r, r) p_{\mathcal{U}(t,s)}(r) dr \\
&= \int_t^s \boldsymbol{v}(\boldsymbol{x}_r, r) \frac{p_{\mathcal{U}(t,s)}(r)}{q(r)} q(r) dr \\
&= \mathbb{E}_{r \sim q(r)} \boldsymbol{v}(\boldsymbol{x}_r, r) \frac{p_{\mathcal{U}(t,s)}(r)}{q(r)},
\end{aligned}
\tag{20}
$$

where $q(r)$ is another probability density function that has positive density value within the limits of integration. This means we can unevenly sample $r$ based on the importance, *i.e.*, how we allocate the training effort across different time intervals. We note that while importance sampling can reduce variance, it is not the focus of our paper. The method used in this paper to reduce variance is to employ a stable target.

**Resource costs.** As the loss formulations suggest, compared to the diffusion loss $\mathcal{L}_{\text{Diff}}$, our proposed loss functions require additional computational resources for forward evaluation and/or backward propagation. We summarize the computational costs in Table 1, and test the practical usage in Appendix G.2.

## 5 Experiments

In this section, we first compare our method with other related approaches. Subsequently, we conduct ablation studies to validate the design choices of the proposed loss functions.

### 5.1 Implementation Details

We conduct experiments on common used image generation datasets including CIFAR-10 [37] and ImageNet-256 $\times$ 256 [38]. For quantitative evaluation, we employ the Fréchet Inception Distance (FID) [87] score for both datasets, with additional Inception Score (IS) metric for ImageNet-256$\times$256. There are UNet-based EDM [6] and transformer-based DiT [7] models involved in the experiments, and we load the pretrained weights of EDM and SiT [39] for the two models respectively. More details can be found in Appendix H.

### 5.2 Main Results

**Overall Comparison.** We conduct a comprehensive comparison among various few-step diffusion approaches, including fast samplers, few-step distillation, and few-step training/fine-tuning methods. The results are summarized in Tables 2 and 3. On CIFAR-10, our SDEI variant achieves intermediate performance between faster samplers and few-step distillation/training methods. Although

Table 2: Unconditional image generation on CIFAR-10.

| Method | FID↓ | Steps↓ |
|---|---|---|
| **Diffusion** | | |
| DDPM [2] | 3.17 | 1000 |
| Score SDE (deep) [4] | 2.20 | 2000 |
| EDM [6] **(Teacher)** | 1.97 | 35 |
| Flow Matching [65] | 6.35 | 142 |
| Rectified Flow [47] | 2.58 | 127 |
| **Fast Samplers** | | |
| DPM-Solver [15] | 4.70 | 10 |
| DPM-Solver++ [16] | 2.91 | 10 |
| DPM-Solver-v3 [17] | 2.51 | 10 |
| DEIS [54] | 4.17 | 10 |
| UniPC [55] | 3.87 | 10 |
| LD3 [64] | 2.38 | 10 |
| **Joint Training** | | |
| Diff-Instruct [29] | 4.53 | 1 |
| DMD [27] | 3.77 | 1 |
| CTM [34] | 1.87 | 2 |
| SiD [43] | 1.92 | 1 |
| SiD$^2$A [69] | 1.5 | 1 |
| SiM [70] | 2.06 | 1 |
| **Few-step Distillation** | | |
| KD [19] | 9.36 | 1 |
| PD [21] | 4.51 | 2 |
| DFNO [20] | 3.78 | 1 |
| 2-Rectified Flow [47] | 4.85 | 1 |
| TRACT [71] | 3.32 | 2 |
| PID [72] | 3.92 | 1 |
| CD [33] | 2.93 | 2 |
| sCD [35] | 2.52 | 2 |
| **Few-step Training/Tuning** | | |
| iCT [73] | 2.46 | 2 |
| ECT [45] | 2.11 | 2 |
| sCT [35] | 2.06 | 2 |
| IMM [74] | 1.98 | 2 |
| **SDEI** (Ours) | 3.23 | 4 |
| | 2.14 | 10 |

Table 3: Class-conditional results on ImageNet-256 × 256.

| Method | FID↓ | IS↑ | Steps↓ | #Params |
|---|---|---|---|---|
| **Diffusion** | | | | |
| ADM [5] | 10.94 | 100.98 | 250 | 554M |
| CDM [75] | 4.88 | 158.71 | 8100 | - |
| SimDiff [76] | 2.77 | 211.8 | 512 | 2B |
| LDM-4 [8] | 3.60 | 247.67 | 250 | 400M |
| U-DiT-L [77] | 3.37 | 246.03 | 250 | 916M |
| U-ViT-H [78] | 2.29 | 263.88 | 50 | 501M |
| DiT-XL/2 [7] | 2.27 | 278.24 | 250 | 675M |
| SiT-XL/2 [39] **(Teacher)** | 2.15 | 258.09 | 250 | 675M |
| **Fast Samplers** | | | | |
| Flow-DPM-Solver [79, 16] | 3.76 | 241.18 | 8 | 675M |
| UniPC [55] | 7.51 | - | 10 | - |
| LD3 [64] | 4.32 | - | 7 | - |
| **GAN** | | | | |
| BigGAN [80] | 6.95 | 171.4 | 1 | 112M |
| GigaGAN [81] | 3.45 | 225.52 | 1 | 590M |
| StyleGAN-XL [82] | 2.30 | 265.12 | 1 | 166M |
| **Masked and AR** | | | | |
| VQGAN [83] | 15.78 | 78.3 | 1024 | 227M |
| MaskGIT [84] | 6.18 | 182.1 | 8 | 227M |
| MAR-H [85] | 1.55 | 303.7 | 256 | 943M |
| VAR-d30 [86] | 1.97 | 334.7 | 10 | 2B |
| **Few-step Training/Tuning** | | | | |
| iCT [73] | 20.3 | - | 2 | 675M |
| Shortcut Models [46] | 7.80 | - | 4 | 675M |
| IMM (XL/2) [74] | 7.77 | - | 1 | 675M |
| | 3.99 | - | 2 | 675M |
| | 2.51 | - | 4 | 675M |
| | 1.99 | - | 8 | 675M |
| **STEI** (Ours) | 7.12 | 241.75 | 1 | 675M |
| | 4.41 | 242.00 | 2 | 675M |
| | 2.78 | 269.87 | 4 | 675M |
| +guidance interval [40] | 2.27 | 273.76 | 4 | 675M |
| | 2.36 | 247.72 | 8 | 675M |
| +guidance interval [40] | 1.96 | 276.12 | 8 | 675M |
| **STEE** (Ours) | 3.02 | 274.00 | 4 | 675M |
| +guidance interval [40] | 2.55 | 275.83 | 4 | 675M |
| | 2.33 | 274.47 | 8 | 675M |
| +guidance interval [40] | 1.96 | 275.81 | 8 | 675M |

consistency models [33, 45, 34, 35] demonstrate superior performance on this dataset, we observe training instability issues. More notably, when scaling to ImageNet-256 × 256 with greater data variation, these models diverge in our experiments[3]. In contrast, our proposed loss functions exhibit consistent stability and rapid convergence, enabling efficient model distillation or fine-tuning that requires merely $1\%$ of the original SiT training duration (50K $\sim$ 100K versus 7M iterations). On the larger-scale ImageNet-256 × 256 dataset, our approach consistently outperforms fast samplers and ranks second only to the recent IMM [74] in both 4-step and 8-step settings. Notably, for one-step generation, STEI even surpasses both IMM and 4-step Shortcut Models [46]. By incorporating interval CFG [40], our method further achieves a 4-step FID of 2.27 and an 8-step FID of 1.96, where we simply ignore the guidance for the first step in 4-step generation and the first two steps in 8-step generation. Fig. 2 showcases generated samples using 8 steps on ImageNet-256 × 256 with our best-performing model. While both effective, our method differs from IMM: ours is grounded in the

---

[3]Due to training divergence of consistency models, we cite the iCT [73] result reported in [74].

Table 4: The effect of the weighting factor $\lambda$ in $\mathcal{L}_{\text{STEI}}$ on ImageNet-256 × 256 with 4 sampling steps.

| $\lambda$ | FID↓ | IS↑ |
|-----------|------|-----|
| 0.1 | 3.96 | 206.14 |
| 0.5 | 3.15 | 232.15 |
| 1.0 | 2.84 | 249.57 |
| 2.0 | 3.96 | 213.56 |

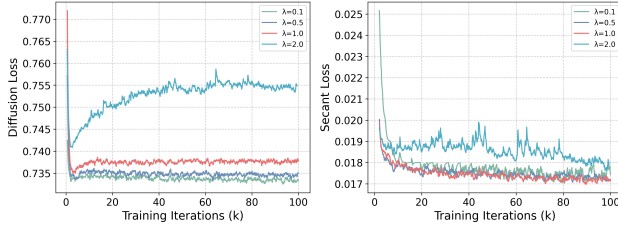

Figure 5: The effect of $\lambda$ in $\mathcal{L}_{\text{STEI}}$ on training dynamics.

exact solution of the PF-ODE (therefore the performance of our models is bounded by the teacher model SiT[39]), whereas IMM relies on distribution-level losses [88]. For classifier-free guidance, IMM follows Eq. (19), while STEI applies the integral-form CFG following Eq. (18). This difference may explain our superior performance in one-step generation.

**Comparison of secant losses.** We evaluate the four types of secant losses proposed in Section 4 on CIFAR-10 and ImageNet-256 × 256 datasets. The result can be found at Table 7 in Appendix G. In the absence of classifier-free guidance, similar trends emerge across both datasets. The distillation variants generally demonstrate superior performance compared to the training counterparts, with the exception of STEI on ImageNet-256 × 256. Furthermore, variants with interior-point estimation perform better and degrade more slowly as the step number decreases, which can be explained by three aspects: i) Input correctness. For inner-point variants, the input to the training network is the clean data $x_t$, while in end-point variants, the input is the estimated $\hat{x}_r$. Fitting a fixed, clean input distribution, is more efficient than fitting an input dependent of the model historical output. ii) Error accumulation path, *i.e.*, the path from clean data to the prediction destination. In inner-point variants, the path is $x_t \rightarrow x_s$, while in end-point variants it is $x_r \rightarrow x_t \rightarrow x_s$. Generally, the path in end-point variants is longer than that in inner-point variants. iii) Model capacity. As per Fig. 1 and Algorithm 2, the end-point variants additionally require inversion, which costs part of the model capacity. With classifier guidance applied, the performance gap narrows on ImageNet-256 × 256, with the STEE variant achieving the best 8-step FID of 2.33. Based on the computational costs presented in Table 1, we recommend using $\mathcal{L}_{\text{SDEI}}$ for distillation and $\mathcal{L}_{\text{STEE}}$ for training. These variants offer optimal performance while minimizing both computational time and GPU memory requirements compared to other methods.

## 5.3 Ablations

**The loss weighting in $\mathcal{L}_{\textbf{STEI}}$.** We investigate the balancing factor $\lambda$ in $\mathcal{L}_{\text{STEI}}$, which comprises the diffusion loss and the secant loss. Figure 5 illustrates the training dynamics of both losses (excluding the $\lambda$ factor in the secant loss). When $\lambda$ increases from 0.1 to 1.0, we observe a reduction in the secant loss while maintaining relatively stable diffusion loss levels. However, at $\lambda = 2$, the model exhibits significant deterioration in diffusion loss, accompanied by increased instability in the secant loss. The quantitative results presented in Table 4 confirm that $\lambda = 1$ achieves the optimal balance, yielding the best overall performance.

**The sampling of $t$ and $s$ in interior-point estimation.** As shown in Fig. 4, $\mathcal{L}_{\text{SDEI}}$ and $\mathcal{L}_{\text{STEI}}$ offer flexibility in sampling the time point $t$ and $s$ during training. Given the fixed model capacity, different sampling strategies lead to varying performance characteristics. For fixed $N$-step generation, discrete sampling suffices; to enable step-performance trade-off, continuous sampling of $t$ becomes necessary; to incorporate the capability of inversion, bidirectional sampling ($t < s$ and $t > s$) is required. The results in Table 5 demonstrate that the best performance is achieved with the discrete sampling and generation-only configuration (first row), and each additional functionality comes at the cost of decreased performance.

Table 5: The effect of $t, s$ sampling on CIFAR-10 with 4 steps.

| Gen. | Inv. | $t$ sampling | FID↓ |
|------|------|--------------|------|
| ✓ | ✗ | discrete | 3.23 |
| ✓ | ✓ | discrete | 3.63 |
| ✓ | ✗ | continuous | 4.29 |
| ✓ | ✓ | continuous | 5.47 |

**The sampling of $r$.** Following Eq. (20), we know that we can use alternative distributions $q(r)$ to sample $r$. Here we investigate sampling strategies for $r$. The cases include standard uniform sampling and biased sampling schemes that favor positions closer to $t$, $s$ or the midpoint $\frac{s+t}{2}$. These biased sampling strategies are implemented using truncated normal distributions, denoted as $\mathcal{TN}(\mu, \sigma, a, b)$ where $\mu$ represents the mean, $\sigma$ the standard deviation, and $[a, b]$ the truncated interval. Details are provided in Appendix E. As the results in Table 6 indicates, all sampling strategies works comparable, and the uniform strategy works slightly better than others.

Table 6: 4-step results of different $r$ distributions on ImageNet-256 × 256 with $\mathcal{L}_{\text{SDEI}}$. We first sample $\tilde{r}$, and obtain $r = t + (s - t)\tilde{r}$.

| $\tilde{r}$ distribution | FID↓ | IS↑ |
|---|---|---|
| $\mathcal{U}(0,1)$ | 3.57 | 222.83 |
| $\mathcal{TN}(0, 0.5, 0, 1)$ | 3.84 | 215.57 |
| $\mathcal{TN}(0.5, 0.5, 0, 1)$ | 3.59 | 221.44 |
| $\mathcal{TN}(1, 0.5, 0, 1)$ | 3.71 | 216.93 |

**Training from scratch.** We evaluate the scalability and from-scratch training capabilities of the secant loss using $\mathcal{L}_{\text{STEE}}$ as the representative loss function. We conduct experiments with varying model capacities, including DiT-S/2, DiT-B/2, DiT-L/2 and DiT-XL/2. Figure 6 presents the evolution of FID scores during the first 1000K training iterations, with a guidance scale of 1.5 used at sampling phase. The results clearly show a correlation between model capacity and generation quality, with larger architectures consistently achieving superior performance. After extending the training to 3000K iterations, our DiT-XL/2 model achieves an 8-step FID of 2.68, which is slightly higher than the fine-tuning FID. This is expected, given that the training period of the teacher model SiT [39] (7000K iterations) is significantly longer than our training duration.

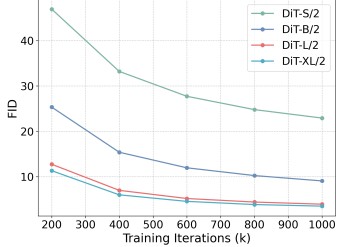

Figure 6: Results of training from scratch on ImageNet-256× 256. The guidance scale is 1.5.

# 6 Limitations

While our method can stably and rapidly convert a diffusion model into a few-step generator, there is still a significant performance gap between 1-step and 8-step generation. Additionally, the performance on ImageNet 256 × 256 relies heavily on classifier-free guidance, and the theoretical relationship between CFG and secant losses may be explored in future work. Lastly, our method requires training data, which may present constraints in data-limited scenarios.

# 7 Conclusion

In this paper, we introduce a novel family of loss functions, termed secant losses, that efficiently learn to integrate diffusion ODEs through Monte Carlo integration and Picard iteration. Our proposed losses operate by estimating and averaging tangents to learn the corresponding secants. We present both theoretical and intuitive interpretations of these secant losses, and empirically demonstrate their robustness and efficiency on CIFAR-10 and ImageNet 256 × 256 datasets.

## Acknowledgments

This work is partially supported by the National Natural Science Foundation of China (Grant No. 62306261), and The Shun Hing Institute of Advanced Engineering (SHIAE) Grant (No. 8115074). This study was supported in part by the Centre for Perceptual and Interactive Intelligence, a CUHK-led InnoCentre under the InnoHK initiative of the Innovation and Technology Commission of the Hong Kong Special Administrative Region Government. This work is also partially supported by Hong Kong RGC Strategic Topics Grant STG1/E-403/24-N, and CUHK-CUHK(SZ)-GDST Joint Collaboration Fund YSP26-4760949.

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

# Appendices

## A Proofs

**Proposition 1.** *When*

$$\mathcal{L}_{Diff}(\theta) = \mathbb{E}_{\boldsymbol{x}_0,\boldsymbol{z},t}\|\boldsymbol{v}_\theta(\boldsymbol{x}_t,t) - (\alpha'_t\boldsymbol{x}_0 + \sigma'_t\boldsymbol{z})\|_2^2$$

*reaches its minimum, the optimal solution $\boldsymbol{v}_\theta^*(\boldsymbol{x}_t,t)$ is*

$$\boldsymbol{v}(\boldsymbol{x}_t,t) = \mathbb{E}_{\boldsymbol{x}_0,\boldsymbol{z}}(\alpha'_t\boldsymbol{x}_0 + \sigma'_t\boldsymbol{z}|\boldsymbol{x}_t).$$

*Proof.* Since

$$
\begin{aligned}
\mathcal{L}_{\mathrm{Diff}}(\theta) &= \mathbb{E}_{\boldsymbol{x}_0,\boldsymbol{z},t}\|\boldsymbol{v}_\theta(\boldsymbol{x}_t,t) - (\alpha'_t\boldsymbol{x}_0 + \sigma'_t\boldsymbol{z})\|_2^2 \\
&= \mathbb{E}_{\boldsymbol{x}_t,t}\mathbb{E}_{\boldsymbol{x}_0,\boldsymbol{z}}[\|\boldsymbol{v}_\theta(\boldsymbol{x}_t,t) - (\alpha'_t\boldsymbol{x}_0 + \sigma'_t\boldsymbol{z})\|_2^2|\boldsymbol{x}_t] \\
&= \mathbb{E}_{\boldsymbol{x}_t,t}[\mathbb{E}_{\boldsymbol{x}_0,\boldsymbol{z}}[\|\boldsymbol{v}_\theta(\boldsymbol{x}_t,t)\|_2^2|\boldsymbol{x}_t] - 2\mathbb{E}_{\boldsymbol{x}_0,\boldsymbol{z}}[\langle\boldsymbol{v}_\theta(\boldsymbol{x}_t,t), \alpha'_t\boldsymbol{x}_0 + \sigma'_t\boldsymbol{z}\rangle|\boldsymbol{x}_t] \\
&\quad + \mathbb{E}_{\boldsymbol{x}_0,\boldsymbol{z}}[\|\alpha'_t\boldsymbol{x}_0 + \sigma'_t\boldsymbol{z}\|_2^2|\boldsymbol{x}_t]] \\
&= \mathbb{E}_{\boldsymbol{x}_t,t}\left(\|\boldsymbol{v}_\theta(\boldsymbol{x}_t,t)\|_2^2 - 2\mathbb{E}_{\boldsymbol{x}_0,\boldsymbol{z}}[\langle\boldsymbol{v}_\theta(\boldsymbol{x}_t,t), \alpha'_t\boldsymbol{x}_0 + \sigma'_t\boldsymbol{z}\rangle|\boldsymbol{x}_t]\right. \\
&\quad \left. + \mathbb{E}_{\boldsymbol{x}_0,\boldsymbol{z}}[\|\alpha'_t\boldsymbol{x}_0 + \sigma'_t\boldsymbol{z}\|_2^2|\boldsymbol{x}_t]\right) \\
&= \mathbb{E}_{\boldsymbol{x}_t,t}\left(\|\boldsymbol{v}_\theta(\boldsymbol{x}_t,t)\|_2^2 - 2\langle\boldsymbol{v}_\theta(\boldsymbol{x}_t,t), \mathbb{E}_{\boldsymbol{x}_0,\boldsymbol{z}}[\alpha'_t\boldsymbol{x}_0 + \sigma'_t\boldsymbol{z}|\boldsymbol{x}_t]\rangle\right. \\
&\quad \left. + \mathbb{E}_{\boldsymbol{x}_0,\boldsymbol{z}}[\|\alpha'_t\boldsymbol{x}_0 + \sigma'_t\boldsymbol{z}\|_2^2|\boldsymbol{x}_t]\right) \\
&= \mathbb{E}_{\boldsymbol{x}_t,t}\left(\|\boldsymbol{v}_\theta(\boldsymbol{x}_t,t)\|_2^2 - 2\langle\boldsymbol{v}_\theta(\boldsymbol{x}_t,t), \boldsymbol{v}(\boldsymbol{x}_t,t)\rangle + \|\boldsymbol{v}(\boldsymbol{x}_t,t)\|_2^2\right. \\
&\quad \left. - \|\boldsymbol{v}(\boldsymbol{x}_t,t)\|_2^2 + \mathbb{E}_{\boldsymbol{x}_0,\boldsymbol{z}}[\|\alpha'_t\boldsymbol{x}_0 + \sigma'_t\boldsymbol{z}\|_2^2|\boldsymbol{x}_t]\right) \\
&= \mathbb{E}_{\boldsymbol{x}_t,t}\left(\|\boldsymbol{v}_\theta(\boldsymbol{x}_t,t) - \boldsymbol{v}(\boldsymbol{x}_t,t)\|_2^2 - \|\boldsymbol{v}(\boldsymbol{x}_t,t)\|_2^2 + \mathbb{E}_{\boldsymbol{x}_0,\boldsymbol{z}}[\|\alpha'_t\boldsymbol{x}_0 + \sigma'_t\boldsymbol{z}\|_2^2|\boldsymbol{x}_t]\right),
\end{aligned}
$$

we have the optimal solution $\boldsymbol{v}_\theta^*(\boldsymbol{x}_t,t) = \boldsymbol{v}(\boldsymbol{x}_t,t)$. $\qquad\square$

The above is a basic result in literature of diffusion models [66]. We prove it here since it has close relation to the following theorems.

**Theorem 2** (SDEI). *Let $\boldsymbol{f}_\theta(\boldsymbol{x}_t,t,s)$ be a neural network, and $\boldsymbol{v}(\boldsymbol{x}_t,t) = \mathbb{E}(\alpha'_t\boldsymbol{x}_0 + \sigma'_t\boldsymbol{z}|\boldsymbol{x}_t)$. Assume $\boldsymbol{v}(\boldsymbol{x}_t,t)$ is L-Lipschitz continuous in its first argument, i.e., $\|\boldsymbol{v}(\boldsymbol{x}_1,t) - \boldsymbol{v}(\boldsymbol{x}_2,t)\|_2 \le L\|\boldsymbol{x}_1 - \boldsymbol{x}_2\|_2$ for all $\boldsymbol{x}_1, \boldsymbol{x}_2 \in \mathbb{R}^n$, $t \in [0,1]$. Then, for each fixed $t$, in a sufficient small neighborhood $|s-t| \le h$ for some $h > 0$, if $\mathcal{L}_{SDEI}(\theta)$ reaches its minimum, we have $\boldsymbol{f}_\theta(\boldsymbol{x}_t,t,s) = \boldsymbol{f}(\boldsymbol{x}_t,t,s)$.*

*Proof.* We prove it in two steps.

First, we show that the following loss function

$$\mathcal{L}_\star(\theta) := \mathbb{E}_{\boldsymbol{x}_0,\boldsymbol{x}_1,s,t}\|\boldsymbol{f}_\theta(\boldsymbol{x}_t,t,s) - \frac{1}{s-t}\int_t^s \boldsymbol{v}(\boldsymbol{x}_t + (r-t)\boldsymbol{f}_{\theta^-}(\boldsymbol{x}_t,t,r), r)dr\|_2^2 \qquad (21)$$

has the same derivative with respect to $\theta$ as $\mathcal{L}_{\mathrm{SDEI}}(\theta)$.

Let $\hat{\boldsymbol{x}}_r = \boldsymbol{x}_t + (r-t)\boldsymbol{f}_{\theta^-}(\boldsymbol{x}_t,t,r)$, it follows since

$$
\begin{aligned}
\nabla_\theta \mathcal{L}_{\mathrm{SDEI}}(\theta) &= \nabla_\theta \mathbb{E}_{\boldsymbol{x}_0,\boldsymbol{z},t,s}\int_t^s \|\boldsymbol{f}_\theta(\boldsymbol{x}_t,t,s) - \boldsymbol{v}(\hat{\boldsymbol{x}}_r,r)\|_2^2 dr \\
&= \nabla_\theta \mathbb{E}_{\boldsymbol{x}_0,\boldsymbol{z},t,s}\int_t^s (\|\boldsymbol{f}_\theta(\boldsymbol{x}_t,t,s)\|_2^2 - 2\langle\boldsymbol{f}_\theta(\boldsymbol{x}_t,t,s), \boldsymbol{v}(\hat{\boldsymbol{x}}_r,r)\rangle + \|\boldsymbol{v}(\hat{\boldsymbol{x}}_r,r)\|_2^2)dr \\
&= \nabla_\theta \mathbb{E}_{\boldsymbol{x}_0,\boldsymbol{z},t,s}\left((s-t)\|\boldsymbol{f}_\theta(\boldsymbol{x}_t,t,s)\|_2^2 - 2\langle\boldsymbol{f}(\boldsymbol{x}_t,t,s), \int_t^s \boldsymbol{v}(\hat{\boldsymbol{x}}_r,r)dr\rangle\right. \\
&\quad \left. + \int_t^s \|\boldsymbol{v}(\hat{\boldsymbol{x}}_r,r)\|_2^2 dr\right) \\
&= \nabla_\theta \mathbb{E}_{\boldsymbol{x}_0,\boldsymbol{z},t,s}(s-t)\|\boldsymbol{f}_\theta(\boldsymbol{x}_t,t,s) - \frac{1}{s-t}\int_t^s \boldsymbol{v}(\hat{\boldsymbol{x}}_r,r)dr\|_2^2.
\end{aligned}
$$

Second, we show that $\mathcal{L}_\star(\theta) = \mathbf{0}$ implies $\boldsymbol{f}_\theta(\boldsymbol{x}_t, t, s) = \boldsymbol{f}(\boldsymbol{x}_t, t, s)$ in each sufficient small neighborhood of $t$. Using Picard iteration, we construct a sequence $\boldsymbol{f}_n$ satisfying $\boldsymbol{f}_0 = 0, \boldsymbol{f}_{n+1} = \frac{1}{s-t} \int_t^s \boldsymbol{v}(\boldsymbol{x}_t + (r-t)\boldsymbol{f}_n(\boldsymbol{x}_t, t, r), r)dr$. Without loss of generality, we may assume $h > 0$. We have

$$\sup_{s \in (t, t+h]} \|\boldsymbol{f}_{n+1}(\boldsymbol{x}_t, t, s) - \boldsymbol{f}(\boldsymbol{x}_t, t, s)\|_2^2$$

$$= \sup_{s \in (t, t+h]} \frac{1}{(s-t)^2} \| \int_t^s \boldsymbol{v}(\boldsymbol{x}_t + (r-t)\boldsymbol{f}_n(\boldsymbol{x}_t, t, r), r)dr$$

$$- \int_t^s \boldsymbol{v}(\boldsymbol{x}_t + (r-t)\boldsymbol{f}(\boldsymbol{x}_t, t, r), r)dr\|_2^2$$

$$\leq L \sup_{s \in (t, t+h]} \frac{1}{(s-t)^2} \| \int_t^s (r-t)(\boldsymbol{f}_n(\boldsymbol{x}_t, t, r) - \boldsymbol{f}(\boldsymbol{x}_t, t, r))dr\|_2^2$$

$$\leq L \sup_{s \in (t, t+h]} \frac{1}{(s-t)^2} \int_t^s (r-t)^2 dr \sup_{r \in [t, t+h]} \|\boldsymbol{f}_n(\boldsymbol{x}_t, t, r) - \boldsymbol{f}(\boldsymbol{x}_t, t, r)\|_2^2$$

$$= \frac{1}{3} Lh \sup_{s \in (t, t+h]} \|\boldsymbol{f}_n(\boldsymbol{x}_t, t, s) - \boldsymbol{f}(\boldsymbol{x}_t, t, s)\|_2^2.$$

When $s = t$, we also have

$$\|\boldsymbol{f}_{n+1}(\boldsymbol{x}_t, t, s) - \boldsymbol{f}(\boldsymbol{x}_t, t, s)\|_2^2 = \frac{1}{3} Lh \|\boldsymbol{f}_n(\boldsymbol{x}_t, t, s) - \boldsymbol{f}(\boldsymbol{x}_t, t, s)\|_2^2.$$

Therefore, $\displaystyle\sup_{s \in [t, t+h]} \|\boldsymbol{f}_{n+1}(\boldsymbol{x}_t, t, s) - \boldsymbol{f}(\boldsymbol{x}_t, t, s)\|_2^2$ converges to 0 when $\frac{1}{3} Lh < 1$, i.e., $h < \frac{3}{L}$. This means we find a neighborhood $[t, t+h]$ for each $t$, such that when $\mathcal{L}_{\text{SDEI}}(\theta)$ reaches its minimum, we have $\boldsymbol{f}_\theta(\boldsymbol{x}_t, t, s) = \boldsymbol{f}(\boldsymbol{x}_t, t, s)$. □

**Theorem 3** (STEE). *Let $\boldsymbol{f}_\theta(\boldsymbol{x}_t, t, s)$ be a neural network, and $\boldsymbol{v}(\boldsymbol{x}_t, t) = \mathbb{E}(\alpha_t' \boldsymbol{x}_0 + \sigma_t' \boldsymbol{z} | \boldsymbol{x}_t)$. Assume $\boldsymbol{f}_\theta(\boldsymbol{x}_t, t, s)$ is $L$-Lipschitz continuous in its first argument, i.e., $\|\boldsymbol{f}_\theta(\boldsymbol{x}_1, t, s) - \boldsymbol{f}_\theta(\boldsymbol{x}_2, t, s)\|_2 \leq L\|\boldsymbol{x}_1 - \boldsymbol{x}_2\|_2$ for all $\boldsymbol{x}_1, \boldsymbol{x}_2 \in \mathbb{R}^n$, $t, s \in [0, 1]$. Then, for each fixed $[a, b] \subseteq [0, 1]$ with $b - a$ sufficiently small, if $\mathcal{L}_{STEE}(\theta)$ reaches its minimum, we have $\boldsymbol{f}_\theta(\boldsymbol{x}_t, t, s) = \boldsymbol{f}(\boldsymbol{x}_t, t, s)$ for any $[t, s] \subseteq [a, b]$.*

*Proof.* First, we show that $\mathcal{L}_{\text{STEE}}(\theta)$ have the same derivative with respect to $\theta$ as the following loss function

$$\mathcal{L}_\star(\theta) := \mathbb{E}_{\boldsymbol{x}_r, s, t, r \sim \mathcal{U}(s, t)} \|\boldsymbol{f}_\theta(\boldsymbol{x}_r + (t - r)\boldsymbol{f}_{\theta^-}(\boldsymbol{x}_r, r, t), t, s) - \boldsymbol{v}(\boldsymbol{x}_r, r)\|_2^2. \tag{22}$$

It follows since

$$\nabla_\theta \mathcal{L}_{\text{STEE}}(\theta) = \nabla_\theta \mathbb{E}_{\boldsymbol{x}_0, \boldsymbol{z}, t, s} \int_t^s \|\boldsymbol{f}_\theta(\boldsymbol{x}_r + (t - r)\boldsymbol{f}_{\theta^-}(\boldsymbol{x}_r, r, t), t, s) - (\alpha_r' \boldsymbol{x}_0 + \sigma_r' \boldsymbol{z})\|_2^2 dr$$

$$= \nabla_\theta \mathbb{E}_{\boldsymbol{x}_r, t, s} \mathbb{E}_{\boldsymbol{x}_0, \boldsymbol{z}} \int_t^s [\|\boldsymbol{f}_\theta(\boldsymbol{x}_r + (t - r)\boldsymbol{f}_{\theta^-}(\boldsymbol{x}_r, r, t), t, s) - \alpha_r' \boldsymbol{x}_0 + \sigma_r' \boldsymbol{z}\|_2^2 | \boldsymbol{x}_r] dr$$

$$= \nabla_\theta \mathbb{E}_{\boldsymbol{x}_r, t, s} \int_t^s (\mathbb{E}_{\boldsymbol{x}_0, \boldsymbol{z}} [\|\boldsymbol{f}_\theta(\boldsymbol{x}_r + (t - r)\boldsymbol{f}_{\theta^-}(\boldsymbol{x}_r, r, t), t, s)\|_2^2 | \boldsymbol{x}_r]$$

$$- 2\mathbb{E}_{\boldsymbol{x}_0, \boldsymbol{z}} [\langle \boldsymbol{f}_\theta(\boldsymbol{x}_r + (t - r)\boldsymbol{f}_{\theta^-}(\boldsymbol{x}_r, r, t), t, s), \alpha_r' \boldsymbol{x}_0 + \sigma_r' \boldsymbol{z} \rangle | \boldsymbol{x}_r]$$

$$+ \mathbb{E}_{\boldsymbol{x}_0, \boldsymbol{z}} [\|\alpha_r' \boldsymbol{x}_0 + \sigma_r' \boldsymbol{z}\|_2^2 | \boldsymbol{x}_r]) dr$$

$$= \nabla_\theta \mathbb{E}_{\boldsymbol{x}_r, t, s} \int_t^s \|\boldsymbol{f}_\theta(\boldsymbol{x}_r + (t - r)\boldsymbol{f}_{\theta^-}(\boldsymbol{x}_r, r, t), t, s)\|_2^2 dr$$

$$- 2 \int_t^s \langle \boldsymbol{f}_\theta(\boldsymbol{x}_r + (t - r)\boldsymbol{f}_{\theta^-}(\boldsymbol{x}_r, r, t), t, s), \mathbb{E}_{\boldsymbol{x}_0, \boldsymbol{z}} [\alpha_r' \boldsymbol{x}_0 + \sigma_r' \boldsymbol{z} | \boldsymbol{x}_r] \rangle dr$$

$$+ \int_t^s \mathbb{E}_{\boldsymbol{x}_0, \boldsymbol{z}} [\|\alpha_r' \boldsymbol{x}_0 + \sigma_r' \boldsymbol{z}\|_2^2 | \boldsymbol{x}_r] dr$$

$$= \nabla_\theta \mathbb{E}_{\boldsymbol{x}_r,t,s} \int_t^s \|\boldsymbol{f}_\theta(\boldsymbol{x}_r + (t-r)\boldsymbol{f}_{\theta^-}(\boldsymbol{x}_r,r,t),t,s) - \boldsymbol{v}(\boldsymbol{x}_r,r)\|_2^2 dr.$$

Then, notice that when $\mathcal{L}_\star(\theta)$ achieves its minimum, we have

$$\mathcal{L}_{\star\star}(\theta) := \|\int_t^s \boldsymbol{f}_\theta(\boldsymbol{x}_r + (t-r)\boldsymbol{f}_{\theta^-}(\boldsymbol{x}_r,r,t),t,s)dr - \int_t^s \boldsymbol{v}(\boldsymbol{x}_r,r)dr\|_2^2 = 0. \qquad (23)$$

If not, denote $\boldsymbol{g}_\theta(\boldsymbol{x}_r,r,t,s) = \boldsymbol{f}_\theta(\boldsymbol{x}_r + (t-r)\boldsymbol{f}_{\theta^-}(\boldsymbol{x}_r,r,t),t,s)$, and assume $\mathcal{L}_\star(\theta)$ reaches the minimum without $\int_t^s \boldsymbol{g}_\theta(\boldsymbol{x}_r,r,t,s)dr = \int_t^s \boldsymbol{v}(\boldsymbol{x}_r,r)dr$.

Let $\tilde{\boldsymbol{g}}_\theta(\boldsymbol{x}_r,r,t,s) = \boldsymbol{g}_\theta(\boldsymbol{x}_r,r,t,s) + \frac{1}{s-t}\int_t^s \boldsymbol{v}(\boldsymbol{x}_r,r)dr - \frac{1}{s-t}\int_t^s \boldsymbol{g}_\theta(\boldsymbol{x}_r,r,t,s)dr$, then we have

$$\int_t^s \|\tilde{\boldsymbol{g}}_\theta(\boldsymbol{x}_r,r,t,s) - \boldsymbol{v}(\boldsymbol{x}_r,r)\|_2^2 dr$$

$$= \int_t^s \|\boldsymbol{g}_\theta(\boldsymbol{x}_r,r,t,s) - \boldsymbol{v}(\boldsymbol{x}_r,r) + \frac{1}{s-t}\int_t^s \boldsymbol{v}(\boldsymbol{x}_r,r)dr - \frac{1}{s-t}\int_t^s \boldsymbol{g}_\theta(\boldsymbol{x}_r,r,t,s)dr\|_2^2$$

$$= \int_t^s \|\boldsymbol{g}_\theta(\boldsymbol{x}_r,r,t,s) - \boldsymbol{v}(\boldsymbol{x}_r,r)\|_2^2 dr$$

$$\quad + \int_t^s \|\frac{1}{s-t}\int_t^s \boldsymbol{v}(\boldsymbol{x}_r,r)dr - \frac{1}{s-t}\int_t^s \boldsymbol{g}_\theta(\boldsymbol{x}_r,r,t,s)dr\|_2^2 dr$$

$$\quad + 2\int_t^s \langle \boldsymbol{g}_\theta(\boldsymbol{x}_r,r,t,s) - \boldsymbol{v}(\boldsymbol{x}_r,r), \frac{1}{s-t}\int_t^s \boldsymbol{v}(\boldsymbol{x}_r,r)dr - \frac{1}{s-t}\int_t^s \boldsymbol{g}_\theta(\boldsymbol{x}_r,r,t,s)dr\rangle dr$$

$$= \int_t^s \|\boldsymbol{g}_\theta(\boldsymbol{x}_r,r,t,s) - \boldsymbol{v}(\boldsymbol{x}_r,r)\|_2^2 dr$$

$$\quad + (s-t)\|\frac{1}{s-t}\int_t^s \boldsymbol{v}(\boldsymbol{x}_r,r)dr - \frac{1}{s-t}\int_t^s \boldsymbol{g}_\theta(\boldsymbol{x}_r,r,t,s)dr\|_2^2$$

$$\quad + 2\langle\int_t^s (\boldsymbol{g}_\theta(\boldsymbol{x}_r,r,t,s) - \boldsymbol{v}(\boldsymbol{x}_r,r))dr, \frac{1}{s-t}\int_t^s \boldsymbol{v}(\boldsymbol{x}_r,r)dr - \frac{1}{s-t}\int_t^s \boldsymbol{g}_\theta(\boldsymbol{x}_r,r,t,s)dr\rangle$$

$$= \int_t^s \|\boldsymbol{g}_\theta(\boldsymbol{x}_r,r,t,s) - \boldsymbol{v}(\boldsymbol{x}_r,r)\|_2^2 dr$$

$$\quad - (s-t)\|\frac{1}{s-t}\int_t^s \boldsymbol{v}(\boldsymbol{x}_r,r)dr - \frac{1}{s-t}\int_t^s \boldsymbol{g}_\theta(\boldsymbol{x}_r,r,t,s)dr\|_2^2$$

$$< \int_t^s \|\boldsymbol{g}_\theta(\boldsymbol{x}_r,r,t,s) - \boldsymbol{v}(\boldsymbol{x}_r,r)\|_2^2 dr,$$

which is a contradictory.

Next, we prove that minimizing $\mathcal{L}_{\star\star}(\theta)$ implies $\boldsymbol{f}_\theta(\boldsymbol{x}_t,t,s) = \boldsymbol{f}(\boldsymbol{x}_t,t,s)$ for all $[t,s] \subseteq [a,b] \subseteq [0,1]$, where $b-a$ is sufficiently small.

Using Picard iteration, we construct a sequence $\boldsymbol{f}_n$ satisfying $\boldsymbol{f}_0 = \boldsymbol{0}, \boldsymbol{f}_{n+1}(\boldsymbol{x}_r + (t-r)\boldsymbol{f}_n(\boldsymbol{x}_r,r,t),t,s) = \boldsymbol{f}(\boldsymbol{x}_t,t,s)$.

When $s \neq t$, we have

$$\sup_{t,s\in[a,b]} \|\boldsymbol{f}_{n+1}(\boldsymbol{x}_t,t,s) - \boldsymbol{f}(\boldsymbol{x}_t,t,s)\|_2^2$$

$$= \sup_{t,s\in[a,b]} \frac{1}{s-t}\int_t^s \|\boldsymbol{f}_{n+1}(\boldsymbol{x}_t,t,s) - \boldsymbol{f}(\boldsymbol{x}_t,t,s)\|_2^2 dr$$

$$\leq \sup_{t,s\in[a,b]} \frac{1}{s-t}\int_t^s \|\boldsymbol{f}_{n+1}(\boldsymbol{x}_t,t,s) - \boldsymbol{f}_{n+1}(\boldsymbol{x}_r + (t-r)\boldsymbol{f}_n(\boldsymbol{x}_r,r,t),t,s)\|_2^2 dr$$

$$\leq L \sup_{t,s\in[a,b]} \frac{1}{s-t}\int_t^s \|\boldsymbol{x}_t - \boldsymbol{x}_r - (t-r)\boldsymbol{f}_n(\boldsymbol{x}_r,r,t)\|_2^2 dr$$

$$= L \sup_{t,s\in[a,b]} \frac{1}{s-t}\int_t^s \|\boldsymbol{x}_r + (t-r)\boldsymbol{f}_n(\boldsymbol{x}_r,r,t) - \boldsymbol{x}_r - (t-r)\boldsymbol{f}(\boldsymbol{x}_r,r,t)\|_2^2 dr$$

$$\leq L \sup_{t,s\in[a,b]} \frac{1}{s-t} \int_t^s |t-r|^2 dr \sup_{r,s\in[a,b]} \|\boldsymbol{f}_n(\boldsymbol{x}_r,r,t) - \boldsymbol{f}(\boldsymbol{x}_r,r,t))\|_2^2$$

$$= L \sup_{t,s\in[a,b]} \frac{(s-t)^2}{3} \sup_{r,s\in[a,b]} \|\boldsymbol{f}_n(\boldsymbol{x}_r,r,t) - \boldsymbol{f}(\boldsymbol{x}_r,r,t))\|_2^2$$

$$\leq \frac{1}{3} L(b-a)^2 \sup_{t,s\in[a,b]} \|\boldsymbol{f}_n(\boldsymbol{x}_t,t,s) - \boldsymbol{f}(\boldsymbol{x}_t,t,s))\|_2^2.$$

Also when $s = t$, the inequality satisfies.

One can see with sufficiently small $b-a$, $\sup_{t,s\in[a,b]} \|\boldsymbol{f}_{n+1}(\boldsymbol{x}_t,t,s) - \boldsymbol{f}(\boldsymbol{x}_t,t,s)\|_2^2$ converges to $\mathbf{0}$. Hence, there exists a small interval, such that when $\mathcal{L}_{\text{STEE}}(\theta)$ reaches the minimum, we have $\boldsymbol{f}_\theta(\boldsymbol{x}_t,t,s) = \boldsymbol{f}(\boldsymbol{x}_t,t,s)$.

$\square$

**Corollary 5** (STEI). *Let $\boldsymbol{f}_\theta(\boldsymbol{x}_t,t,s)$ be a neural network, and let $\boldsymbol{v}(\boldsymbol{x}_t,t)$ be the true velocity field. Assume $\boldsymbol{v}(\boldsymbol{x}_t,t)$ is L-Lipschitz continuous in its first argument. If $\mathcal{L}_{STEI}(\theta)$ reaches its minimum, then for any interval where $|s-t| \leq h$ and $h < \frac{3}{L}$, we have $\boldsymbol{f}_\theta(\boldsymbol{x}_t,t,s) = \boldsymbol{f}(\boldsymbol{x}_t,t,s)$.*

*Proof.* The loss $\mathcal{L}_{\text{STEI}}(\theta)$ is defined as a sum of a secant term and a diffusion term

$$\mathcal{L}_{\text{STEI}}(\theta) = \mathbb{E}[\|\boldsymbol{f}_\theta(\boldsymbol{x}_t,t,s) - \boldsymbol{f}_{\theta^-}(\hat{\boldsymbol{x}}_r,r,r)\|_2^2] + \lambda\mathbb{E}[\|\boldsymbol{f}_\theta(\boldsymbol{x}_\tau,\tau,\tau) - (\alpha'_\tau\boldsymbol{x}_0 + \sigma'_\tau\boldsymbol{z})\|_2^2]$$

where $\hat{\boldsymbol{x}}_r = \boldsymbol{x}_t + (r-t)\boldsymbol{f}_{\theta^-}(\boldsymbol{x}_t,t,r)$. For $\mathcal{L}_{\text{STEI}}(\theta)$ to reach the minimum, both two terms must reach the minimum. By Proposition 1, the diffusion term achieves the minimum if and only if

$$\boldsymbol{f}_\theta(\boldsymbol{x}_t,t,t) = \mathbb{E}(\alpha'_t\boldsymbol{x}_0 + \sigma'_t\boldsymbol{z}|\boldsymbol{x}_t) = \boldsymbol{v}(\boldsymbol{x}_t,t).$$

Substituting this result into the secant term, it becomes the one studied in the proof of Theorem 2. The remainder of the proof follows identically. $\square$

**Corollary 6** (SDEE). *Let $\boldsymbol{f}_\theta(\boldsymbol{x}_t,t,s)$ be a neural network that is L-Lipschitz continuous in its first argument. For any fixed interval $[a,b] \subseteq [0,1]$ with $b-a$ sufficiently small, if $\mathcal{L}_{SDEE}(\theta)$ reaches its minimum, then $\boldsymbol{f}_\theta(\boldsymbol{x}_t,t,s) = \boldsymbol{f}(\boldsymbol{x}_t,t,s)$.*

*Proof.* The SDEE loss is defined as

$$\mathcal{L}_{\text{SDEE}}(\theta) = \mathbb{E}[\|\boldsymbol{f}_\theta(\boldsymbol{x}_r + (t-r)\boldsymbol{f}_{\theta^-}(\boldsymbol{x}_r,r,t),t,s) - \boldsymbol{v}(\boldsymbol{x}_r,r)\|_2^2].$$

This loss function is identical to the intermediate objective $\mathcal{L}_*(\theta)$ analyzed in the proof of Theorem 3. The proof, therefore, follows that of Theorem 3 directly. $\square$

## B  Derivations

### B.1  Applying to EDM

The training objective of EDM [6] is

$$\mathcal{L}_{\text{EDM}}(\theta) = \mathbb{E}_{\sigma,\boldsymbol{y},\boldsymbol{n}} w(\sigma)\|\boldsymbol{F}_\theta(c_{\text{in}}(\sigma)\cdot(\boldsymbol{y}+\boldsymbol{n}),c_{\text{noise}}(\sigma)) - \frac{1}{c_{\text{out}}(\sigma)}(\boldsymbol{y} - c_{\text{skip}}(\sigma)\cdot(\boldsymbol{y}+\boldsymbol{n}))\|_2^2, \quad (24)$$

where $\boldsymbol{F}_\theta$ is the neural network, $w(\sigma)$ the loss weighting factor, $c_{\text{in}}(\sigma) = \frac{1}{\sqrt{\sigma^2+\sigma_d^2}}$, $c_{\text{out}}(\sigma) = \frac{\sigma\cdot\sigma_d}{\sqrt{\sigma^2+\sigma_d^2}}$, $c_{\text{skip}}(\sigma) = \frac{\sigma_d^2}{\sigma^2+\sigma_d^2}$, $c_{\text{noise}} = \frac{1}{4}\ln(\sigma)$, and $\sigma_d = 0.5$ is the variance of the image data $p_d$. Since $\boldsymbol{y} \sim p_d$, $\boldsymbol{n} \sim \mathcal{N}(\boldsymbol{0},\sigma^2\boldsymbol{I})$, we can denote $\boldsymbol{y}$ by $\boldsymbol{x}_0$, $\boldsymbol{n}$ by $\sigma\boldsymbol{z}$ in our notation. Then Eq. (24) can be simplified to

$$\mathcal{L}_{\text{EDM}}(\theta) = \mathbb{E}_{\sigma,\boldsymbol{x}_0,\boldsymbol{z}} \frac{w(\sigma)}{\sigma_d^2} \left\| -\sigma_d \boldsymbol{F}_\theta\left(\frac{1}{\sigma_d}\left(\frac{\sigma_d}{\sqrt{\sigma^2+\sigma_d^2}}\boldsymbol{x}_0 + \frac{\sigma}{\sqrt{\sigma^2+\sigma_d^2}}\sigma_d\boldsymbol{z}\right), c_{\text{noise}}(\sigma)\right) \right.$$
$$\left. - \left(-\frac{\sigma}{\sqrt{\sigma^2+\sigma_d^2}}\boldsymbol{x}_0 + \frac{\sigma_d}{\sqrt{\sigma^2+\sigma_d^2}}\sigma_d\boldsymbol{z}\right) \right\|_2^2. \quad (25)$$

The Trigflow framework [35] demonstrates that if letting $t = \arctan(\frac{\sigma}{\sigma_d})$ and $\boldsymbol{x}_t = \cos(t)\boldsymbol{x}_0 + \sin(t)\sigma_d\boldsymbol{z}$ where $t \in [0, \frac{\pi}{2}]$, training with the simplified objective

$$\mathcal{L}_{\text{EDM}}(\theta) = \mathbb{E}_{\boldsymbol{x}_0,\boldsymbol{z},t} \frac{w(t)}{\sigma_d^2} \left\| -\sigma_d\boldsymbol{F}_\theta \left( \frac{\boldsymbol{x}_t}{\sigma_d}, c_{\text{noise}}(\sigma_d \cdot \tan(t)) \right) - (-\sin(t)\boldsymbol{x}_0 + \cos(t)\sigma_d\boldsymbol{z}) \right\|_2^2 \quad (26)$$

leads to the diffusion ODE

$$\frac{d\boldsymbol{x}_t}{dt} = -\sigma_d\boldsymbol{F}_\theta \left( \frac{\boldsymbol{x}_t}{\sigma_d}, c_{\text{noise}}(\sigma_d \cdot \tan(t)) \right). \quad (27)$$

As a result, to adapt the notions in Section 3 to the EDM loss, we can modify the range of time to $[0, \frac{\pi}{2}]$, substitute $\boldsymbol{z}$ with $\sigma_d\boldsymbol{z}$ and let $\boldsymbol{v}(\boldsymbol{x}_t, t) = -\sigma_d\boldsymbol{F}_\theta \left( \frac{\boldsymbol{x}_t}{\sigma_d}, c_{\text{noise}}(\sigma_d \cdot \tan(t)) \right)$, $\alpha_t = \cos(t)$, $\sigma_t = \sin(t)$.

For practical choices in transferring EDM to the secant version, we simply set the loss weight $\frac{w(t)}{\sigma_d} = 1$. Besides, since $t \to 0$ or $t \to \frac{\pi}{2}$ makes $c_{\text{noise}}(\sigma_d \tan(t)) = \frac{1}{4}\ln(\sigma_d \tan(t)) \to \infty$, we constrain $t$ and $s$ within the range of $[0.001, \frac{\pi}{2} - 0.00625]$.

### B.2 Applying to SiT

The training objective of SiT [39][4] is the flow matching loss

$$\mathcal{L}_{\text{FM}}(\theta) = \mathbb{E}_{\boldsymbol{x}_0,\boldsymbol{x}_1,t} \left\| \boldsymbol{v}_\theta(\boldsymbol{x}_t, t) - (\boldsymbol{x}_1 - \boldsymbol{x}_0) \right\|_2^2. \quad (28)$$

To adapt our notation to loss Eq. (28), we can substitute $\boldsymbol{x}_0$ with $\boldsymbol{x}_1$, and $\boldsymbol{z}$ with $\boldsymbol{x}_0$, and set $\alpha_t = t$ and $\sigma_t = 1 - t$.

## C  Model Parametrization

### C.1  The Secant Version of EDM

The parametrization is similar to that in CTM [34]. Specifically, we apply the same time embedder for the extra $s$ input as that employed for $t$. In each UNet block, we add affine layers for the $s$ embedding, which projects the $s$ embedding into the AdaGN [5] parameters, akin to that for $t$ embedding. The resulting AdaGN parameters from both $s$ and $t$ are then added up. The weights of all the added layers are randomly initialized.

### C.2  The Secant Version of DiT

We clone (including the pretrained parameters) the time embedder of $t$ as the $s$ embedder. The original time embedding is replaced with half of the summed embeddings from $t$ and $s$. This design ensures $\boldsymbol{f}_\theta(\boldsymbol{x}_t, t, t) = \boldsymbol{v}_\theta(\boldsymbol{x}_t, t)$ when loading the pretrained SiT weights. For classifier-free guidance, we add another time embedder for the CFG scale $w$, whose weights are randomly initialized.

We also tried the parametrization in Section C.1 for DiT, and find that the performances are comparable. However, this implementation introduces significantly more parameters ($\sim 226\text{M}$), since the module for producing AdaLN parameters is very large.

## D  Training Algorithms for STEI and SDEE

Similar to Algorithm 1 for $\mathcal{L}_{\text{SDEI}}$ and Algorithm 2 for $\mathcal{L}_{\text{STEE}}$, we provide the training algorithms with $\mathcal{L}_{\text{STEI}}$ and $\mathcal{L}_{\text{SDEE}}$ in Algorithm 3 and Algorithm 4, respectively.

---

[4]The loss configuration follows *linear path* and *velocity prediction*. Details can be found at https://github.com/willisma/SiT.

**Algorithm 3** Secant Training by Estimating the Interior Point (STEI)

**Input:** dataset $\mathcal{D}$, neural network $\boldsymbol{f}_\theta$, learning rate $\eta$
**repeat**
  $\theta^- \leftarrow \theta$
  Sample $\boldsymbol{x} \sim \mathcal{D}$, $\boldsymbol{z} \sim \mathcal{N}(\boldsymbol{0}, \boldsymbol{I})$
  Sample $t$ and $s$
  Sample $r \sim \mathcal{U}[0,1]$, $r \leftarrow t + r(s-t)$
  $\boldsymbol{x}_t \leftarrow t\boldsymbol{x} + (1-t)\boldsymbol{z}$
  $\hat{\boldsymbol{x}}_r \leftarrow \boldsymbol{x}_t + (r-t)\boldsymbol{f}_{\theta^-}(\boldsymbol{x}_t, t, r)$
  $\boldsymbol{v}_r \leftarrow \boldsymbol{v}(\hat{\boldsymbol{x}}_r, r)$
  Sample $\tau \in \mathcal{U}[0,1]$
  $\boldsymbol{x}_\tau \leftarrow \tau\boldsymbol{x} + (1-\tau)\boldsymbol{z}$
  $\boldsymbol{u}_\tau \leftarrow \boldsymbol{x} - \boldsymbol{z}$
  $\mathcal{L}(\theta) = \mathbb{E}_{\boldsymbol{x}, \boldsymbol{z}, t, s, r}\|\boldsymbol{f}_\theta(\boldsymbol{x}_t, t, s) - \boldsymbol{f}_\theta(\boldsymbol{x}_r, r, r)\|_2^2$
  $+\lambda\mathbb{E}_{\boldsymbol{x}_0, \boldsymbol{z}, \tau}\|\boldsymbol{f}_\theta(\boldsymbol{x}_\tau, \tau, \tau) - \boldsymbol{u}_\tau\|_2^2$
  $\theta \leftarrow \theta - \eta\nabla_\theta\mathcal{L}(\theta)$
**until** convergence

**Algorithm 4** Secant Distillation by Estimating the End Point (SDEE)

**Input:** dataset $\mathcal{D}$, neural network $\boldsymbol{f}_\theta$, teacher diffusion model $\boldsymbol{v}$, learning rate $\eta$
**repeat**
  $\theta^- \leftarrow \theta$
  Sample $\boldsymbol{x} \sim \mathcal{D}$, $\boldsymbol{z} \sim \mathcal{N}(\boldsymbol{0}, \boldsymbol{I})$
  Sample $t$ and $s \sim \mathcal{U}[0,1]$
  Sample $r \sim \mathcal{U}[0,1]$, $r \leftarrow t + r(s-t)$
  $\boldsymbol{x}_r \leftarrow r\boldsymbol{x} + (1-r)\boldsymbol{z}$
  $\hat{\boldsymbol{x}}_t \leftarrow \boldsymbol{x}_r + (t-r)\boldsymbol{f}_{\theta^-}(\boldsymbol{x}_r, r, t)$
  $\mathcal{L}(\theta) = \mathbb{E}_{\boldsymbol{x}, \boldsymbol{z}, t, s, r}\|\boldsymbol{f}_\theta(\hat{\boldsymbol{x}}_t, t, s) - \boldsymbol{v}(\boldsymbol{x}_r, r)\|_2^2$
  $\theta \leftarrow \theta - \eta\nabla_\theta\mathcal{L}(\theta)$
**until** convergence

**Algorithm 5** Sampling of $t, s$ in SDEI and STEI for EDM

**Input:** Gaussian distribution parameter $P_{\text{mean}}$ and $P_{\text{std}}$, number of steps $N$, boundary constants $\epsilon_1$ and $\epsilon_2$
**Output:** sampled time point $t$ and $s$
Sample $\sigma \sim \mathcal{N}(P_{\text{mean}}, P_{\text{std}})$
$t \leftarrow \arctan(\frac{\sigma}{\sigma_d})$
Clip $t$ into $[\epsilon_1, \frac{\pi}{2} - \epsilon_2]$
Round $t$ to be discrete
Sample $d \sim \mathcal{U}(0, \frac{1}{N})$
$s \leftarrow t - d$
Clip $s$ into $[\epsilon_1, \frac{\pi}{2} - \epsilon_2]$

**Algorithm 6** Sampling of $t, s$ in SDEE and STEE for EDM

**Input:** Gaussian distribution parameter $P_{\text{mean}}$ and $P_{\text{std}}$, number of steps $N$, boundary constants $\epsilon_1$ and $\epsilon_2$
**Output:** sampled time point $t$ and $s$
Sample $\sigma \sim \mathcal{N}(P_{\text{mean}}, P_{\text{std}})$
$t \leftarrow \arctan(\frac{\sigma}{\sigma_d})$
Clip $t$ into $[\epsilon_1, \frac{\pi}{2} - \epsilon_2]$
Sample $d \sim \mathcal{U}(0, \frac{1}{N})$
$d \leftarrow -d$ with probability of $0.5$
$s \leftarrow t - d$
Clip $s$ into $[\epsilon_1, \frac{\pi}{2} - \epsilon_2]$

**Algorithm 7** Sampling of $t, s$ in SDEI and STEI for DiT

**Input:** number of steps $N$
**Output:** sampled time point $t$ and $s$
Sample $d \sim \mathcal{U}(0, \frac{1}{N})$
Sample $t \sim \mathcal{U}(0, 1-d)$
Round $t$ to be discrete (SDEI only)
$s \leftarrow t + d$

**Algorithm 8** Sampling of $t, s$ in SDEE and STEE for DiT

**Input:** number of steps $N$
**Output:** sampled time point $t$ and $s$
Sample $d \sim \mathcal{U}(0, \frac{1}{N})$
Sample $t \sim \mathcal{U}(0, 1-d)$
$s \leftarrow t + d$
Swap $t$ and $s$ with probability of $0.5$

# E  Sampling Time Points During Training

**Sampling $t$ and $s$ for EDM.** We sample $\sigma$ from the Gaussian proposal distribution with $P_{\text{mean}} = -1.0$ and $P_{\text{std}} = 1.4$ [35], which we find sightly better than the default configuration of $P_{\text{mean}} = -1.2$ and $P_{\text{std}} = 1.2$ in EDM [6], and derive $t$ by $t = \arctan(\frac{\sigma}{\sigma_d})$. Then, we sample the distance $d = |s - t|$ by $d \sim \mathcal{U}(0, \frac{1}{N})$, where $N$ is the pre-set number of steps. If the loss estimates the interior point, we round the sampled $t$ to be discrete; while for losses that estimate the end point, we randomly multiply $-1$ to $d$ with a probability of $0.5$. Finally, we get $s$ by $s = t - d$. The sampling processes are illustrated in Algorithm 5 and Algorithm 6.

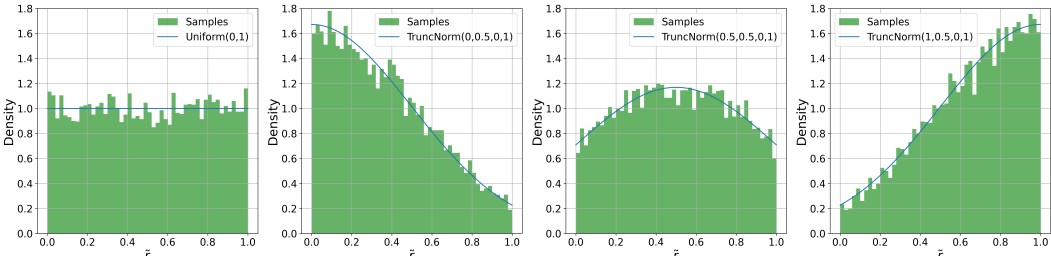

Figure 7: Uniform distribution and truncated normal distribution under different $\mu$ values.

**Sampling $t$ and $s$ for DiT.** We first sample the distance $d = |s - t|$ by $d \sim \mathcal{U}(0, \frac{1}{N})$. Then, we sample $t$ by $t \sim \mathcal{U}(0, 1 - d)$. If estimating the interior point, we round $t$ to be discrete and derive $s$ by $s = t + d$; if estimating the end point, we get $s$ by $s = t + d$ and randomly swap $t$ and $s$ with a probability of $0.5$. The sampling strategies are illustrated in Algorithm 7 and Algorithm 8.

**Sampling $r$.** The default sampling strategy for $r$ follows a uniform distribution between the endpoints $t$ and $s$. Specifically, we first draw $\tilde{r} \sim \mathcal{U}(0, 1)$ from a standard uniform distribution, then obtain $r$ through the linear transformation $r = t + \tilde{r}(s - t)$.

In our ablation studies, we explore alternative sampling strategies using the truncated normal distribution. Specifically, we sample $\tilde{r}$ from a truncated normal distribution $\tilde{r} \sim \mathcal{TN}(\mu, \sigma, 0, 1)$, where the distribution is confined to the interval $[0, 1]$. The value of $r$ is then obtained through the same linear transformation $r = t + \tilde{r}(s - t)$. Let $\tilde{q}(\tilde{r})$ denote the probability density function of $\mathcal{TN}(\mu, \sigma, 0, 1)$. Assuming $s \neq t$ without loss of generality, we can derive

$$r \sim q(r) = \frac{1}{|s - t|}\tilde{q}\left(\frac{r - t}{s - t}\right). \tag{29}$$

Then Eq. (20) becomes

$$
\begin{aligned}
\boldsymbol{f}(\boldsymbol{x}_t, t, s) &= \mathbb{E}_{r \sim q(r)} \boldsymbol{v}(\boldsymbol{x}_r, r) \frac{p_{\mathcal{U}(t,s)}(r)}{q(r)} \\
&= \mathbb{E}_{\tilde{r} \sim \tilde{q}(\tilde{r})} \boldsymbol{v}(\boldsymbol{x}_r, r) \frac{\frac{1}{|s-t|}}{\frac{1}{|s-t|}\tilde{q}(\tilde{r})} \\
&= \mathbb{E}_{\tilde{r} \sim \tilde{q}(\tilde{r})} \boldsymbol{v}(\boldsymbol{x}_r, r) \frac{1}{\tilde{q}(\tilde{r})}.
\end{aligned}
\tag{30}
$$

The probability distributions of $\tilde{r}$ under different $\mu$ values are illustrated in Fig. 7.

## F  Sampling Algorithms at Inference

For simplicity, we always sample adjacent time steps with uniform spacing. The sampling process differs between STEE-trained models and those trained with SDEI, STEI, or SDEE, due to their distinct handling of classifier-free guidance. The detailed sampling processes are presented in Algorithm 9 and Algorithm 10.

---

**Algorithm 9** Sampling Using Models Trained with SDEI, STEI and SDEE

**Input:** model $\boldsymbol{f}_\theta$, number of steps $N$, guidance scale $w$
**Output:** sampled image $\boldsymbol{x}_0$
Sample $\boldsymbol{x}_1 \sim \mathcal{N}(\boldsymbol{0}, \boldsymbol{I})$
**for** $i = 0$ to $N - 1$ **do**
    $\boldsymbol{x}_{\frac{N-i-1}{N}} \leftarrow \boldsymbol{x}_{\frac{N-i}{N}} - \frac{1}{N}\boldsymbol{f}_\theta(\boldsymbol{x}_{\frac{N-i}{N}}, \frac{N-i}{N}, \frac{N-i-1}{N}, w)$
**end for**

---

Table 7: The performance comparison of secant losses on CIFAR-10 and ImageNet-256 × 256.

| | | CIFAR-10 | ImageNet-256 × 256 | | | |
|---|---|---|---|---|---|---|
| Type | Steps | FID↓ | FID↓ | IS↑ | FID (w/ CFG)↓ | IS (w/ CFG)↑ |
| SDEI | 1 | 22.67 | 43.49 | 54.40 | 8.97 | 253.25 |
| SDEI | 2 | 5.88 | 20.83 | 93.17 | 4.81 | 257.56 |
| SDEI | 4 | 3.23 | 14.21 | 116.03 | 3.11 | 258.81 |
| SDEI | 8 | 2.27 | 9.14 | 139.69 | 2.46 | 248.36 |
| STEI | 1 | 36.87 | 38.44 | 56.60 | 7.12 | 241.75 |
| STEI | 2 | 9.21 | 21.10 | 95.77 | 4.41 | 241.99 |
| STEI | 4 | 4.04 | 10.91 | 135.03 | 2.78 | 269.87 |
| STEI | 8 | 2.59 | 7.64 | 157.03 | 2.36 | 274.72 |
| SDEE | 4 | 10.19 | 19.99 | 96.87 | 3.96 | 247.20 |
| SDEE | 8 | 3.18 | 9.46 | 136.97 | 2.46 | 258.94 |
| STEE | 4 | 10.55 | 22.82 | 83.87 | 3.02 | 274.00 |
| STEE | 8 | 3.78 | 12.98 | 110.03 | 2.33 | 274.47 |

---

**Algorithm 10** Sampling Using Models Trained with STEE

**Input:** model $f_\theta$, number of steps $N$, guidance scale $w$
**Output:** sampled image $x_0$
Sample $x_1 \sim \mathcal{N}(\mathbf{0}, \boldsymbol{I})$
**for** $i = 0$ to $N - 1$ **do**
  **if** $w > 1$ **then**
    $f_\theta(x_{\frac{N-i}{N}}, \frac{N-i}{N}, \frac{N-i-1}{N}) \leftarrow f_\theta^u(x_{\frac{N-i}{N}}, \frac{N-i}{N}, \frac{N-i-1}{N}) + w(f_\theta^c(x_{\frac{N-i}{N}}, \frac{N-i}{N}, \frac{N-i-1}{N}) - f_\theta^u(x_{\frac{N-i}{N}}, \frac{N-i}{N}, \frac{N-i-1}{N}))$
  **end if**
  $x_{\frac{N-i-1}{N}} \leftarrow x_{\frac{N-i}{N}} - \frac{1}{N} f_\theta(x_{\frac{N-i}{N}}, \frac{N-i}{N}, \frac{N-i-1}{N})$
**end for**

---

# G  Additional Quantitative Results

## G.1  Additional Performance

For a detailed comparison among secant losses, we provide more results on CIFAR-10 and ImageNet-256 × 256 in Table 7.

## G.2  Training Efficiency

In practical application, methods like continuous-time CMs [33, 35] and MeanFlow [67] require analytical Jacobian-vector product (JVP) operations to compute the loss. This imposes a significant computational burden, especially in PyTorch. To provide a direct comparison, we benchmark the training speed and memory usage under the same EDM setup (the batch size is 512 on 8 A100 GPUs with 64 per GPU) on CIFAR-10. The results are shown in Table 8.

# H  Experimental Settings

The detailed experimental configurations are presented in Table 9. Basically, we follow the settings of EDM and SiT. Except for STEE on ImageNet-256 × 256, we multiply the learning rate by a factor of 0.1. For CIFAR-10 dataset, we use the DDPM++ architecture, and adopts the Trigflow framework. For ImageNet-256 × 256, we cache the latent codes on disk, and for simplicity we disable the horizontal flip data augmentation. For SDEI, STEI and SDEE, we embed CFG scale as a conditional input to the model, with the CFG range $[1, 2]$ for 2-step, 4-step and 8-step models and $[1, 2.5]$ for 1-step ones. In the experiment concerning training from scratch on ImageNet-256 × 256,

Table 8: Comparison on training efficiency among different approaches.

| Method | Training Speed (sec/KIMG) | Memory Usage (G, Per GPU) |
|---|---|---|
| EDM [6] (teacher diffusion model) | 0.57 | 8.69 |
| MeanFlow [67] | 1.04 | 38.73 |
| SDEI | 0.91 | 9.64 |
| STEI | 1.42 | 16.95 |
| SDEE | 0.91 | 9.64 |
| STEI | 0.72 | 9.34 |

Table 9: Experimental settings of four loss functions on different models and datasets.

| | CIFAR-10 | | | | ImageNet-256 $\times$ 256 | | | |
|---|---|---|---|---|---|---|---|---|
| | SDEI | STEI | SDEE | STEE | SDEI | STEI | SDEE | STEE |
| **Model Setting** | | | | | | | | |
| Architecture | DDPM++ | DDPM++ | DDPM++ | DDPM++ | DiT-XL/2 | DiT-XL/2 | DiT-XL/2 | DiT-XL/2 |
| Params (M) | 55 | 55 | 55 | 55 | 675 | 675 | 675 | 675 |
| $\sigma_d$ | 0.5 | 0.5 | 0.5 | 0.5 | - | - | - | - |
| $c_{\text{noise}}(t)$ | $\frac{1}{4}\ln(\sigma_d \tan t)$ | $\frac{1}{4}\ln(\sigma_d \tan t)$ | $\frac{1}{4}\ln(\sigma_d \tan t)$ | $\frac{1}{4}\ln(\sigma_d \tan t)$ | $t$ | $t$ | $t$ | $t$ |
| Boundary $\epsilon_1$ | 0.001 | 0.001 | 0.001 | 0.001 | - | - | - | - |
| Boundary $\epsilon_2$ | 0.00625 | 0.00625 | 0.00625 | 0.00625 | - | - | - | - |
| Initialization | EDM | EDM | EDM | EDM | SiT-XL/2 | SiT-XL/2 | SiT-XL/2 | SiT-XL/2 |
| **Training Setting** | | | | | | | | |
| Precision | fp32 | fp32 | fp32 | fp32 | fp16 | fp16 | fp16 | fp16 |
| Dropout | 0 | 0.2 | 0 | 0.2 | 0 | 0 | 0 | 0 |
| Optimizer | RAdam | RAdam | RAdam | RAdam | AdamW | AdamW | AdamW | AdamW |
| Optimizer $\epsilon$ | 1e-8 | 1e-8 | 1e-8 | 1e-8 | 1e-8 | 1e-8 | 1e-8 | 1e-8 |
| $\beta_1$ | 0.9 | 0.9 | 0.9 | 0.9 | 0.9 | 0.9 | 0.9 | 0.9 |
| $\beta_2$ | 0.99 | 0.99 | 0.99 | 0.99 | 0.999 | 0.999 | 0.999 | 0.999 |
| Learning Rate | 1e-4 | 1e-4 | 1e-4 | 1e-4 | 1e-5 | 1e-5 | 1e-5 | 1e-4 |
| Weight Decay | 0 | 0 | 0 | 0 | 0 | 0 | 0 | 0 |
| Batch Size | 512 | 512 | 512 | 512 | 256 | 256 | 256 | 256 |
| Training iters | 100K | 100k | 100k | 100k | 100K | 100k | 100k | 100k |
| $t, s$ sampling | discrete | discrete | continuous | continuous | discrete | continuous | continuous | continuous |
| $r$ sampling | Uniform | Uniform | Uniform | Uniform | Uniform | Uniform | Uniform | Uniform |
| EMA Rate | EDM's | EDM's | EDM's | EDM's | 0.9999 | 0.9999 | 0.9999 | 0.9999 |
| Label Dropout | - | - | - | - | - | 0.1 | - | 0.1 |
| Embed CFG | - | - | - | - | ✓ | ✓ | ✓ | ✗ |
| $x$-flip | ✗ | ✗ | ✗ | ✗ | ✗ | ✗ | ✗ | ✗ |

we maintain identical settings to those specified in the STEE column, while only disable the pretrained weights and alter the model size. To calculate the FID and IS score at evaluation, we use the codebase provided in MAR [85] for simplicity. The result of Flow-DPM-Solver [79, 16] in Table 3 is with CFG scale of $1.5$. All the experiments can be done on a sever with $8$ NVIDIA A100 GPUs.

# I  More Visualizations

Additional visualization results are presented for both datasets. For CIFAR-$10$ dataset, we provide $4$-step and $8$-step results using $\mathcal{L}_{\text{SDEI}}$ in Fig. 8 and Fig. 9, respectively. For ImageNet-$256 \times 256$, extended visualizations of $8$-step generation using $\mathcal{L}_{\text{STEE}}$ are displayed in Fig. 10 and Fig. 11.

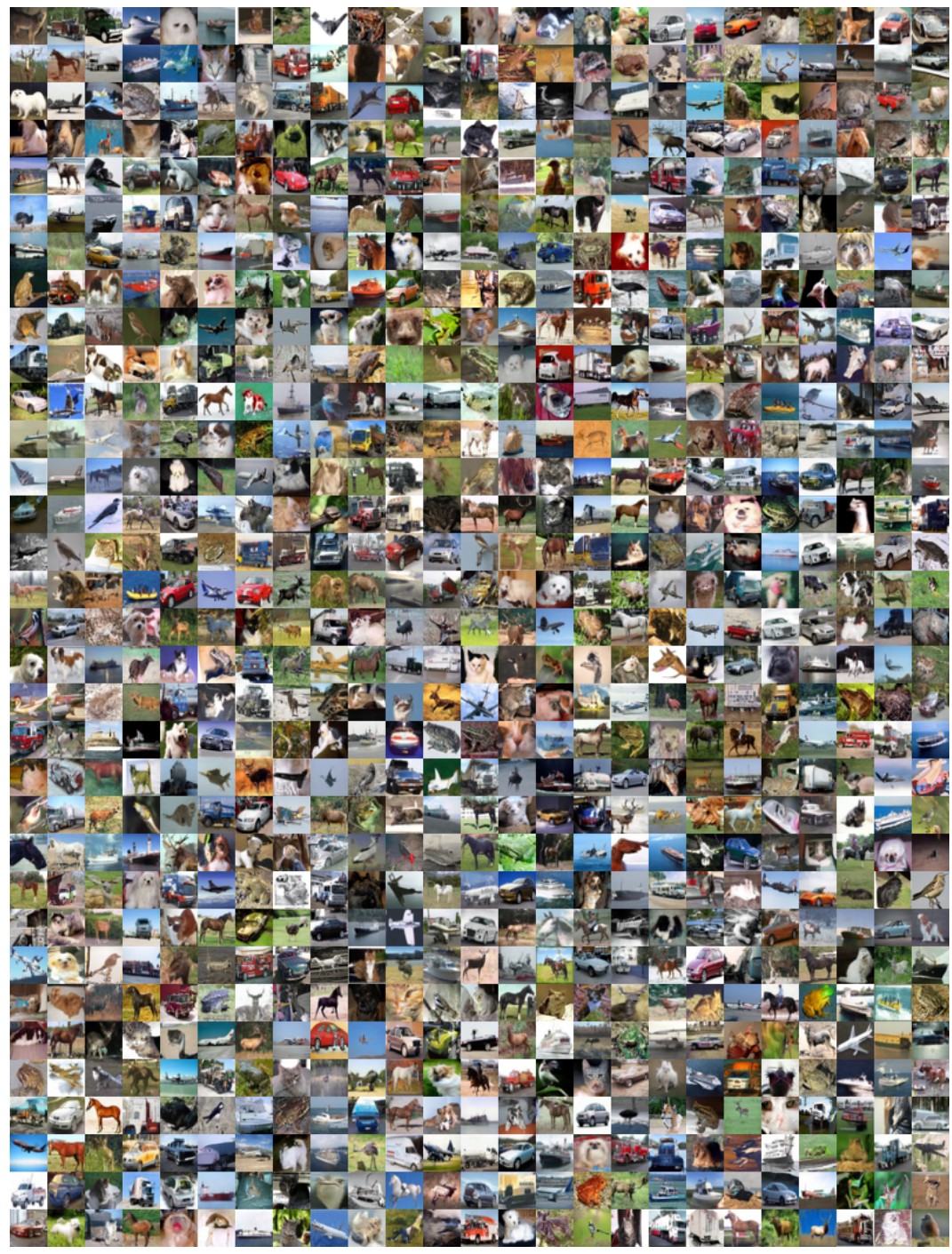

Figure 8: Uncurated 4-step samples on unconditional CIFAR-10.

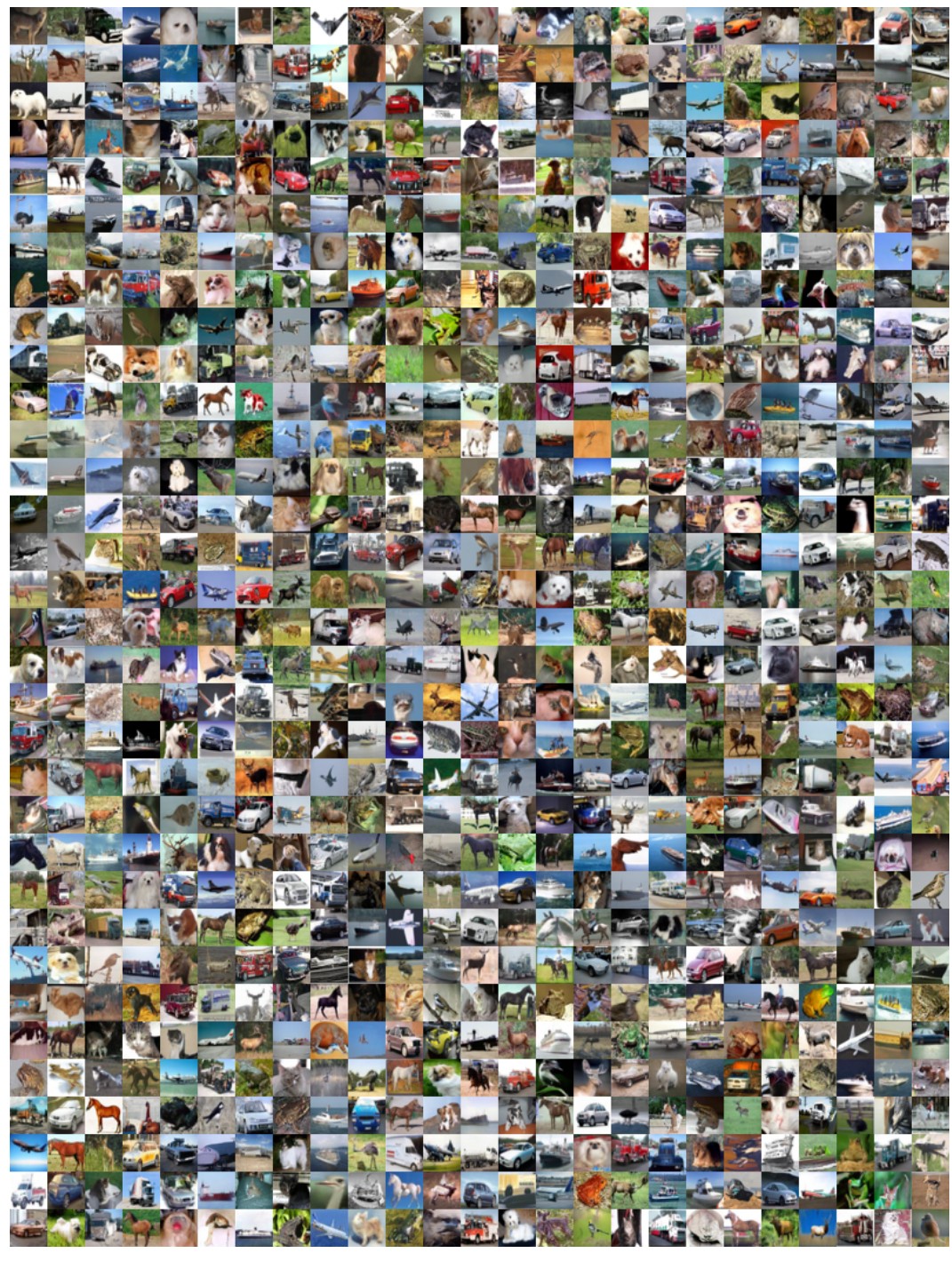

Figure 9: Uncurated 8-step samples on unconditional CIFAR-10.

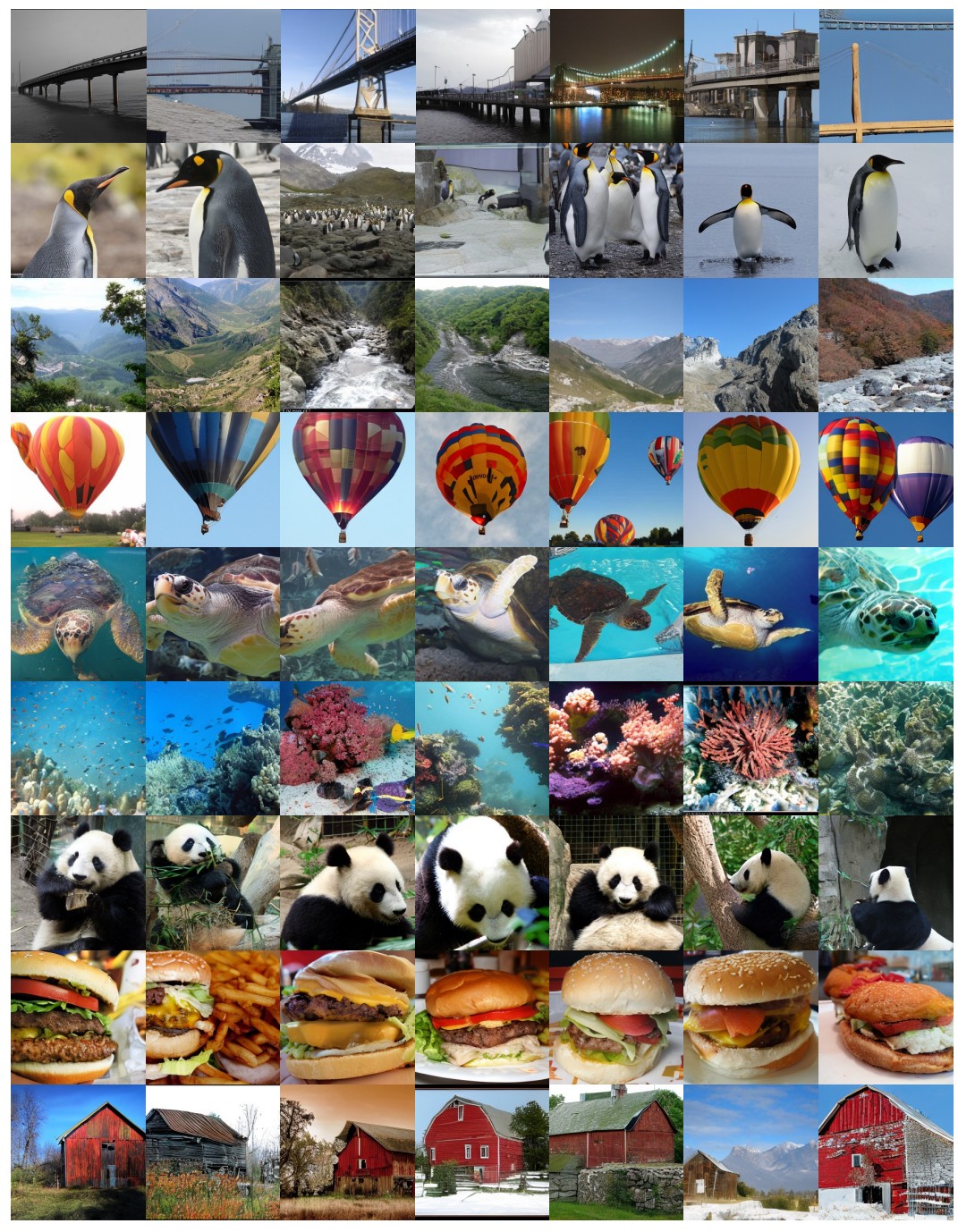

Figure 10: Uncurated 8-step samples on ImageNet-$256 \times 256$. Guidance scale is 2.1, and the guidance of first two steps is ignored.

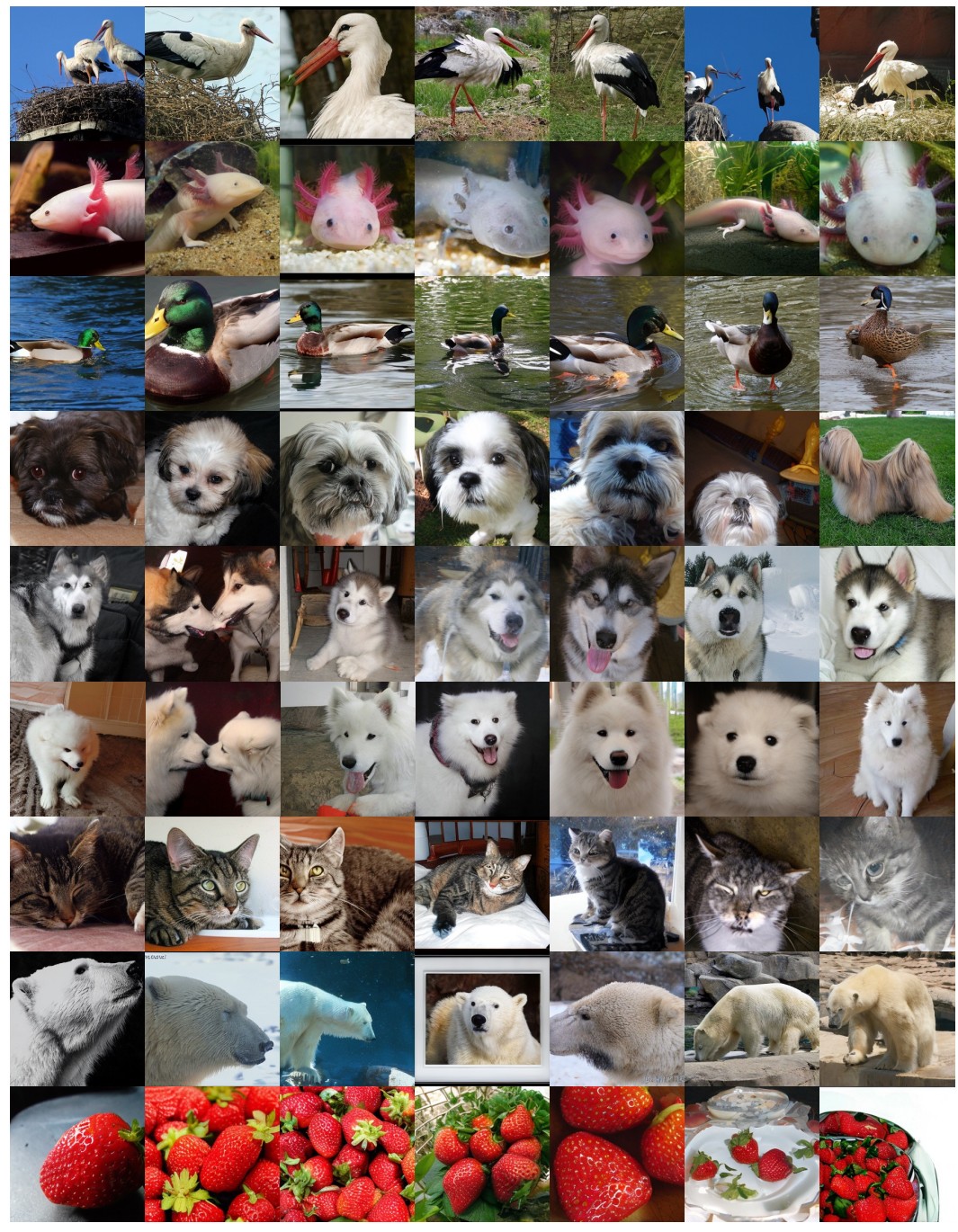

Figure 11: Uncurated 8-step samples on ImageNet-$256 \times 256$. Guidance scale is 2.1, and the guidance of first two steps is ignored.

