# OpenReview forum: "Learning to Integrate Diffusion ODEs by Averaging the Derivatives"
_NeurIPS.cc/2025/Conference — NeurIPS 2025 poster_

### Official Review · Reviewer_RFUp · 2025-07-01

**Clarity:** 3
**Significance:** 3
**Originality:** 2
**Rating:** 4
**Confidence:** 4

**Summary:**

This paper proposes a method for few-step diffusion. Existing fast samplers suffer from significant performance degradation when using a small number of time steps, while previous diffusion distillation methods are prone to training instability and complexity.
In the proposed method, a neural network learns to approximate the secant function, representing the average rate of change along the solution paths of the PF-ODE. This training is performed using a loss function designed from the perspective of Monte Carlo integration of the tangent function, inspired by Picard iteration.
This paper introduces four loss variants, corresponding to combinations of two time-step sampling schemes during training and whether the training is a distillation process or performed from scratch. Theoretical justification for the proposed losses is also provided.
Experiments on CIFAR-10 and ImageNet 256 x 256 demonstrate that the proposed method exhibits consistent stability across various hyperparameter settings and achieves relatively good performance with a small number of time steps.

**Questions:**

* Could you please provide more detailed discussion and information regarding the failure cases of consistency models? Alternatively, performance comparison with consistency models on another benchmark would be also helpful.
* Could you please address the issues raised in Weakness (3)?

**Ethical Concerns:**

["NO or VERY MINOR ethics concerns only"]

**Final Justification:**

The additional discussion and tables in the rebuttal support the advantage of the proposed method with respect to training stability.
Although the issues regarding the Lipschitz condition would remain on theoretical side, the practical experiments verify the effectiveness of the proposed method as a trade-off between performance and efficiency to a certain degree.

**Limitations:**

One potential limitation is that $|s-t|$ should be very small to ensure the convergence to the desirable solution in practical cases.

**Paper Formatting Concerns:**

No concerns.

**Quality:**

3

**Strengths And Weaknesses:**

### Strengths
* The paper is well organized.
* The paper provides theoretical guarantees for the validity of the proposed losses. The theories show an interesting connection between training iterations with stop gradient and Picard iteration.
* The proposed method demonstrates performance comparable to or better than most existing methods, as shown in the tables.
* Extensive evaluations were conducted across various settings to examine the behavior of the proposed method.

### Weaknesses
1. The comparison to and discussion of previous consistency models on ImageNet-256 x 256 appears to be insufficient. The paper only mentions training instability issues encountered by previous consistency models on ImageNet-256 x 256. It remains unclear whether appropriate hyperparameters and settings were used for training these models. A more detailed discussion of the failure cases of consistency models would enhance the clarity and quality of the paper and highlight the stability of the proposed method.
2. The condition for $|s - t|$ in Theorems 2 and 3 appears to be too restrictive for practical few-step diffusions. The magnitude should be $O(L^{-1})$ for Theorem 2 and $O(L^{-1/2})$ for Theorem 3, where $L$ denotes the Lipschitz constant of $v(x_t, t)$ and $f_\theta (x_t, t, s)$, respectively. It is generally not guaranteed that the neural network or the flow is Lipschitz in practical cases. This could be a limitation of the proposed method.
3. There is subtle missing information and errors in the formulas.
    * The definition of the prime symbol ', around Eq. (1) and Proposition 1, is unclear, while I presume it denotes the derivative with respect to time t.
    * The variables a and b are not used in the statement of Theorem 3 in the main part of the paper. Please clarify the relationship between $[a, b]$ and $[t, s]$ in Theorem 3.
    * In the proof of Th.3, there are confusing duplications of the variables, $dr$, in the nested integrations. Please use different variables for the nested integrals, e.g., $dr$ and $d\tilde{r}$.
    * $=0$ in the middle of the equation around L554 is possibly a typo.

---

> ### Author Rebuttal · Authors · 2025-07-30
>
> We sincerely thank the reviewer for the valuable feedback. We are grateful for the recognition of our paper's clear organization and the interesting connections provided by our theory. We address the reviewer's concerns below.
>
> ### **Regarding Weakness 1**
> > **Reviewer:** The comparison to and discussion of previous consistency models on ImageNet-256$\times$256 appears to be insufficient. The paper only mentions training instability issues encountered by previous consistency models on ImageNet-256$\times$256. It remains unclear whether appropriate hyperparameters and settings were used for training these models. A more detailed discussion of the failure cases of consistency models would enhance the clarity and quality of the paper and highlight the stability of the proposed method.
>
> We thank the reviewer for indicating this important aspect. For more detailed discussion, we focus on the analysis of the **target stability**. Target stability is a rather crucial factor to train few-step diffusion models, which means the stability of the prediction target in the loss function [1]. The difference in target stability between CM and ours is essential in principle, which cannot be influenced by specific hyperparameters or settings. We apologize for not elaborating on this in the main text. For simplicity, let's compare the targets under the OT-FM interpolant ($d$ denotes some distance metric):
>
> * **Diffusion Models (Flow Matching):** $\mathcal{L}(\theta)=d(\\boldsymbol{v}\_\theta(\\boldsymbol{x}\_t,t)-(-\\boldsymbol{x}_0+\\boldsymbol{z}))$. The target is the velocity estimation $-\\boldsymbol{x}\_0+\\boldsymbol{z}$.
> * **MeanFlow / CM / CTM (similar):** $\mathcal{L}(\theta)=d(\\boldsymbol{f}\_\theta(\\boldsymbol{x}\_t,t,s)-((-\\boldsymbol{x}_0+\\boldsymbol{z})+(s-t)\\frac{d}{dt}\\boldsymbol{f}\_{\theta^-}(\\boldsymbol{x}\_t,t,s))$. The target is $-\\boldsymbol{x}_0+\\boldsymbol{z} + (s - t)\\frac{d}{dt}\\boldsymbol{f}\_{\theta^-}(\\boldsymbol{x}\_t,t,s)$. The latter item $\\frac{d}{dt}\\boldsymbol{f}\_{\theta^-}(\\boldsymbol{x}\_t,t,s)$ (or the discrete version $\\frac{\\boldsymbol{f}\_{\theta^-}(\\boldsymbol{x}\_t,t,s)-\\boldsymbol{f}\_{\theta^-}(\\boldsymbol{x}\_{t-\Delta t},t-\Delta t,s)}{\Delta t}$) is highly unstable, which has been substantially studied in [1].
> * **Our Method:** The targets are either the velocity estimation $(-\\boldsymbol{x}\_0+\\boldsymbol{z})$ or the velocity $\\boldsymbol{v}(\\boldsymbol{x}_t,t)$ itself.
>
> Our method is the **only** one among these to feature a target with stability comparable to that of standard diffusion models. This inherent stability of the prediction target helps explain the robust training behavior we observe. To see the practical target stability, we also conduct an experiment with EDM on CIFAR-10 (the first table below) and also SiT on ImageNet-256 (the second table below), where we track the specific items in the loss functions, and record the **standard deviation** in the following table.
>
> | Loss (Inspected item)/Training iteration | 10 | 20 | 30 | 40 | 50 | 100 | 200 | 300 | 400 | 500 |
> | :--- | :---: | :---: | :---: | :---: | :--- | :---: | :---: | :--- | :---: | :---: |
> | CM ($\\frac{d}{dt}\\boldsymbol{f}\_{\theta^-}(\\boldsymbol{x}\_t,t,s))$) | 0.2627 | 15.4339 | 30.9154 | 40.7293 | 79.1750 | 101.8844 | 97.2164 | 134.4441 | 72.3149 | 97.4782 |
> | CM ($\\frac{\\boldsymbol{f}\_{\theta^-}(\\boldsymbol{x}\_t,t,s)-\\boldsymbol{f}\_{\theta^-}(\\boldsymbol{x}\_{t-\Delta t},t-\Delta t,s)}{\Delta t}$) | 0.2810 | 10.1618 | 34.4472 | 88.6847 | 39.1018 | 59.5919 | 143.4175 | 62.3149 | 69.5990 | 137.7932 |
> | SDEI ($\\boldsymbol{v}(\\boldsymbol{x}_r,r)$) | 3.1713 | 3.0427 | 4.1342 | 3.4504 | 3.3060 | 2.6381 | 3.2294 | 3.5089 | 3.4080 | 3.2673 |
> | STEI ($\\boldsymbol{f}_{\theta^-}(\\boldsymbol{x}_r,r,r)$) | 4.4030 | 4.2680 | 4.1115 | 3.8282 | 4.3783 | 4.5072 | 4.3712 | 4.0905 | 3.9820 | 4.0967 |
> | SDEE ($\\boldsymbol{v}(\\boldsymbol{x}_r,r)$) | 2.8809 | 3.2181 | 3.5562 | 3.0221 | 3.7394 | 3.0247 | 3.8801 | 3.2247 | 4.0112 | 3.5830 |
> | STEE ($-\\text{sin}(t)\\boldsymbol{x}_0+\\text{cos}(t)\\boldsymbol{z}$) | 4.2049 | 4.1783 | 4.6112 | 4.0290 | 4.6454 | 4.5360 | 4.9838 | 4.9387 | 5.0872 | 4.3094 |
>
> | Loss (Inspected item)/Training iteration | 10 | 20 | 30 | 40 | 50 | 100 | 200 | 300 | 400 | 500 |
> | :--- | :---: | :---: | :---: | :---: | :--- | :---: | :---: | :--- | :---: | :---: |
> | CM ($\\frac{d}{dt}\\boldsymbol{f}\_{\theta^-}(\\boldsymbol{x}\_t,t,s))$) | 32.7539 | 60.0200 | 62.2378 | 98.2755 | 139.4081 | 299.9317 | 307.6916 | 843.4188 | 1444.6832 | 1512.0031 |
> | CM ($\\frac{\\boldsymbol{f}\_{\theta^-}(\\boldsymbol{x}\_t,t,s)-\\boldsymbol{f}\_{\theta^-}(\\boldsymbol{x}\_{t-\Delta t},t-\Delta t,s)}{\Delta t}$) | 44.9663 | 75.9018 | 145.2046 | 158.4578 | 279.9238 | 373.5405 | 316.0669 | 740.3381 | 1397.3980 | 1622.9102 |
> | SDEI ($\\boldsymbol{v}(\\boldsymbol{x}_r,r)$) | 3.9272 | 4.2610 | 5.5458 | 4.7783 | 5.0217 | 4.3936 | 4.2743 | 4.8920 | 5.6929 | 5.7791 |
> | STEI ($\\boldsymbol{f}_{\theta^-}(\\boldsymbol{x}_r,r,r)$) | 5.4390 | 5.9166 | 4.8327 | 6.4482 | 6.0106 | 4.8009 | 6.2592 | 6.3983 | 4.0272 | 5.8081 |
> | SDEE ($\\boldsymbol{v}(\\boldsymbol{x}_r,r)$) | 4.3985 | 6.5008 | 4.4519 | 3.6164 | 5.2973 | 4.7842 | 5.3170 | 5.3725 | 5.2417 | 6.0224 |
> | STEE ($-\\boldsymbol{x}_0+\\boldsymbol{z}$) | 7.4445 | 6.8171 | 10.1249 | 9.7260 | 10.0295 | 10.1284 | 8.5793 | 9.6876 | 9.2053 | 9.9978 |
>
> The statistics shown in the tables clearly support the stability of our method. While, for CMs, the target is unstable, especially when the data has large variance (e.g., the latent space of ImageNet-256). The instability largely affects the training dynamics, potentially leading to worse performance, or even training divergence.
>
> ### **Regarding Weakness 2**
> > **Reviewer:** The condition for $|s−t|$ in Theorems 2 and 3 appears to be too restrictive for practical few-step diffusions. The magnitude should be $O(L^{−1})$ for Theorem 2 and $O(L^{−1/2})$ for Theorem 3, where $L$ denotes the Lipschitz constant of $v(x_t,t)$ and $f_\theta(x_t,t,s)$, respectively. It is generally not guaranteed that the neural network or the flow is Lipschitz in practical cases. This could be a limitation of the proposed method.
>
> We agree with the reviewer that Lipschitz continuity is not generally guaranteed for a standard neural network in practice. However, we work in the context of diffusion models, where this condition is a necessary condition to apply foundational results like the Picard-Lindelöf theorem, which guarantees the existence and uniqueness of the ODE solution. This provides a rigorous framework for analyzing our algorithm's convergence, stability, and correctness, a practice common to many works in this field. For example, the seminal work, and perhaps the most related to our work, consistency models [2], and consistency trajectory models [3], both include the Lipchitz condition in the Theorems. And the theoretical guarantee is under the worst case; in practice, the strong empirical stability of our method suggests that our training procedure implicitly guides the network to a well-behaved state. The learned function appears to be locally smooth in the relevant regions of the state space, thus closely approximating the conditions required by the theory.
>
> And we also agree with the reviewer that, the requirement on $|s-t|$ affects the performance in extremely low-NFE settings. In this situation, the small numerical error from one step could be amplified in subsequent steps. Here we would like to highlight that it is a key trade-off compared to consistency models in the theoretical design. CMs emphasize global accuracy, but introduce either discretization error or explicit differentiation part in the loss function, which significantly affects the training stability. Our method achieves local accuracy without explicit differentiation, and has a very stable target as diffusion models. We believe this presents a valuable and complementary approach to CMs. And we position our method between fast samplers and diffusion distillation/training methods, as a good trade-off between performance and efficiency.
>
> ### **Regarding Weakness 3**
> > **Reviewer:** There is subtle missing information and errors in the formulas.
> * The definition of the prime symbol ', around Eq. (1) and Proposition 1, is unclear, while I presume it denotes the derivative with respect to time $t$.
> * The variables $a$ and $b$ are not used in the statement of Theorem 3 in the main part of the paper. Please clarify the relationship between $[a,b]$ and $[t,s]$ in Theorem 3.
> * In the proof of Th.3, there are confusing duplications of the variables, $dr$, in the nested integrations. Please use different variables for the nested integrals, e.g., $dr$ and $d\tilde{r}$.
> * $=0$ in the middle of the equation around L554 is possibly a typo.
>
> i) You are correct; the prime symbol denotes the derivative with respect to time $t$. We will explicitly state this definition in the revised paper.
>
> ii) In Theorem 2, we fix a $t$ and move $s$ in a small neighborhood. While in Theorem 3, it requires sampling time pairs where $t$ and $s$ can appear in any order (also see Figure 1). Therefore, we use a interval $[a,b]$ to depict that increasing the distance $|s-t|$ is accompanied by expanding the interval where $t$ and $s$ are randomly sampled rather than simply moving $s$, as discussed in Remark 4.
>
> iii) Thank you for catching this typo. We will use different variables for the nested integrals in our revision to avoid confusion.
>
> iv) Thank you for pointing out this error. We will correct it in the revised manuscript.
>
> [1] Simplifying, Stabilizing and Scaling Continuous-time Consistency Models. Cheng Lu and Yang Song. ICLR 2025
>
> [2] Consistency models. Yang Song et al. ICML 2023
>
> [3] Consistency trajectory models: Learning probability flow ode trajectory of diffusion. Dongjun Kim et al. ICLR 2024

---

> > ### Comment · Reviewer_RFUp · 2025-08-05
> >
> > I appreciate the authors for the detailed responses to my questions.
> > I think that the additional discussion and tables in the rebuttal support the advantage of the proposed method regarding training stability. I will raise my rating accordingly.
> >
> > Regarding the relationship between $[a, b]$ and $[t, s]$ in Theorem 3, I suggest that the condition $[t,s] \subseteq [a,b]$ be explicitly stated in Theorem 3. I could not find a clear explanation of this condition in the main part of the paper. Would you consider this?

---

> > > ### Author Response · Authors · 2025-08-05
> > >
> > > We sincerely appreciate your positive feedback and for raising your rating. We will incorporate the additional discussions into the final manuscript. Thank you also for pointing out the omission of the condition $[t,s] \subseteq [a,b]$ in Theorem 3. We apologize for the oversight and will ensure the theorem is revised by explicitly stating the condition $[t,s] \subseteq [a,b]$ in the final version.

---

> ### Comment · Area_Chair_vUT8 · 2025-08-04
> **Action Required: Author–Reviewer Discussion Closing Soon**
>
> Dear Reviewer,
>
>
>
> This is a gentle reminder that the **Author–Reviewer Discussion** phase ends within just three days (by **August 6**). Please take a moment to read the authors’ rebuttal thoroughly and engage in the discussion. Ideally, every reviewer will respond so the authors know their rebuttal has been seen and considered.
>
>
>
> Thank you for your prompt participation!
>
>
>
> Best regards,
>
>
>
> AC

---

### Official Review · Reviewer_NLNA · 2025-07-02

**Clarity:** 2
**Significance:** 3
**Originality:** 3
**Rating:** 4
**Confidence:** 4

**Summary:**

The paper proposes a novel approach to accelerate diffusion model inference by learning to integrate ODEs through "secant losses," which approximate the average derivative (secant) between two time points instead of relying on traditional numerical solvers or complex distillation techniques. The authors introduce several variants of secant losses (SDEI, STEI, SDEE, STEE) and demonstrate their effectiveness on CIFAR-10 and ImageNet-256×256, achieving competitive FID scores with fewer steps (e.g., 10-step FID of 2.14 on CIFAR-10 and 8-step FID of 1.96 on ImageNet).

**Questions:**

1. Please quantify the sample complexity introduced by nested Monte Carlo. How much does the inner-loop Monte Carlo increase sample complexity? Does it negate the claimed efficiency gains?

2. Why does the method underperform alternatives (Table 2) despite elegant theory? Are the assumptions (e.g., Lipschitz tangents) unrealistic in practice?

3. For Table 7, why do some losses (STEE) work better? Is the theory missing key practical considerations?

4. Provide proofs for STEI and SDEE unbiasedness. Currently, only SDEI/STEE are covered.

5. Justify the importance sampling in Eq. 18. Why not uniform sampling given enough iterations?

6. Compare wall-clock time (not just NFE) against distillation methods.

**Ethical Concerns:**

["NO or VERY MINOR ethics concerns only"]

**Final Justification:**

The paper proposes a simple and stable method for integrating diffusion ODEs using a derivative-free loss. I find the core idea — learning secants — to be both effective and well-motivated. While the method section could be more clearly presented, especially regarding the loss formulation and the rationale behind the observed stability, the authors have addressed my concerns and clarified key points. I have also independently validated the method's effectiveness, and thus I believe the paper makes a meaningful contribution. I have accordingly raised my score.

**Limitations:**

yes

**Paper Formatting Concerns:**

No more issues or concerns. Please see "Weaknesses" and "Questions" for details. We will raise rating score after all these concerns being well addressed.

**Quality:**

2

**Strengths And Weaknesses:**

Strengths
1. The idea of learning secant approximations instead of relying on traditional ODE solvers or distillation is innovative and well-motivated from a geometric perspective.

2. The paper provides theoretical guarantees (Theorems 2 and 3) that justify the proposed losses under certain conditions.

3. The method achieves competitive results on standard benchmarks, outperforming some fast samplers and distillation techniques.

4. The approach can be applied via fine-tuning or distillation and supports classifier-free guidance.

Weaknesses
1. Table 2 shows underwhelming results: At comparable or larger NFEs, the method underperforms "Joint Training," "Few-step Distillation," and "Few-step Training/Tuning" approaches (e.g., CTM, IMM, rectified flows). This raises concerns about whether the theory overlooks critical practical factors (e.g., error accumulation in large steps).

2. Theoretical assumptions may be too restrictive: The Lipschitz condition is not empirically validated, and the paper does not discuss how violations affect performance.

3. Performance varies drastically across loss types (e.g., SDEI vs. STEE), suggesting the current formulation may not be SOTA. The authors should justify why certain losses (e.g., STEE) perform better and whether the theoretical framework aligns with empirical trends.

4. Theorems 2 and 3 only prove unbiasedness for SDEI and STEE. Missing proofs for STEI and SDEE are a significant gap, as these losses are empirically evaluated.

5. Importance sampling (Eq. 18) is unmotivated: The paper does not explain why uniform sampling is insufficient, especially since training iterations typically compensate for variance.

6. The nested Monte Carlo estimation (e.g., sampling $r$ and $t$) likely requires large sample sizes for unbiasedness, but the paper omits analysis of convergence rates or computational trade-offs.

7. Presentation Issues: (1) Typos: Theorem 3 mixes bold/non-bold variables ($x_1$ vs. $\mathbf{x}_2$).  (2) Proposition 1 lacks attribution: This is a known result (e.g., from diffusion/flow matching literature) but is presented without citation.

---

> ### Author Rebuttal · Authors · 2025-07-30
>
> We sincerely thank the reviewer for the valuable and constructive feedback. We are also grateful for the recognition of our method as innovative and well-motivated. We will address each of the reviewer's concerns below.
> ### **Weaknesses 1 (Question 2)**
>
> Our theoretical framework grounded in Theorems 2 and 3 is designed to emphasize **local accuracy** in the absence of explicit differentiation, which guarantees that our method can work well at short intervals, empirically like 4 or 8 steps. As a result, we position our method between faster samplers and few-step diffusion distillation/training methods, as our method provides good trade-off between performance and efficiency. We provide additional experiments to show the stability and efficiency against other methods. Please also see the response to Reviewer v8s1 regrading Weakness 2.
>
> ### **Weaknesses 2**
>
> We agree with the reviewer that Lipschitz continuity is not generally guaranteed for a neural network in practice. However, we work in the context of diffusion models, where it is a necessary condition to apply foundational results like the Picard-Lindelöf theorem, which guarantees the existence and uniqueness of the ODE solution. This provides a rigorous framework for analyzing our algorithm's convergence, stability, and correctness, common to many works in this field. For example, the seminal work, consistency models [1] and consistency trajectory models [2], both include the Lipchitz condition in the Theorems. And the theoretical guarantee is under the worst case; in practice, the strong empirical stability of our method suggests that our training procedure implicitly guides the network to a well-behaved state, thus closely approximating the conditions required by the theory.
>
> For violations, i) if the function is not Lipschitz, the convergence is not guaranteed. However, according to the experiments, our method consistently trains models with great stability and performance, suggesting the function appears to be locally smooth in the relevant regions of the state space. ii) The Lipchitz constant is not small enough. In this situation, the small numerical error from one step could be amplified in subsequent steps. Considering this, we position our method between fast samplers and diffusion distillation/training methods, as a good trade-off between performance and efficiency.
>
> ### **Weaknesses 3 (Question 3)**
>
> First, we would like to correct an error in Table 7 of the appendix. The 4- and 8-step FIDs for STEI on CIFAR-10 should be **4.04** and **2.59**. We discovered that dropout was not correctly enabled in our previous experiments, contrary to the setting of 0.2 specified in Table 8.
>
> In Table 7, the difference happens mainly between estimating inner or end point (training/distillation achieves similar performance), which can be explained by three aspects.
> i) **Input correctness**. For inner-point variants, the input to the training network is the clean data $\boldsymbol{x}_t$. While in end-point variants, the input is the estimated $\\hat{\boldsymbol{x}}\_t$. Fitting a fixed, clean input distribution, is more efficient than fitting an input dependent of the model historical output. ii) **Error accumulation path**, i.e., the path from clean data to the prediction destination. In inner-point variants, the path is $\boldsymbol{x}_t\to \boldsymbol{x}_s$, while in end-point variants it is $\boldsymbol{x}_r\to \boldsymbol{x}_t \to \boldsymbol{x}_s$. Generally, the path in end-point variants is longer than that in inner-point variants. iii) **Model capacity**. As per Figure 1 and Algorithm 2, the end-point variants additionally require inversion, which costs part of the model capacity.
>
> ### **Weaknesses 4 (Question 4)**
>
> We thank the reviewer for pointing out this omission. We include the proofs for STEI and SDEE below. We will add these to the final manuscript. (\boldsymbol is omitted to save the character count.)
>
> Corollary 4. (STEI) Let $f_\theta(x_t,t,s)$ be a neural network, and let $v(x,t)$ be the true velocity field. Assume $v(x,t)$ is L-Lipschitz continuous in its first argument. If $\mathcal{L}\_{STEI}(\theta)$ reaches its minimum, then for any interval where $|s-t| \le h$ and $h < 3/L$, we have $f_{\theta}(x_{t},t,s)=f(x_{t},t,s)$.
>
> Proof. The loss $\mathcal{L}\_{STEI}(\theta)$ is defined as a sum of a secant term and a diffusion term: $$ \mathcal{L}\_{STEI}(\theta) = \mathbb{E}[||f_\theta(x_t,t,s)-f_{\theta^-}(\hat{x}\_r,r,r)||\_2^2] + \lambda\mathbb{E}[||f_\theta(x_\tau,\tau,\tau)-(\alpha_\tau'x_0+\sigma_\tau'z)||\_2^2],$$ where $\hat{x}\_r = x_t+(r-t)f_{\theta^-}(x_t,t,r)$. For $\mathcal{L}\_{STEI}(\theta)$ to reach the minimum, both two terms must reach the minimum. By Proposition 1 of our paper, the diffusion term ahieves the minimum if and only if $$ f_\theta(x_t, t, t) = \mathbb{E}(\alpha_t'x_0+\sigma_t'z|x_t) = v(x_t,t).$$ Subsitituting this result into the secant term, it becomes the one studied in the proof of Theorem 2. The remainder of the proof follows identically.
>
> Corollary 5. (SDEE) Let $f_\theta(x_t,t,s)$ be a neural network that is L-Lipschitz continuous in its first argument. For any fixed interval $[a, b]\subseteq[0,1]$ with $b-a$ sufficiently small, if $\mathcal{L}\_{SDEE}(\theta)$ reaches its minimum, then $f_\theta(x_t,t,s)=f(x_t,t,s)$.
>
> Proof. The SDEE loss is defined as: $$ \mathcal{L}\_{SDEE}(\theta) = \mathbb{E}\_{x_0,z,t,s,r\sim\mathcal{U}(t,s)}[||f_\theta(x_r+(t-r)f_{\theta^-}(x_r,r,t),t,s)-v(x_r,r)||\_2^2].$$ This loss function is identical to the intermediate objective $\mathcal{L}\_{*}(\theta)$ analyzed in the proof of Theorem 3. The proof, therefore, follows that of Theorem 3 directly.
>
> ### **Weakness 5 (Question 5)**
>
> Mathematically, uniform sampling and a non-uniform distribution rebalanced with importance sampling are equivalent. However, in the context of deep learning, the choice of sampling distribution corresponds to how we allocate the **training effort** across different time points. We apologize for not clarifying this motivation in the main text. Inspired by recent findings in the diffusion model literature (e.g., EDM [3], SD3 [4], FasterDiT [5]), where focusing training efforts on specific time intervals was shown to accelerate convergence, we decided to explore if a similar strategy would benefit our model. However, empirical results indicate that allocating training effort uniformly works best for our method.
>
> ### **Weakness 6**
>
> As stated in the main text below Eq. (9), “This approach is inspired by the diffusion objective that leads from Eq. (1) to Eq. (2).”, our use of Monte Carlo estimation for sampling data and time points **within one batch** is analogous to the standard practice in training diffusion models. We follow this manner and choose the same batch size (e.g., 256 for ImageNet, 512 for CIFAR-10) as these baseline models. The experimental results validate that it is sufficient for stable convergence. Regarding the analysis of convergence rates, to our knowledge, it is still an open research problem in the field of deep generative modeling. To directly address the reviewer's point on sample sizes and computational trade-offs, we conduct ablation study on the effect of batch size on performance. We train models with SDEI and EDM on CIFAR-10 with varying batch sizes and report the 4-step FID. The table below suggests that our method is robust to the batch size. We will add these results to the appendix of our paper.
> | Batch size | FID (same iteration) | FID (same epoch) |
> | :--- | :---: | :---: |
> | 64 | 3.62 | 3.24 |
> | 128 | 3.48 | 3.23 |
> | 256 | 3.33 | 3.22 |
> | 512 | 3.23 | 3.23 |
>
> ### **Weakness 7**
>
> We thank the reviewer for indicating the presentation issues. We will correct the typos. For Proposition 1, as stated in the appendix after the proof of Proposition 1, “The above is a basic result in literature of diffusion models [66].” We will also add the attribution and references to the main text as well.
>
> ### **Question 1**
>
> The additional complexity is only introduced to the **training phase**, i.e., the extra forward and backward passes as summarized in Table 1. Other minor computations, such as sampling time steps $s$ and $r$, are negligible. Practical running speed and memory cost can be referred to the response to Reviewer v8s1, regarding Weakness 2. Our method has almost no impact on the inference/sampling phase, as the only architectural modification is the injection of an additional time condition, which has a negligible computational cost. Therefore, it does not affect the claimed efficiency gains.
>
> ### **Question 6**
>
> Since few-step diffusion distillation/training methods typically do not modify the model architecture, the running time per NFE is the same as diffusion models. We list some results in the table below (the first is on CIFAR-10, and the second is on ImageNet-256), where we use one A100 GPU with a batchsize of 8.
>
> | Method | FID | NFE | Run time (ms) |
> | :--- | :---: | :---: | :---: |
> | EDM (Teacher) | 1.97 | 35 | 677.43 |
> | CD | 2.93 | 2 | 53.71 |
> | IMM | 1.98 | 2 | 54.82 |
> | SDEI (Ours) | 3.23 | 4 | 88.42 |
> | SDEI (Ours) | 2.14 | 10 | 216.76 |
>
> | Method | FID | NFE | Run time (ms) |
> | :--- | :---: | :---: | :---: |
> | SiT-XL/2 (Teacher) | 2.15 | 250($\times$2) | 16041.28 |
> | Shortcut Models | 7.80 | 4 | 238.27 |
> | IMM | 2.51 | 4($\times$2) | 359.98 |
> | IMM | 1.99 | 8($\times$2) | 624.48 |
> | STEI (Ours) | 2.27 | 4 | 240.05 |
> | STEI (Ours) | 1.96 | 8 | 369.51 |
>
> [1] Consistency models. Yang Song et al. ICML 2023
>
> [2] Consistency trajectory models: Learning probability flow ode trajectory of diffusion. Dongjun Kim et al. ICLR 2024
>
> [3] Elucidating the design space of diffusion-based generative models. Tero Karras et al. NeurIPS 2022
>
> [4] Scaling rectified flow transformers for high-resolution image synthesis. Patrick Esser et al. ICML 2024
>
> [5] FasterDiT: Towards faster diffusion transformers training without architecture modification. Jingfeng Yao et al. NeurIPS 2024

---

> > ### Comment · Reviewer_NLNA · 2025-08-05
> >
> > Thank you for your rebuttal. We found your perspective — particularly the use of secant and tangent views — inspiring and thought-provoking.
> >
> > > **Regarding Weakness 1 (Question 2)**
> >
> > Thank you for the clarification. We also reviewed your response to Reviewer v8s1 regarding Weakness 2. While your loss curves do appear more stable in the provided table, the baseline results (e.g., the first row) seem surprisingly poor. In our own experiments using this type of loss — along with techniques such as *Diffusion Finetuning and Tangent Warmup* (Section 4.2 of [1]) and mean flow adjustments to increase the tangent ratio (as in Table 1(a) of your cited paper) — we were able to achieve fairly stable training. **This raises some concern about whether the baselines in your comparison were intentionally under-tuned to emphasize the benefit of your approach.**
> >
> > Moreover, while your loss curves are indeed quite smooth, **we could not find a convincing theoretical explanation in Theorems 2 and 3 for such stability**. These theorems primarily describe what the network learns in idealized settings. Even assuming Lipschitz continuity, this alone cannot explain such a drastic reduction in loss variance. From our practical experience, what *does* reduce variance in MSE-type losses is importance sampling [2], but we did not see any mention of such techniques in the relevant lines (L187–200, L257–268).
> >
> > Based on both our understanding and experimentation, one possible reason for the observed stability is that the secant formulation (Equation 8) effectively averages over tangents — rather than the tangent-matching mechanism itself. However, even with this averaging, additional techniques are usually needed to achieve the kind of stability you reported. In practice, **the tangent-matching loss alone tends to be unstable.**
> >
> > > **Regarding Weakness 5 (Question 5)**
> >
> > As partially addressed above, we also experimented with various time schedules, including uniform sampling. We found the performance of uniform to be suboptimal — as you also acknowledged — and the variance across different sampling schemes to be quite high. Therefore, **we believe your method still lacks comprehensive ablation studies in this regard.**
> >
> > Additionally, your reported results in Table 2 and Table 3 are not particularly compelling, and I found it puzzling that Figure 2 highlights the 8-step performance rather than results for smaller step counts (e.g., 3-step). If your method struggles with lower step counts, it may not be ideal to feature 8-step results so prominently, as this may negatively affect the perceived practicality and competitiveness of your method.
> >
> > > **Overall Assessment**
> >
> > Overall, the core idea of this paper is interesting. However, the presentation feels somewhat rushed — the theoretical analysis is not profound, and the experimental support is not yet convincing. Based on these reasons, I tend to retain my score.

---

> > > ### Author Response · Authors · 2025-08-07
> > > **Response to Reviewer NLNA**
> > >
> > > We thank the reviewer for the valuable feedback. We will address the concerns as follows.
> > >
> > > > **Reviewer:**  In our own experiments using this type of loss — along with techniques such as Diffusion Finetuning and Tangent Warmup (Section 4.2 of [1]) and mean flow adjustments to increase the tangent ratio (as in Table 1(a) of your cited paper) — we were able to achieve fairly stable training.
> > >
> > > 1. Agreement and Highlighting the Core Issue
> > >
> > > We agree with you entirely: by employing advanced techniques, it is possible to achieve stable training for Consistency Models (CMs). In the experiment on target stability with EDM on CIFAR-10, we also observe that the training proceeds normally and the results are recorded under this condition (our setting is uniform sampling and pseudo-huber loss).
> > >
> > > However, our central claim is not about whether the training process fails or succeeds, but rather about the **intrinsic properties and inherent stability of the loss targets themselves**. We argue that the item $\\frac{d}{dt}\\boldsymbol{f}\_{\theta^-}(\\boldsymbol{x}\_t,t,s)$ in CM loss target has an intrinsically much larger variance than our proposed secant-based targets. This is a fundamental difference between the two approaches.
> > >
> > > 2. The Need for Tuning as Evidence for Instability
> > >
> > > You mentioned that achieving stability required a combination of several specialized techniques. We believe this observation **directly supports our central argument**. The very fact that CMs require significant efforts—such as model architectures, loss weighting schemes, and specialized warmup or finetuning strategies as seen in the sCM paper and your own experiments—serves as evidence for the underlying instability of its core loss objective. This extensive tuning process is essentially a set of countermeasures for this high variance.
> > >
> > > 3. The Simplicity and Robustness of Our Method
> > >
> > > In contrast, our proposed method is designed for simplicity and robustness, mirroring the design of standard diffusion models. It uses:
> > > * A simple time sampling strategy (following diffusion models).
> > > * Standard loss weighting (following diffusion models).
> > > * A simple Mean Squared Error (MSE) loss (following diffusion models).
> > > * A stable loss target, $-\\boldsymbol{x}_0 + \\boldsymbol{z}$ (the same as the training target of diffusion models) or $\\mathbb{E}(-\\boldsymbol{x}_0 + \\boldsymbol{z}|\\boldsymbol{x}_t)$ (similar to a trained diffusion model).
> > >
> > > The strong parallels between our method and standard diffusion models provide a powerful cue: if a standard diffusion model loss works well on a given dataset, it is highly likely our secant losses will too.
> > >
> > > 4. New Empirical Evidence Under the Reviewer's Setting
> > >
> > > To provide direct evidence for this, we conducted a new experiment that incorporates the very techniques you mentioned for sCM (diffusion finetuning, and tangent normalization, per the sCM paper; we find that tangent warmup sometimes has a negative effect to the training process). The results are presented below.

---

> > > > ### Author Response · Authors · 2025-08-07
> > > >
> > > > | Loss (Inspected item)/Training iteration | Time sampling | 100 | 200 | 300 | 400 | 500 | 600 | 700 | 800 | 900 | 1000 |
> > > > | :--- | :---: | :---: | :---: | :---: | :--- | :---: | :---: | :--- | :---: | :---: | :---: |
> > > > | sCT ($\\frac{d}{dt}\\boldsymbol{f}\_{\theta^-}(\\boldsymbol{x}\_t,t))$) | sCM-style | 58.5817 | 39.5814 | 40.4797 | 41.2633 | 44.0948 | 34.2345 | 64.7170 | 72.9639 | 60.5475 | 59.0707 |
> > > > | sCT ($\\frac{d}{dt}\\boldsymbol{f}\_{\theta^-}(\\boldsymbol{x}\_t,t))$) | uniform   | 73.6489 | 68.5763 | 55.9710| 109.4940 | 101.6546| 72.0015 | 61.5663 | 58.0534 | 137.1868| 42.4191 |
> > > > | sCD ($\\frac{d}{dt}\\boldsymbol{f}\_{\theta^-}(\\boldsymbol{x}\_t,t))$) | sCM-style | 27.2778 | 45.9737 | 28.0768 | 39.7054 | 65.7447 | 14.7490 | 37.4557 | 30.7352 | 28.8754 | 35.4749 |
> > > > | sCD ($\\frac{d}{dt}\\boldsymbol{f}\_{\theta^-}(\\boldsymbol{x}\_t,t))$) | uniform   | 39.2204 | 98.1394 | 105.7974| 43.3716 | 73.6825 | 76.5498 | 47.6203 | 41.4083 | 115.6248| 92.6327 |
> > > > | SDEI ($\\boldsymbol{v}(\\boldsymbol{x}_r,r)$) | sCM-style | 5.4268 | 4.6310 | 4.4608 | 5.1106 | 5.1107 | 5.3483 | 3.3009 | 5.0257 | 4.8428 | 4.4073 |
> > > > | SDEI ($\\boldsymbol{v}(\\boldsymbol{x}_r,r)$) | uniform   | 4.4440 | 4.0587 | 5.0219 | 5.0895 | 4.5227 | 4.3183 | 3.8064 | 3.8358 | 4.3588 | 4.8104 |
> > > > | STEI ($\\boldsymbol{f}_{\theta^-}(\\boldsymbol{x}_r,r,r)$) | sCM-style | 4.9498 | 4.3028 | 3.5500 | 4.2938 | 4.2709 | 4.1400 | 3.7830 | 4.0299 | 3.6475 | 4.0111 |
> > > > | STEI ($\\boldsymbol{f}_{\theta^-}(\\boldsymbol{x}_r,r,r)$) | uniform   | 3.9716 | 4.2328 | 3.8777 | 4.6646 | 3.9758 | 5.0697 | 5.0835 | 3.8016 | 4.0628 | 4.8425 |
> > > > | SDEE ($\\boldsymbol{v}(\\boldsymbol{x}_r,r)$) | sCM-style | 3.7461 | 4.1117 | 3.7013 | 3.6001 | 4.7936 | 4.0002 | 4.2830 | 4.3777 | 4.0948 | 2.9608 |
> > > > | SDEE ($\\boldsymbol{v}(\\boldsymbol{x}_r,r)$) | uniform   | 3.8232 | 3.7177 | 3.3039 | 3.5467 | 3.1078 | 3.9357 | 3.7119 | 5.3439 | 3.2826 | 4.2481 |
> > > > | STEE ($-\\text{sin}(t)\\boldsymbol{x}_0+\\text{cos}(t)\\boldsymbol{z}$) | sCM-style | 3.2402 | 3.0077 | 2.7767 | 3.7332 | 2.9161 | 3.6943 | 3.8329 | 3.2464 | 5.2368 | 3.3445 |
> > > > | STEE ($-\\text{sin}(t)\\boldsymbol{x}_0+\\text{cos}(t)\\boldsymbol{z}$) | uniform   | 4.9020 | 3.1031 | 3.7018 | 3.0115 | 3.0127 | 2.9246 | 3.0980 | 2.4086 | 2.7033 | 2.4870 |
> > > >
> > > > As the table shows, even when CMs are augmented with these stabilization techniques, our loss targets have a significantly and consistently smaller variance than those of CMs. We also observed that our method remains less sensitive to the time sampling strategy. This empirically confirms the inherent stability advantage of our approach.

---

> ### Comment · Area_Chair_vUT8 · 2025-08-04
> **Action Required: Author–Reviewer Discussion Closing Soon**
>
> Dear Reviewer,
>
>
>
> This is a gentle reminder that the **Author–Reviewer Discussion** phase ends within just three days (by **August 6**). Please take a moment to read the authors’ rebuttal thoroughly and engage in the discussion. Ideally, every reviewer will respond so the authors know their rebuttal has been seen and considered.
>
>
>
> Thank you for your prompt participation!
>
>
>
> Best regards,
>
>
>
> AC

---

> > ### Comment · Area_Chair_vUT8 · 2025-08-05
> > **Action Required: Author–Reviewer Discussion Closing Soon**
> >
> > Dear Reviewer,
> >
> > A gentle reminder that the extended Author–Reviewer Discussion phase ends on **August 8 (AoE)**.
> >
> > Please read the authors’ rebuttal and participate in the discussion **ASAP**. Regardless of whether your concerns have been addressed, kindly communicate:
> >
> > - If resolved — please acknowledge.
> >
> > - If unresolved — please specify what remains.
> >
> > The “Mandatory Acknowledgement” should only be submitted **after**:
> >
> > - Reading the rebuttal,
> >
> > - Engaging in author discussion,
> >
> > - Completing the "Final Justification" (and updating your rating).
> >
> > **As per policy, I may flag any missing or unresponsive reviews and deactivate them once additional reviewer feedback has been posted.**
> >
> > Thank you for your timely and thoughtful contributions.
> >
> >
> >
> >
> > Best regards,
> >
> > AC

---

> ### Author Response · Authors · 2025-08-07
>
> > **Reviewer:**  This raises some concern about whether the baselines in your comparison were intentionally under-tuned to emphasize the benefit of your approach.
>
> Thank you for raising this critical point. We take this concern very seriously and want to assure you that the baselines were tuned to ensure they can be trained normally with EDM on CIFAR-10. And we observe the item variantion under this condition. Specifically, we apply the pseudo-huber loss specified in iCT paper, which has the similar effect to the tangent normalization technique in sCM paper, and under which the loss can decrease normally. We also let it grow from $s=t$ in the first several iterations. And we keep other parts the same, for example the time sampling strategy is uniform sampling, which do not affect the overall training. In the following, we will add more experimental results ablating the time sampling strategy for a comprehensive comparison. To provide full transparency, we will upload the core source code below.
>
> > **Reviewer:**  Moreover, while your loss curves are indeed quite smooth, we could not find a convincing theoretical explanation in Theorems 2 and 3 for such stability. These theorems primarily describe what the network learns in idealized settings. Even assuming Lipschitz continuity, this alone cannot explain such a drastic reduction in loss variance. From our practical experience, what does reduce variance in MSE-type losses is importance sampling [2], but we did not see any mention of such techniques in the relevant lines (L187–200, L257–268).
>
> Theorems 2 and 3 work to establish the theoretical feasibility and local accuracy of our method, rather than to provide a direct proof of its variance reduction. You are also correct that importance sampling is a well-known technique for reducing loss variance, but we do not use, and it is not a focus in our paper.
>
> The primary source of the drastic stability improvement you observed is not a sophisticated sampling strategy, but rather the **inherent stability of our proposed loss target**. Our targets, such as the velocity estimation $-\\boldsymbol{x}\_0 + \\boldsymbol{z}$ (identical to the diffusion target) or its conditional expectation $\\mathbb{E}(-\\boldsymbol{x}\_0 + \\boldsymbol{z}|\\boldsymbol{x}\_t)$ are analogous to those in highly stable, standard diffusion models. Our training process inherits this stability directly. In contrast, the loss target in CMs, $\\frac{d}{dt}\\boldsymbol{f}\_{\theta^-}(\\boldsymbol{x}\_t,t,s))$, relies on a numerical derivative. This is computed via a finite difference between two noisy model evaluations, which is then divided by a small time step. This process inherently amplifies noise and leads to instability, especially early or middle in training when the model $\boldsymbol{f}_{\theta^-}$ is not yet accurate.
>
> In the final version, we will:
>  1.  Revise the main text (around L187–200) to explicitly articulate that the stability stems from the nature of the loss target, better connecting our theory to the empirical results.
>  2.  Add a discussion to the related work section on variance reduction techniques, including importance sampling, to better contextualize our approach and its advantages.

---

> > ### Author Response · Authors · 2025-08-07
> >
> > > **Reviewer:**  one possible reason for the observed stability is that the secant formulation (Equation 8) effectively averages over tangents — rather than the tangent-matching mechanism itself.
> >
> > Thank you for this insightful hypothesis. We conduct experiments where we both use the secant formulation (Eq. (8)) as the model parametrization, and use diffusion pretraining, to test sCM losses. As the results in the table show, the high variance of the CM loss target persists even when using the secant parameterization. This suggests that the secant parameterization alone may not be the primary source of stability, and that the nature of the loss target itself plays a more critical role. We believe this new experiment helps to clarify this point.
> >
> > | Loss (Inspected item)/Training iteration | Time sampling | 100 | 200 | 300 | 400 | 500 | 600 | 700 | 800 | 900 | 1000 |
> > | :--- | :---: | :---: | :---: | :---: | :--- | :---: | :---: | :--- | :---: | :---: | :---: |
> > | sCT+Eq.(8) ($\\frac{d}{dt}\\boldsymbol{f}\_{\theta^-}(\\boldsymbol{x}\_t,t,s))$) | sCM-style | 40.4587 | 53.3100 | 53.2495 | 53.3127 | 38.7443 | 60.6008 | 49.8528 | 51.7059 | 28.1779 | 24.6173 |
> > | sCT+Eq.(8) ($\\frac{d}{dt}\\boldsymbol{f}\_{\theta^-}(\\boldsymbol{x}\_t,t,s))$) | uniform   | 130.1099| 59.2177 | 57.5015| 47.8698 | 125.2506| 47.0202 | 132.4226 | 155.2914 | 55.7508| 159.9047 |
> > | sCD+Eq.(8) ($\\frac{d}{dt}\\boldsymbol{f}\_{\theta^-}(\\boldsymbol{x}\_t,t,s))$) | sCM-style | 46.6887 | 42.8773 | 47.7213 | 45.3440 | 39.5877 | 45.8336 | 30.8990 | 75.8576 | 41.7149 | 38.7751 |
> > | sCD+Eq.(8) ($\\frac{d}{dt}\\boldsymbol{f}\_{\theta^-}(\\boldsymbol{x}\_t,t,s))$) | uniform   | 59.2249 | 23.0211 | 44.3810| 136.6149 | 56.1703 | 85.9707 | 81.1506 | 62.0049 | 39.5047| 84.7024 |
> >
> > > **Reviewer:**  However, even with this averaging, additional techniques are usually needed to achieve the kind of stability you reported.
> >
> > We can confirm that the stability shown in our results was achieved without any extra techniques beyond what is described in the paper. We believe this highlights the inherent robustness of our proposed method. (It can also be seen in the attached code implementation.)
> >
> > > **Reviewer:**  In practice, the tangent-matching loss alone tends to be unstable.
> >
> > We respectfully hold a different perspective on this point. We believe our specific formulation, which uses a stable, diffusion-like target, effectively mitigates these instability issues. Our empirical results in the tables support this view.
> >
> > > **Reviewer:**  we also experimented with various time schedules, including uniform sampling. We found the performance of uniform to be suboptimal — as you also acknowledged — and the variance across different sampling schemes to be quite high. Therefore, we believe your method still lacks comprehensive ablation studies in this regard.
> >
> > Our rationale for focusing the ablation on the $r$ is that, to be consistent with diffusion models, we keep the settings related to $t$ and $s$ align the settings of $t$ of diffusion models, due to the similar status between $t$ and $s$ in our models and $t$ in diffusion models. Thus, we only study the extra $r$, which has no similar counterpart in diffusion models. For the sampling of $r$, as stated in the rebuttal, we would like to study how to distribute the training effort. According to the results in Section 5.3 Ablations, we compare four variants as shown in Figure 6, and find that the four performs similarly, and uniform sampling works the best. We acknowledge that a more comprehensive study including different time sampling schemes would strengthen the paper. Due to time and resource constraints during the rebuttal period, we were unable to complete an exhaustive sweep. However, based on your valuable feedback, we will commit to adding a more thorough ablation study on time sampling to the final version of the paper.
> >
> > > **Reviewer:**  Additionally, your reported results in Table 2 and Table 3 are not particularly compelling, and I found it puzzling that Figure 2 highlights the 8-step performance rather than results for smaller step counts (e.g., 3-step). If your method struggles with lower step counts, it may not be ideal to feature 8-step results so prominently, as this may negatively affect the perceived practicality and competitiveness of your method.
> >
> > Thank you for pointing this out. Our primary goal was to find a good trade-off between performance and efficiency, which is why we highlighted the strong results at 8 steps. You are right that performance at fewer steps is also critical for evaluating practicality. We will revise the figure and the main text to present a more balanced view across different step counts to better reflect these trade-offs.

---

> > > ### Author Response · Authors · 2025-08-07
> > >
> > > ### **Implementation Details**
> > >
> > > 1. Reproduction of sCM loss:
> > >
> > > ```
> > > @persistence.persistent_class
> > > class SCMLoss:
> > >     def __init__(self, P_mean=-1.2, P_std=1.2, sigma_data=0.5):
> > >         self.P_mean = P_mean
> > >         self.P_std = P_std
> > >         self.sigma_data = sigma_data
> > >         self.tmp_tick = -1
> > >
> > >     def __call__(self, net, teacher, images, labels=None, augment_pipe=None, cur_tick=None):
> > >         t0 = 0.001
> > >         tT = math.pi / 2 - 0.00625
> > >
> > >         type = 'training'
> > >         # type = 'distillation'
> > >
> > >         sampling_type = 'scm'
> > >         # sampling_type = 'uniform'
> > >
> > >         if sampling_type == 'scm':
> > >             rnd_normal = torch.randn([images.shape[0], 1, 1, 1], device=images.device)
> > >             sigma = (rnd_normal * 1.4 - 1.0).exp() # according to the settings of sCM
> > >             t = torch.atan(sigma / self.sigma_data)
> > >             t = torch.clamp(t, min=t0, max=tT)
> > >         elif sampling_type == 'uniform':
> > >             t = torch.rand([images.shape[0], 1, 1, 1], device=images.device)
> > >             t = t0 + t * (tT - t0)
> > >
> > >         y, augment_labels = augment_pipe(images) if augment_pipe is not None else (images, None)
> > >         z = torch.randn_like(y) * self.sigma_data
> > >
> > >         xt = torch.cos(t) * y + torch.sin(t) * z
> > >
> > >         if type == 'distillation':
> > >             with torch.no_grad():
> > >                 ut = teacher(xt, t, labels, augment_labels=augment_labels)
> > >         else:
> > >             assert type == 'training'
> > >             ut = -torch.sin(t) * y + torch.cos(t) * z
> > >
> > >         diff_type = 'continous'
> > >         # diff_type = 'discrete'
> > >
> > >         # loss_type = 'l2' # the plain l2 loss
> > >         # loss_type = 'pseudo-huber' # the loss used in iCT and eCT
> > >         loss_type = 'tannorm' # the loss used in sCM
> > >         tan_warmup = True
> > >         loss_weight = 1
> > >
> > >         if diff_type == 'continous':
> > >             unwrapped_net = net.module if hasattr(net, 'module') else net
> > >
> > >             def model_wrapper(x_input, t_input):
> > >                 output = unwrapped_net(x_input, t_input, labels, augment_labels=augment_labels)
> > >                 return output
> > >             f_ts, diff = torch.func.jvp(model_wrapper, (xt, t), (ut, torch.ones_like(t)))
> > >         else:
> > >             assert diff_type == 'discrete'
> > >             dt = 5e-3
> > >             r = (t - dt).clamp(min=t0)
> > >             xr = xt + (r - t) * ut
> > >             rng_state = torch.cuda.get_rng_state()
> > >             f_ts = net(xt, t, labels, augment_labels=augment_labels)
> > >             with torch.no_grad():
> > >                 torch.cuda.set_rng_state(rng_state)
> > >                 f_rs = net(xr, r, labels, augment_labels=augment_labels)
> > >                 diff = (f_rs - f_ts) / dt
> > >
> > >         if loss_type == 'tannorm':
> > >             with torch.no_grad():
> > >                 if tan_warmup:
> > >                     gamma = cur_tick / 100 if cur_tick < 100 else 1 # tannorm for the first 10k iters
> > >                 else:
> > >                     gamma = 1
> > >                 dfdt = -torch.cos(t) * (f_ts - ut) - gamma * torch.sin(t) * (xt + diff)
> > >                 dfdt = dfdt * torch.cos(t) # the weight specified in sCM
> > >                 norm = torch.linalg.vector_norm(dfdt, dim=(1, 2, 3), keepdim=True) + 0.1
> > >                 dfdt = dfdt / norm
> > >
> > >             loss = (f_ts - f_ts.detach() - loss_weight * dfdt) ** 2
> > >
> > >         elif loss_type == 'l2':
> > >             with torch.no_grad():
> > >                 target = ut - (t - 0) * diff
> > >             loss = loss_weight * ((f_ts - target)) ** 2
> > >
> > >         elif loss_type == 'pseudo-huber':
> > >             with torch.no_grad():
> > >                 target = ut - (t - 0) * diff
> > >             loss = ((f_ts - target)) ** 2
> > >             loss = torch.sum(loss.reshape(loss.shape[0], -1), dim=-1)
> > >             loss = loss_weight * torch.sqrt(loss)
> > >         else:
> > >             raise NotImplementedError
> > >
> > >         if self.tmp_tick != cur_tick:
> > >             n = torch.linalg.vector_norm(diff, dim=(1, 2, 3))
> > >             print(f"norm: mean: {n.mean().item():.4f}, std: {n.std().item():.4f}, max: {n.max().item():.4f}")
> > >             self.tmp_tick = cur_tick
> > >
> > >         return loss
> > > ```

---

> > > > ### Author Response · Authors · 2025-08-07
> > > >
> > > > 2. Implementation of our losses:
> > > >
> > > > i) SDEI loss:
> > > >
> > > > ```
> > > > @persistence.persistent_class
> > > > class SDEILoss:
> > > >     def __init__(self, P_mean=-1.2, P_std=1.2, sigma_data=0.5):
> > > >         self.P_mean = P_mean
> > > >         self.P_std = P_std
> > > >         self.sigma_data = sigma_data
> > > >         self.tmp_tick = -1
> > > >
> > > >     def __call__(self, net, teacher, images, labels=None, augment_pipe=None, cur_tick=None):
> > > >         t0 = 0.001
> > > >         tT = math.pi / 2 - 0.00625
> > > >         # sampling_type = 'scm'
> > > >         sampling_type = 'uniform'
> > > >
> > > >         if sampling_type == 'scm':
> > > >             rnd_normal = torch.randn([images.shape[0], 1, 1, 1], device=images.device)
> > > >             sigma = (rnd_normal * 1.4 - 1.0).exp() # according to the settings of sCM
> > > >             u = torch.atan(sigma / self.sigma_data)
> > > >             t = torch.clamp(u, min=t0, max=tT)
> > > >             rnd_normal = torch.randn([images.shape[0], 1, 1, 1], device=images.device)
> > > >             sigma = (rnd_normal * 1.4 - 1.0).exp() # according to the settings of sCM
> > > >             u = torch.atan(sigma / self.sigma_data)
> > > >             s = torch.clamp(u, min=t0, max=tT)
> > > >             s, t = torch.min(s, t), torch.max(s, t)
> > > >         elif sampling_type == 'uniform':
> > > >             u = torch.rand([images.shape[0], 1, 1, 1], device=images.device)
> > > >             t = t0 + u * (tT - t0)
> > > >             u = torch.rand([images.shape[0], 1, 1, 1], device=images.device)
> > > >             s = t0 + u * (t - t0)
> > > >         else:
> > > >             raise NotImplementedError
> > > >
> > > >         r = t + torch.rand([images.shape[0], 1, 1, 1], device=images.device) * (s - t)
> > > >
> > > >         y, augment_labels = augment_pipe(images) if augment_pipe is not None else (images, None)
> > > >
> > > >         z = torch.randn_like(y) * self.sigma_data
> > > >
> > > >         xt = torch.cos(t) * y + torch.sin(t) * z
> > > >
> > > >         rng_state = torch.cuda.get_rng_state()
> > > >         secant_ts = net(xt, t, s, labels, augment_labels=augment_labels, return_logvar=False)
> > > >
> > > >         with torch.no_grad():
> > > >             torch.cuda.set_rng_state(rng_state)
> > > >             xr_hat = xt + (r - t) * net(xt, t, r, labels, augment_labels=augment_labels)
> > > >             vr_hat = teacher(xr_hat, r, labels, augment_labels=augment_labels)
> > > >
> > > >         loss = (secant_ts - vr_hat) ** 2
> > > >
> > > >         if self.tmp_tick != cur_tick:
> > > >             n = torch.linalg.vector_norm(vr_hat, dim=(1, 2, 3))
> > > >             print(f"norm: mean: {n.mean().item():.4f}, std: {n.std().item():.4f}, max: {n.max().item():.4f}")
> > > >             self.tmp_tick = cur_tick
> > > >
> > > >         return loss
> > > > ```

---

> > > > > ### Author Response · Authors · 2025-08-07
> > > > >
> > > > > ii) STEI loss:
> > > > >
> > > > > ```
> > > > > @persistence.persistent_class
> > > > > class STEILoss:
> > > > >     def __init__(self, P_mean=-1.2, P_std=1.2, sigma_data=0.5):
> > > > >         self.P_mean = P_mean
> > > > >         self.P_std = P_std
> > > > >         self.sigma_data = sigma_data
> > > > >         self.tmp_tick = -1
> > > > >
> > > > >     def __call__(self, net, teacher, images, labels=None, augment_pipe=None, cur_tick=None):
> > > > >         t0 = 0.001
> > > > >         tT = math.pi / 2 - 0.00625
> > > > >         # sampling_type = 'scm'
> > > > >         sampling_type = 'uniform'
> > > > >
> > > > >         if sampling_type == 'scm':
> > > > >             rnd_normal = torch.randn([images.shape[0], 1, 1, 1], device=images.device)
> > > > >             sigma = (rnd_normal * 1.4 - 1.0).exp() # according to the settings of sCM
> > > > >             u = torch.atan(sigma / self.sigma_data)
> > > > >             t = torch.clamp(u, min=t0, max=tT)
> > > > >             rnd_normal = torch.randn([images.shape[0], 1, 1, 1], device=images.device)
> > > > >             sigma = (rnd_normal * 1.4 - 1.0).exp() # according to the settings of sCM
> > > > >             u = torch.atan(sigma / self.sigma_data)
> > > > >             s = torch.clamp(u, min=t0, max=tT)
> > > > >             s, t = torch.min(s, t), torch.max(s, t)
> > > > >         elif sampling_type == 'uniform':
> > > > >             u = torch.rand([images.shape[0], 1, 1, 1], device=images.device)
> > > > >             t = t0 + u * (tT - t0)
> > > > >             u = torch.rand([images.shape[0], 1, 1, 1], device=images.device)
> > > > >             s = t0 + u * (t - t0)
> > > > >         else:
> > > > >             raise NotImplementedError
> > > > >
> > > > >         r = t + torch.rand([images.shape[0], 1, 1, 1], device=images.device) * (s - t)
> > > > >
> > > > >         y, augment_labels = augment_pipe(images) if augment_pipe is not None else (images, None)
> > > > >
> > > > >         z = torch.randn_like(y) * self.sigma_data
> > > > >
> > > > >         xt = torch.cos(t) * y + torch.sin(t) * z
> > > > >
> > > > >         rnd_normal = torch.randn([images.shape[0], 1, 1, 1], device=images.device)
> > > > >         sigma = (rnd_normal * 1.4 - 1.0).exp()
> > > > >         tau = torch.atan(sigma / self.sigma_data)
> > > > >         xtau = torch.cos(tau) * y + torch.sin(tau) * z
> > > > >         utau = -torch.sin(tau) * y + torch.cos(tau) * z
> > > > >
> > > > >         rng_state = torch.cuda.get_rng_state()
> > > > >         tangent_tau = net(xtau, tau, tau, labels, augment_labels=augment_labels, return_logvar=False)
> > > > >         torch.cuda.set_rng_state(rng_state)
> > > > >         secant_ts = net(xt, t, s, labels, augment_labels=augment_labels, return_logvar=False)
> > > > >
> > > > >         with torch.no_grad():
> > > > >             torch.cuda.set_rng_state(rng_state)
> > > > >             xr_hat = xt + (r - t) * net(xt, t, r, labels, augment_labels=augment_labels)
> > > > >             torch.cuda.set_rng_state(rng_state)
> > > > >             vr_hat = net(xr_hat, r, r, labels, augment_labels=augment_labels)
> > > > >         loss = (secant_ts - vr_hat) ** 2 + (tangent_tau - utau) ** 2
> > > > >
> > > > >         if self.tmp_tick != cur_tick:
> > > > >             n = torch.linalg.vector_norm(vr_hat, dim=(1, 2, 3))
> > > > >             print(f"norm: mean: {n.mean().item():.4f}, std: {n.std().item():.4f}, max: {n.max().item():.4f}")
> > > > >             self.tmp_tick = cur_tick
> > > > >
> > > > >         return loss
> > > > > ```

---

> > > > > > ### Author Response · Authors · 2025-08-07
> > > > > >
> > > > > > iii) SDEE loss:
> > > > > >
> > > > > > ```
> > > > > > @persistence.persistent_class
> > > > > > class SDEELoss:
> > > > > >     def __init__(self, P_mean=-1.2, P_std=1.2, sigma_data=0.5):
> > > > > >         self.P_mean = P_mean
> > > > > >         self.P_std = P_std
> > > > > >         self.sigma_data = sigma_data
> > > > > >         self.tmp_tick = -1
> > > > > >
> > > > > >     def __call__(self, net, teacher, images, labels=None, augment_pipe=None, cur_tick=None):
> > > > > >         t0 = 0.001
> > > > > >         tT = math.pi / 2 - 0.00625
> > > > > >         sampling_type = 'scm'
> > > > > >         # sampling_type = 'uniform'
> > > > > >
> > > > > >         if sampling_type == 'scm':
> > > > > >             rnd_normal = torch.randn([images.shape[0], 1, 1, 1], device=images.device)
> > > > > >             sigma = (rnd_normal * 1.4 - 1.0).exp() # according to the settings of sCM
> > > > > >             u = torch.atan(sigma / self.sigma_data)
> > > > > >             t = torch.clamp(u, min=t0, max=tT)
> > > > > >             rnd_normal = torch.randn([images.shape[0], 1, 1, 1], device=images.device)
> > > > > >             sigma = (rnd_normal * 1.4 - 1.0).exp() # according to the settings of sCM
> > > > > >             u = torch.atan(sigma / self.sigma_data)
> > > > > >             s = torch.clamp(u, min=t0, max=tT)
> > > > > >         elif sampling_type == 'uniform':
> > > > > >             u = torch.rand([images.shape[0], 1, 1, 1], device=images.device)
> > > > > >             t = t0 + u * (tT - t0)
> > > > > >             u = torch.rand([images.shape[0], 1, 1, 1], device=images.device)
> > > > > >             s = t0 + u * (tT - t0)
> > > > > >         else:
> > > > > >             raise NotImplementedError
> > > > > >
> > > > > >         r = t + torch.rand([images.shape[0], 1, 1, 1], device=images.device) * (s - t)
> > > > > >
> > > > > >         y, augment_labels = augment_pipe(images) if augment_pipe is not None else (images, None)
> > > > > >
> > > > > >         z = torch.randn_like(y) * self.sigma_data
> > > > > >
> > > > > >         xr = torch.cos(r) * y + torch.sin(r) * z
> > > > > >
> > > > > >         with torch.no_grad():
> > > > > >
> > > > > >             vr_hat = teacher(xr, r, labels, augment_labels=augment_labels)
> > > > > >
> > > > > >             rng_state = torch.cuda.get_rng_state()
> > > > > >             xt_hat = xr + (t - r) * net(xr, r, t, labels, augment_labels=augment_labels)
> > > > > >
> > > > > >         torch.cuda.set_rng_state(rng_state)
> > > > > >         secant_ts = net(xt_hat, t, s, labels, augment_labels=augment_labels)
> > > > > >         loss = (secant_ts - vr_hat) ** 2
> > > > > >
> > > > > >         if self.tmp_tick != cur_tick:
> > > > > >             n = torch.linalg.vector_norm(vr_hat, dim=(1, 2, 3))
> > > > > >             print(f"norm: mean: {n.mean().item():.4f}, std: {n.std().item():.4f}, max: {n.max().item():.4f}")
> > > > > >             self.tmp_tick = cur_tick
> > > > > >
> > > > > >         return loss
> > > > > > ```
> > > > > >
> > > > > > iv) STEE loss:
> > > > > >
> > > > > > ```
> > > > > > @persistence.persistent_class
> > > > > > class STEELoss:
> > > > > >     def __init__(self, P_mean=-1.2, P_std=1.2, sigma_data=0.5):
> > > > > >         self.P_mean = P_mean
> > > > > >         self.P_std = P_std
> > > > > >         self.sigma_data = sigma_data
> > > > > >         self.tmp_tick = -1
> > > > > >
> > > > > >     def __call__(self, net, teacher, images, labels=None, augment_pipe=None, cur_tick=None):
> > > > > >         t0 = 0.001
> > > > > >         tT = math.pi / 2 - 0.00625
> > > > > >         sampling_type = 'scm'
> > > > > >         # sampling_type = 'uniform'
> > > > > >
> > > > > >         if sampling_type == 'scm':
> > > > > >             rnd_normal = torch.randn([images.shape[0], 1, 1, 1], device=images.device)
> > > > > >             sigma = (rnd_normal * 1.4 - 1.0).exp() # according to the settings of sCM
> > > > > >             u = torch.atan(sigma / self.sigma_data)
> > > > > >             t = torch.clamp(u, min=t0, max=tT)
> > > > > >             rnd_normal = torch.randn([images.shape[0], 1, 1, 1], device=images.device)
> > > > > >             sigma = (rnd_normal * 1.4 - 1.0).exp() # according to the settings of sCM
> > > > > >             u = torch.atan(sigma / self.sigma_data)
> > > > > >             s = torch.clamp(u, min=t0, max=tT)
> > > > > >         elif sampling_type == 'uniform':
> > > > > >             u = torch.rand([images.shape[0], 1, 1, 1], device=images.device)
> > > > > >             t = t0 + u * (tT - t0)
> > > > > >             u = torch.rand([images.shape[0], 1, 1, 1], device=images.device)
> > > > > >             s = t0 + u * (tT - t0)
> > > > > >         else:
> > > > > >             raise NotImplementedError
> > > > > >
> > > > > >         r = t + torch.rand([images.shape[0], 1, 1, 1], device=images.device) * (s - t)
> > > > > >
> > > > > >         y, augment_labels = augment_pipe(images) if augment_pipe is not None else (images, None)
> > > > > >
> > > > > >         z = torch.randn_like(y) * self.sigma_data
> > > > > >
> > > > > >         xr = torch.cos(r) * y + torch.sin(r) * z
> > > > > >
> > > > > >         with torch.no_grad():
> > > > > >
> > > > > >             rng_state = torch.cuda.get_rng_state()
> > > > > >             xt_hat = xr + (t - r) * net(xr, r, t, labels, augment_labels=augment_labels)
> > > > > >             vr_hat = -torch.sin(r) * y + torch.cos(r) * z
> > > > > >
> > > > > >         torch.cuda.set_rng_state(rng_state)
> > > > > >         secant_ts = net(xt_hat, t, s, labels, augment_labels=augment_labels)
> > > > > >         loss = (secant_ts - vr_hat) ** 2
> > > > > >
> > > > > >         if self.tmp_tick != cur_tick:
> > > > > >             n = torch.linalg.vector_norm(vr_hat, dim=(1, 2, 3))
> > > > > >             print(f"norm: mean: {n.mean().item():.4f}, std: {n.std().item():.4f}, max: {n.max().item():.4f}")
> > > > > >             self.tmp_tick = cur_tick
> > > > > >
> > > > > >         return loss
> > > > > > ```

---

> > > > > > > ### Author Response · Authors · 2025-08-07
> > > > > > >
> > > > > > > Once again, we thank you for your valuable time and constructive feedback. We hope our response has addressed your concerns. We would be happy to answer any further questions you may have.

---

> > > > > > > > ### Comment · Reviewer_NLNA · 2025-08-07
> > > > > > > >
> > > > > > > > Thank you very much for your detailed response. The three advantages you highlighted — (1) a simple time sampling strategy (following diffusion models), (2) standard loss weighting (following diffusion models), and (3) a simple Mean Squared Error (MSE) loss (following diffusion models) — are points I completely agree with, and they align well with my own observations. These are also the reasons why I believe your method is better than Meanflow.
> > > > > > > >
> > > > > > > > That said, compared to Meanflow, I found your method section somewhat harder to follow, especially the parts related to the loss function. Since the core of your method lies in how the secant is learned — and the loss function plays a crucial role — I think this part deserves to be presented more clearly. Currently, you include two formulations of the objective function in the main text, which may be redundant; including just one (others to appendix), with a more focused explanation, could be more effective.
> > > > > > > >
> > > > > > > > More importantly, I think the method section could better highlight the main advantages of your approach compared to prior work — namely, its simplicity and stability. These are the aspects that truly make your contribution valuable. Additionally, the paper currently lacks an explanation for why your method tends to be more stable during training, as your experiments convincingly demonstrate. This is a point I mentioned in my earlier comments. In fact, I have both theoretically and empirically validated this kind of stability in similar settings, so I trust and appreciate your experimental findings.
> > > > > > > >
> > > > > > > > Overall, I recognize and appreciate your contribution — proposing a derivative-free, more stable loss function, which I believe is a meaningful and practical advancement in the field. However, I feel this core insight could have been emphasized more clearly throughout the paper.
> > > > > > > >
> > > > > > > > That said, since you have addressed the key concerns I raised, and given that I’ve personally verified and agree with the value of your contributions, I’ve decided to raise my score for your submission. Nice work.

---

> > > > > > > > > ### Author Response · Authors · 2025-08-07
> > > > > > > > >
> > > > > > > > > Thank you so much for your thoughtful and incredibly constructive comments. We are sincerely grateful for the time and expertise you dedicated throughout this entire review process. Your feedback has been instrumental in helping us sharpen our thinking and improve the paper. We are especially thankful for your decision to raise your score.
> > > > > > > > > We have taken all of your final suggestions to heart. You are absolutely right that while the core contribution is valuable, its presentation can be significantly improved.
> > > > > > > > >
> > > > > > > > > In the final version of our paper, we will focus on:
> > > > > > > > >
> > > > > > > > > 1. Clarifying the Method Section: We will revise it to be more focused and easier to follow, especially the part concerning the loss function, and will adopt your suggestion of moving one formulation to the appendix.
> > > > > > > > >
> > > > > > > > > 2. Highlighting the Core Advantages: We will explicitly emphasize the benefits of simplicity and stability in comparison to prior work.
> > > > > > > > >
> > > > > > > > > 3. Explaining the reason of Stability: We will add a clear, intuitive explanation for why our derivative-free approach leads to the observed stability, bridging the gap between our empirical results and the core mechanism.
> > > > > > > > >
> > > > > > > > > Once again, thank you for your expert guidance. It will help us to write a much stronger and more impactful paper.

---

### Official Review · Reviewer_JYzA · 2025-07-02

**Clarity:** 3
**Significance:** 3
**Originality:** 3
**Rating:** 5
**Confidence:** 3

**Summary:**

This paper introduces a novel acceleration and distillation method aimed at significantly speeding up the inference process of diffusion models. By leveraging the framework of ordinary differential equation (ODE) integration, their approach learns an efficient solver that approximates the continuous reverse diffusion dynamics. This enables a dramatic reduction in the number of sampling steps required during generation, thereby balancing the trade-off between computational cost and model performance. The proposed method not only preserves the high quality of generated samples but also reduces inference latency, making diffusion models more practical for real-world applications. Through extensive experiments, the authors demonstrate that their approach achieves competitive performance compared to standard diffusion solvers while requiring substantially fewer function evaluations

**Questions:**

See details in above weakness. Authors should answer these questions corresponding to above comments: 1. Key Differences with MeanFlow; 2. Scalability and Applicability; 3. One-step Generation Performance; 4. Clarity on Method Variants

**Ethical Concerns:**

["NO or VERY MINOR ethics concerns only"]

**Final Justification:**

The author has addressed most of my concerns, I am satisified with the rebuttal, I would like to keep my positive rating

**Limitations:**

Yes.  The authors adequately addressed the limitations and potential negative societal impact of their work.

**Quality:**

3

**Strengths And Weaknesses:**

Strengths:

1.The paper introduces a geometrically motivated “secant loss” that bridges the gap between diffusion models (tangent) and numerical integration (secant), offering a fresh perspective on few-step diffusion inference.

2.The use of Monte Carlo integration and Picard iteration to derive the loss is theoretically sound and practically simple.

Weakness:

1.The proposed method shares a similar starting point with MeanFlow, as both optimize the average velocity field based on the integral of the instantaneous velocity field. Could the authors clarify the key differences between their approach and MeanFlow?

2.MeanFlow directly learns the average velocity field, making its objective function simpler, with complexity concentrated in differentiation. In contrast, the proposed method eliminates reliance on differentiation but focuses more on learning a more consistent (straighter) sampling trajectory, which seems more suitable for distillation rather than training a model from scratch. This suggests that MeanFlow might have broader applicability. Could the authors further discuss the scalability of their method?

3.The one-step generation performance appears relatively weak. Could this be due to the limitations of the proposed loss function, which does not directly learn the average velocity field itself? More results on one-step generation would be helpful.

4.The paper introduces many similarly named variants of the method, making the methodology and experiments somewhat difficult to follow. A clearer distinction between these variants would improve readability.

---

> ### Author Rebuttal · Authors · 2025-07-30
>
> We sincerely thank the reviewer for the valuable feedback and for recognizing the theoretical soundness and simplicity of our method. We will address the concerns below.
>
> ### **Question 1. Key Differences with MeanFlow**
>
> The core difference can be summarized succinctly: **To achieve the same objective of fitting the average velocity (the secant), MeanFlow uses differentiation to construct its loss, whereas our method uses integration.** With identical model parameterization, the two methods can be viewed as differential and integral counterparts.
>
> The authors of MeanFlow notes in Section 4.1 that, “However, directly using the average velocity defined by Eq. (3) as ground truth for training a network is intractable, as it requires evaluating an integral during training.” The essential difficulty is, evaluating the integral requires a large amount of samples (i.e., a full inference-like trajectory). But in deep learning, using such many samples to evaluate the integration in one iteration is impractical. MeanFlow circumvents this by using differentiation, while we use Monte Carlo integration, where the key is to convert the many-sample evaluation to one-sample evaluation of the integral.
>
> The above fundamental difference leads to our method's distinctive features: local accuracy in theory, target stability in principle, efficiency in application.
>
> 1.  **Theoretical Soundness:** From a theoretical standpoint, a key feature that distinguishes our method is its ability to achieve local accuracy without requiring explicit differentiation in the loss function, as shown in Theorem 2 and Theorem 3.
>
> 2. **Target Stability:**
>     Another crucial point for stable training is the target stability, i.e., the stability of the prediction target in the loss function. We apologize for not elaborating on this in the main text. For simplicity, let's compare the targets under the OT-FM interpolant ($d$ denotes some distance metric):
>
>       * **Diffusion Models (Flow Matching):** $\mathcal{L}(\theta)=d(\\boldsymbol{v}\_\theta(\\boldsymbol{x}\_t,t)-(-\\boldsymbol{x}_0+\\boldsymbol{z}))$. The target is the velocity estimation $-\\boldsymbol{x}\_0+\\boldsymbol{z}$.
>       * **MeanFlow / CM / CTM (similar):** $\mathcal{L}(\theta)=d(\\boldsymbol{f}\_\theta(\\boldsymbol{x}\_t,t,s)-((-\\boldsymbol{x}_0+\\boldsymbol{z})+(s-t)\\frac{d}{dt}\\boldsymbol{f}\_{\theta^-}(\\boldsymbol{x}\_t,t,s))$. The target is $-\\boldsymbol{x}_0+\\boldsymbol{z} + (s - t)\\frac{d}{dt}\\boldsymbol{f}\_{\theta^-}(\\boldsymbol{x}\_t,t,s)$. The latter item $\\frac{d}{dt}\\boldsymbol{f}\_{\theta^-}(\\boldsymbol{x}\_t,t,s)$ (or the discrete version $\\frac{\\boldsymbol{f}\_{\theta^-}(\\boldsymbol{x}\_t,t,s)-\\boldsymbol{f}\_{\theta^-}(\\boldsymbol{x}\_{t-\Delta t},t-\Delta t,s)}{\Delta t}$) is highly unstable, which has been substantially studied in [1].
>       * **Our Method:** The targets are either the velocity estimation $(-\\boldsymbol{x}\_0+\\boldsymbol{z})$ or the velocity $\\boldsymbol{v}(\\boldsymbol{x}_t,t)$ itself.
>
>     Our method is the **only** one among these to feature a target with stability comparable to that of standard diffusion models. This inherent stability of the prediction target helps explain the robust training behavior we observe. To see the practical target stability, we also conduct an experiment with EDM on CIFAR-10, where we track the specific items in the loss functions, and record the **standard deviation** in the following table. The statistics shown in the table clearly support the stability of our method.
>
>     | Loss (Inspected item)/Training iteration | 10 | 20 | 30 | 40 | 50 | 100 | 200 | 300 | 400 | 500 |
>     | :--- | :---: | :---: | :---: | :---: | :--- | :---: | :---: | :--- | :---: | :---: |
>     | CM ($\\frac{d}{dt}\\boldsymbol{f}\_{\theta^-}(\\boldsymbol{x}\_t,t,s))$) | 0.2627 | 15.4339 | 30.9154 | 40.7293 | 79.1750 | 101.8844 | 97.2164 | 134.4441 | 72.3149 | 97.4782 |
>     | CM ($\\frac{\\boldsymbol{f}\_{\theta^-}(\\boldsymbol{x}\_t,t,s)-\\boldsymbol{f}\_{\theta^-}(\\boldsymbol{x}\_{t-\Delta t},t-\Delta t,s)}{\Delta t}$) | 0.2810 | 10.1618 | 34.4472 | 88.6847 | 39.1018 | 59.5919 | 143.4175 | 62.3149 | 69.5990 | 137.7932 |
>     | SDEI ($\\boldsymbol{v}(\\boldsymbol{x}_r,r)$) | 3.1713 | 3.0427 | 4.1342 | 3.4504 | 3.3060 | 2.6381 | 3.2294 | 3.5089 | 3.4080 | 3.2673 |
>     | STEI ($\\boldsymbol{f}_{\theta^-}(\\boldsymbol{x}_r,r,r)$) | 4.4030 | 4.2680 | 4.1115 | 3.8282 | 4.3783 | 4.5072 | 4.3712 | 4.0905 | 3.9820 | 4.0967 |
>     | SDEE ($\\boldsymbol{v}(\\boldsymbol{x}_r,r)$) | 2.8809 | 3.2181 | 3.5562 | 3.0221 | 3.7394 | 3.0247 | 3.8801 | 3.2247 | 4.0112 | 3.5830 |
>     | STEE ($-\\text{sin}(t)\\boldsymbol{x}_0+\\text{cos}(t)\\boldsymbol{z}$) | 4.2049 | 4.1783 | 4.6112 | 4.0290 | 4.6454 | 4.5360 | 4.9838 | 4.9387 | 5.0872 | 4.3094 |
>
> 3.  **Training Efficiency (Speed and Memory):** In practical application, methods like continuous-time CM and MeanFlow require Jacobian-vector product (JVP) operations to compute the loss. This imposes a significant computational burden, especially in PyTorch (while the MeanFlow authors use JAX, PyTorch remains the more widely adopted framework). To provide a direct comparison, we benchmarked training speed and memory usage under the same EDM setup (the batch size is 512 on 8 A100 GPUs with 64 per GPU) on CIFAR-10 as in our main text.
>
>     | Method | Training Speed (sec/kimg) | Memory Usage (Per GPU, GB) |
>     | :--- | :---: | :---: |
>     | Diffusion Model (Teacher) | 0.57 | 8.69 |
>     | MeanFlow | 1.04 | 38.73 |
>     | SDEI | 0.91 | 9.64 |
>     | STEI | 1.42 | 16.95 |
>     | SDEE | 0.91 | 9.64 |
>     | STEI | 0.72 | 9.34 |
>
>     As the table shows, our method's training overhead is modest. For example, STEI adds only **26%** to the teacher model's training time, whereas MeanFlow adds over **82%**. The difference in memory consumption is even more stark: STEI increases usage by only **7%**, while MeanFlow requires a **346%** increase. And the total training duration required of our method is much less. For example, both fine-tuning from existing teacher models, our method converges with only 50M images on CIFAR-10, whereas sCM requires 200M.
>
> ### **Question 2. Scalability and Applicability**
>
> We would like to clarify that our method, like MeanFlow and Consistency Models, is also based on the exact solution of the diffusion ODE and **learns the same average velocity field**. Our approach does not straighten or alter the ODE trajectory itself, which is a technique used by other methods like 2-Rectified Flow [2]. Therefore, there is no inherent difference in applicability between our method and MeanFlow. Our method is designed for, and has been demonstrated to work effectively in both distillation and training settings. Regarding scalability, we refer to Section 5.3 in the main text, where we scale the model size. The results show that performance consistently improves with a larger model, which can be a supporting evidence for scalability on model size.
>
> ### **Question 3. One-step Generation Performance**
>
> We provide more 1- and 2-step results in the table below.
>
> | Type | Steps | CIFAR-10 FID | ImgNet-256 FID | ImgNet-256 IS |
> | :--- | :---: | :---: | :---: | :---: |
> | SDEI | 1 | 22.67 | 8.97 | 253.25 |
> | SDEI | 2 | 5.88 | 4.81 | 257.56 |
> | STEI | 1 | 36.87 | 7.12 | 241.75 |
> | STEI | 2 | 9.21 | 4.41 | 241.99 |
>
> As explained in Question 2, akin to MeanFlow, our method also learns the average velocity. For explanation of the weaker one-step generation, we can see in Theorem 2 and Theorem 3 in the main text, our loss functions emphasize on **local accuracy** without explicit differentiation. Our loss function is most accurate over shorter integration intervals, and its performance naturally degrades as the step interval becomes very large. We would like to highlight that it is a trade-off. Methods like CM, MeanFlow optimize for global accuracy but do so by introducing either discretization errors or explicit derivative terms in their loss functions, which leads to potential training instability. Our method's "sweet spot" is in the modest few-step regime (e.g., 4 or 8 steps). For scenarios where training speed, memory efficiency, and stability are critical—especially with large models or limited resources—and a modest number of steps is acceptable, our approach provides a highly practical and efficient solution. We acknowledge this limitation and note the performance gap between 1-step and 8-step generation in the paper.
>
> ### **Question 4. Clarity on Method Variants**
>
> The naming convention is based on two key design choices, estimation strategy and training mode. According to Figure 1, the main goal is to learn the secant function $\\boldsymbol{f}(\\boldsymbol{x}_t,t,s)$ such that we can jump from $\\boldsymbol{x}_t$ to $\\boldsymbol{x}_s$ at inference. First, our loss target is set as a random tangent (the green line in the figure). To apply Monte Carlo integration and Picard iteration, we should derive an end point $\\boldsymbol{x}_t$ and a uniformly selected interior point $\\boldsymbol{x}_r$. Considering that we can only access to one of these two at each training iteration, we propose to use the model itself to estimate either the interior point from the true end point, or the end point from the true interior point. We denote the former as “EI”, i.e., “estimating the interior point”, while name the latter as “EE”, i.e., “estimating the end point”. Furthermore, since the target can either be a pretrained diffusion model, or be estimated during training, both “EI” and “EE” include a distillation (D) version and a training (T) version. At the front, we add an “S” denoting “secant losses”, leading to our total four variants: SDEI, STEI, SDEE and STEE.
>
> [1] Simplifying, Stabilizing and Scaling Continuous-time Consistency Models. Cheng Lu and Yang Song. ICLR 2025
>
> [2] Flow straight and fast: Learning to generate and transfer data with rectified flow. Xingchao Liu et al. ICLR 2023

---

> > ### Comment · Reviewer_JYzA · 2025-08-04
> >
> > I am satisified with the rebuttal, I would like to keep my positive rating

---

> > > ### Author Response · Authors · 2025-08-05
> > >
> > > Thank you for your positive feedback. We are pleased that our rebuttal addressed your concerns and will incorporate these discussions into the final version of the manuscript.

---

> ### Comment · Area_Chair_vUT8 · 2025-08-04
> **Action Required: Author–Reviewer Discussion Closing Soon**
>
> Dear Reviewer,
>
>
>
> This is a gentle reminder that the **Author–Reviewer Discussion** phase ends within just three days (by **August 6**). Please take a moment to read the authors’ rebuttal thoroughly and engage in the discussion. Ideally, every reviewer will respond so the authors know their rebuttal has been seen and considered.
>
>
>
> Thank you for your prompt participation!
>
>
>
> Best regards,
>
>
>
> AC

---

### Official Review · Reviewer_v8s1 · 2025-07-03

**Clarity:** 3
**Significance:** 2
**Originality:** 3
**Rating:** 4
**Confidence:** 4

**Summary:**

The paper introduces a novel family of training objectives, termed "secant losses", to train few-step diffusion/flow neural solvers. This is framed as learning the "secant" of the ODE trajectory by matching it to the average of the "tangents" along that path. The final, tractable loss functions are derived using principles from Monte Carlo integration and a Picard-iteration-inspired estimation scheme. The framework is flexible, offering four distinct variants for both distilling a pre-trained teacher model and training a few-step generator from scratch. Empirically, the method demonstrates competitive performance and superior training stability.

**Questions:**

1. I do not quite understand the significance of the sampling of r. The theory demands a uniform weighted average right? So even if one utilizes a different sampling distribution, they need to rebalance via importance sampling.
2. The end-point version of the training requires the model to be able to invert (from data to noise), which could be a waste of model capacity.

**Ethical Concerns:**

["NO or VERY MINOR ethics concerns only"]

**Final Justification:**

Overall, I like the originality of the proposed method, but I still have reservations regarding its practical usefulness and impact. At its current stage, it seems like just another way of doing things rather than a convincingly better way.

**Limitations:**

Yes.

**Quality:**

3

**Strengths And Weaknesses:**

**Strength**
1. A primary strength of this work is its originality and the clarity of its core concept. The essential proposal of the paper, that the average velocity is the average of (instantaneous) velocities, is very intuitive.
2. Writing is clear and easy to follow, and the figures are well-made, e.g. Fig.1.
3. A good collection of ablation studies that validate the design choices and provide practical guidance to users.

**Weakness**
1. Consistency models, or any model that does any-size jumps (like MeanFlow, shortcut models), can be understood as a gradual transition from tangents to secants. The contribution of the work is not really a framework or the contrast between tangents and secants, but rather a new way of training these any-step predictors.
2. Since the contribution of the paper is really a new way of training an existing family of models, we need to compare their empirical performances. It seems like in almost all cases, it lags behind IMM without the help of guidance intervals. The scores on CIFAR are not really convincing either, given the competition. And if we include the comparisons with MeanFlow, the presented results do not inspire confidence in the proposed method. I understand MeanFlow is concurrent work (so I do not really consider this in my evaluation), but it still will greatly hinder the impact of the paper, i.e. why use the proposed losses over others?

---

> ### Author Rebuttal · Authors · 2025-07-30
>
> We sincerely thank the reviewer for the valuable and constructive feedback. We are also grateful for the recognition of our work's originality and clarity. We will address the reviewer's concerns below.
>
> ### **Regarding Weakness 1**
>
> > **Reviewer:** Consistency models, or any model that does any-size jumps (like MeanFlow, shortcut models), can be understood as a gradual transition from tangents to secants. The contribution of the work is not really a framework or the contrast between tangents and secants, but rather a new way of training these any-step predictors.
>
> We agree with the reviewer's assessment. Our core contribution is indeed the development of the **secant expectation algorithms** for training few-step, ODE-based generative models. The discussion of the relationship between tangents and secants is intended to provide a  clearer geometric intuition (like that in MeanFlow paper) and a foundational basis for our model parametrization and loss construction, rather than being the central contribution of this paper.
>
> ### **Regarding Weakness 2**
>
> > **Reviewer:** Since the contribution of the paper is really a new way of training an existing family of models, we need to compare their empirical performances. It seems like in almost all cases, it lags behind IMM without the help of guidance intervals. The scores on CIFAR are not really convincing either, given the competition. And if we include the comparisons with MeanFlow, the presented results do not inspire confidence in the proposed method. [...] why use the proposed losses over others?
>
> We thank the reviewer for raising this important point. We compare the mentioned methods as follows.
>
> **Comparison with IMM:** We wish to highlight a fundamental conceptual distinction between the two methods, which places them in different categories of few-step generative models. IMM adopts distribution-level loss—the Maximum Mean Discrepancy (MMD) loss.  In contrast, our approach is particle-based and grounded in finding the exact solution of the ODE. And our contribution lies in developing a stable and efficient new method for particle-based few-step diffusions. In this context, our work has its unique advantages compared to other related methods as follows.
>
> **Comparison with MeanFlow, Consistency Models (CM), Consistency Trajectory Models (CTM), etc.:** We wish to highlight three key advantages of our approach: local accuracy in theory, target stability in principle, efficiency in application.
>
> 1.  **Theoretical Soundness:** From a theoretical standpoint, a key feature that distinguishes our method is its ability to achieve local accuracy without requiring explicit differentiation in the loss function, as shown in Theorem 2 and Theorem 3.
>
> 2. **Target Stability:**
>     Another crucial point for stable training is the target stability, i.e., the stability of the prediction target in the loss function. We apologize for not elaborating on this in the main text. For simplicity, let's compare the targets under the OT-FM interpolant ($d$ denotes some distance metric):
>
>       * **Diffusion Models (Flow Matching):** $\mathcal{L}(\theta)=d(\\boldsymbol{v}\_\theta(\\boldsymbol{x}\_t,t)-(-\\boldsymbol{x}_0+\\boldsymbol{z}))$. The target is the velocity estimation $-\\boldsymbol{x}\_0+\\boldsymbol{z}$.
>       * **MeanFlow / CM / CTM (similar):** $\mathcal{L}(\theta)=d(\\boldsymbol{f}\_\theta(\\boldsymbol{x}\_t,t,s)-((-\\boldsymbol{x}_0+\\boldsymbol{z})+(s-t)\\frac{d}{dt}\\boldsymbol{f}\_{\theta^-}(\\boldsymbol{x}\_t,t,s))$. The target is $-\\boldsymbol{x}_0+\\boldsymbol{z} + (s - t)\\frac{d}{dt}\\boldsymbol{f}\_{\theta^-}(\\boldsymbol{x}\_t,t,s)$. The latter item $\\frac{d}{dt}\\boldsymbol{f}\_{\theta^-}(\\boldsymbol{x}\_t,t,s)$ (or the discrete version $\\frac{\\boldsymbol{f}\_{\theta^-}(\\boldsymbol{x}\_t,t,s)-\\boldsymbol{f}\_{\theta^-}(\\boldsymbol{x}\_{t-\Delta t},t-\Delta t,s)}{\Delta t}$) is highly unstable, which has been substantially studied in [1].
>       * **Our Method:** The targets are either the velocity estimation $(-\\boldsymbol{x}\_0+\\boldsymbol{z})$ or the velocity $\\boldsymbol{v}(\\boldsymbol{x}_t,t)$ itself.
>
>     Our method is the **only** one among these to feature a target with stability comparable to that of standard diffusion models. This inherent stability of the prediction target helps explain the robust training behavior we observe. To see the practical target stability, we also conduct an experiment with EDM on CIFAR-10, where we track the specific items in the loss functions, and record the **standard deviation** in the following table. The statistics shown in the table clearly support the stability of our method.
>
>     | Loss (Inspected item)/Training iteration | 10 | 20 | 30 | 40 | 50 | 100 | 200 | 300 | 400 | 500 |
>     | :--- | :---: | :---: | :---: | :---: | :--- | :---: | :---: | :--- | :---: | :---: |
>     | CM ($\\frac{d}{dt}\\boldsymbol{f}\_{\theta^-}(\\boldsymbol{x}\_t,t,s))$) | 0.2627 | 15.4339 | 30.9154 | 40.7293 | 79.1750 | 101.8844 | 97.2164 | 134.4441 | 72.3149 | 97.4782 |
>     | CM ($\\frac{\\boldsymbol{f}\_{\theta^-}(\\boldsymbol{x}\_t,t,s)-\\boldsymbol{f}\_{\theta^-}(\\boldsymbol{x}\_{t-\Delta t},t-\Delta t,s)}{\Delta t}$) | 0.2810 | 10.1618 | 34.4472 | 88.6847 | 39.1018 | 59.5919 | 143.4175 | 62.3149 | 69.5990 | 137.7932 |
>     | SDEI ($\\boldsymbol{v}(\\boldsymbol{x}_r,r)$) | 3.1713 | 3.0427 | 4.1342 | 3.4504 | 3.3060 | 2.6381 | 3.2294 | 3.5089 | 3.4080 | 3.2673 |
>     | STEI ($\\boldsymbol{f}_{\theta^-}(\\boldsymbol{x}_r,r,r)$) | 4.4030 | 4.2680 | 4.1115 | 3.8282 | 4.3783 | 4.5072 | 4.3712 | 4.0905 | 3.9820 | 4.0967 |
>     | SDEE ($\\boldsymbol{v}(\\boldsymbol{x}_r,r)$) | 2.8809 | 3.2181 | 3.5562 | 3.0221 | 3.7394 | 3.0247 | 3.8801 | 3.2247 | 4.0112 | 3.5830 |
>     | STEE ($-\\text{sin}(t)\\boldsymbol{x}_0+\\text{cos}(t)\\boldsymbol{z}$) | 4.2049 | 4.1783 | 4.6112 | 4.0290 | 4.6454 | 4.5360 | 4.9838 | 4.9387 | 5.0872 | 4.3094 |
>
> 3.  **Training Efficiency (Speed and Memory):** In practical application, methods like continuous-time CM and MeanFlow require Jacobian-vector product (JVP) operations to compute the loss. This imposes a significant computational burden, especially in PyTorch (while the MeanFlow authors use JAX, PyTorch remains the more widely adopted framework). To provide a direct comparison, we benchmarked training speed and memory usage under the same EDM setup (the batch size is 512 on 8 A100 GPUs with 64 per GPU) on CIFAR-10 as in our main text.
>
>     | Method | Training Speed (sec/kimg) | Memory Usage (Per GPU, GB) |
>     | :--- | :---: | :---: |
>     | Diffusion Model (Teacher) | 0.57 | 8.69 |
>     | MeanFlow | 1.04 | 38.73 |
>     | SDEI | 0.91 | 9.64 |
>     | STEI | 1.42 | 16.95 |
>     | SDEE | 0.91 | 9.64 |
>     | STEI | 0.72 | 9.34 |
>
>     As the table shows, our method's training overhead is modest. For example, STEI adds only **26%** to the teacher model's training time, whereas MeanFlow adds over **82%**. The difference in memory consumption is even more stark: STEI increases usage by only **7%**, while MeanFlow requires a **346%** increase. And the total training duration required of our method is much less. For example, both fine-tuning from existing teacher models, our method converges with only 50M images on CIFAR-10, whereas sCM requires 200M.
>
> In summary, while our method may not achieve state-of-the-art FID scores, it presents a compelling trade-off. For scenarios where training speed, memory efficiency, and stability are critical—especially when working with large models or limited computational resources, and modest step numbers like 4 or 8 are acceptable, our approach provides a highly practical and efficient solution.
>
> ### **Regarding Question 1**
>
> > **Reviewer:** I do not quite understand the significance of the sampling of $r$. The theory demands a uniform weighted average right? So even if one utilizes a different sampling distribution, they need to rebalance via importance sampling.
>
> Mathematically, a uniform weighted average and a non-uniform distribution rebalanced with importance sampling are equivalent. However, in the context of deep learning, the choice of sampling distribution corresponds to how we allocate the **training effort** across different time intervals. We apologize for not clarifying this motivation in the main text. Inspired by recent findings in the diffusion model literature (e.g., EDM[2], SD3[3], FasterDiT[4]), where focusing training efforts on specific time intervals was shown to accelerate convergence, we decide to explore if a similar strategy would benefit our method. We experiment with different distributions for sampling the time points $r$. However, our empirical results indicate that allocating training effort uniformly across all intervals works best.
>
> ### **Regarding Question 2**
>
> > **Reviewer:** The end-point version of the training requires the model to be able to invert (from data to noise), which could be a waste of model capacity.
>
> The end-point variant of our training algorithm indeed requires the model to learn the inversion from data back to noise, which could command additional model capacity compared to the intermediate-point variant. And here we would like to emphasize that there is a trade-off between model capacity and training computation. As shown in Table 1 in the main text, the benefit of this approach is a more efficient training step: while STEE may use more model capacity, it saves **two forward pass evaluations and one backpropagation step** per iteration compared to STEI.
>
> [1] Simplifying, Stabilizing and Scaling Continuous-time Consistency Models. Cheng Lu and Yang Song. ICLR 2025
>
> [2] Elucidating the design space of diffusion-based generative models. Tero Karras et al. NeurIPS 2022
>
> [3] Scaling rectified flow transformers for high-resolution image synthesis. Patrick Esser et al. ICML 2024
>
> [4] FasterDiT: Towards faster diffusion transformers training without architecture modification. Jingfeng Yao et al. NeurIPS 2024

---

> > ### Comment · Area_Chair_vUT8 · 2025-08-04
> > **Action Required: Author–Reviewer Discussion Closing Soon**
> >
> > Dear Reviewer,
> >
> >
> >
> > This is a gentle reminder that the **Author–Reviewer Discussion** phase ends within just three days (by **August 6**). Please take a moment to read the authors’ rebuttal thoroughly and engage in the discussion. Ideally, every reviewer will respond so the authors know their rebuttal has been seen and considered.
> >
> >
> >
> > Thank you for your prompt participation!
> >
> >
> >
> > Best regards,
> >
> >
> >
> > AC

---

> > > ### Comment · Area_Chair_vUT8 · 2025-08-05
> > > **Action Required: Author–Reviewer Discussion Closing Soon**
> > >
> > > Dear Reviewer,
> > >
> > > A gentle reminder that the extended Author–Reviewer Discussion phase ends on **August 8 (AoE)**.
> > >
> > > Please read the authors’ rebuttal and participate in the discussion **ASAP**. Regardless of whether your concerns have been addressed, kindly communicate:
> > >
> > > - If resolved — please acknowledge.
> > >
> > > - If unresolved — please specify what remains.
> > >
> > > The “Mandatory Acknowledgement” should only be submitted **after**:
> > >
> > > - Reading the rebuttal,
> > >
> > > - Engaging in author discussion,
> > >
> > > - Completing the "Final Justification" (and updating your rating).
> > >
> > > **As per policy, I may flag any missing or unresponsive reviews and deactivate them once additional reviewer feedback has been posted.**
> > >
> > > Thank you for your timely and thoughtful contributions.
> > >
> > >
> > >
> > >
> > > Best regards,
> > >
> > > AC

---

> > ### Comment · Reviewer_v8s1 · 2025-08-06
> >
> > I thank the authors for their detailed response. Overall, I like the originality of the proposed method, but I still have reservations regarding its practical usefulness and impact. Hence, I decide to keep my original borderline assessment. A more detailed response to the rebuttal is as follows.
> >
> > I think we all agree on the fact that this paper's contribution is on the new way of training these kinds of few-step, ODE-based models rather than the framework itself. With that in mind, the fact that the performance is not better than some of the baselines raises concerns about the actual impact of the proposal. Regarding the stability factor, CM-like models have gone through many iterations of stabilization, and in my own experience, are not hard to train (especially the latest ones like sCM). Better stability only in terms of loss values is not really a convincing advantage. Ideally, a more stable objective should be more stable in terms of hyperparameters, and better yet, just better performance. Right now, this does not seem to be the case. And the current results may even suggest that these stabilities come at a cost of worse performances. In terms of efficiency, CM and the proposed method have the exact same requirement. One can always switch the autodiff with numerical differentiation (discrete-time CM, 1024 steps will be very close to continuous-time and train well in practice) to work with PyTorch.

---

> > > ### Author Response · Authors · 2025-08-07
> > >
> > > Thank you for your detailed and insightful follow-up. We sincerely appreciate your perspective and agree with many of your astute observations.
> > >
> > > We agree with your assessment that, based on the current results, our method does not yet demonstrate superior performance over a heavily-tuned baseline like sCM. This raises a valid question about its practical impact. Our intention, however, is not to claim a new state-of-the-art in performance, but to propose a new training paradigm for few-step, ODE-based models that offers a different set of advantages, primarily centered on robustness and transferability.
> > >
> > > The key insight, is our method's unique parallel to the standard diffusion model framework, whose loss target is identical or analogous to that of a standard diffusion model. This is not just a superficial similarity; it provides a strong inductive bias. It suggests that if a standard diffusion model is known to work well on a new dataset or modality, our method is highly likely to succeed as well, without requiring the extensive, bespoke tuning that often accompanies novel architectures or loss formulations.
> > >
> > > This addresses what we see as a practical challenge with the CM paradigm. While we agree that a skilled practitioner can stabilize modern CMs, our own experience mirrors a broader challenge in the community. We have observed that CMs can be highly sensitive to hyperparameters. For example: the training can be sensitive to FP32 vs. FP16 precision; many implementations on EDM switch from Adam (standard for diffusion) to RAdam or other alternatives specifically for CMs to achieve stability. In our own experiments, we found that a stable configuration for CIFAR-10 did not transfer well to a different architecture like DiT, requiring substantial effort to even begin training without divergence.
> > >
> > > In contrast, our method's reliance on the same core components as diffusion models (e.g., loss target, loss weighting, sampling strategy) suggests a much smoother path for transfer and adoption across the diverse tasks and modalities where diffusion models have already succeeded, but CMs have not yet been widely applied.
> > >
> > > Therefore, we believe the primary impact of our work lies in providing a more robust, predictable, and easily transferable training algorithm. We see this as a valuable alternative path for the community, especially for those working on new domains where extensive, architecture-specific tuning is infeasible.
> > >
> > > Thank you again for your critical and thought-provoking feedback. It has been instrumental in helping us refine our perspective and will be central to how we frame the contributions of this work moving forward.

---

### Comment · Area_Chair_vUT8 · 2025-08-01
**Kindly Engage with Author Responses**

Dear Reviewers,


The authors have submitted their responses to your reviews. At your earliest convenience, please take a moment to engage with their replies. Your continued discussion and clarifications will be invaluable in ensuring a fair and constructive review process for all parties.

Thank you again for your thoughtful contributions and dedication!


Warm regards,

Your AC

---

### Decision · Program_Chairs · 2025-09-17

**Decision:**

Accept (poster)

**Comment:**

While Reviewer v8s1 raised concerns regarding the stability v.s. performance claim of the proposed loss, the reviewers overall appreciated the novelty and interest of the method for learning ODE-based generative models. Given the proliferation of similar one-step ODE-based approaches with related parameterizations and objectives, the authors are encouraged to provide more detailed comparisons to better highlight their contribution. Overall, I recommend **acceptance**.